# More Edits, More Stable: Understanding the Lifelong Normalization in Sequential Model Editing

Xin Ma[1]  Wei Chen[1]  Qi Liu[1]  Derong Xu[1,2]  Zhi Zheng[1]  Tong Xu[1]  Enhong Chen[1]

## Abstract

Lifelong Model Editing aims to continuously update evolving facts in Large Language Models while preserving unrelated knowledge and general capabilities, yet it remains plagued by catastrophic forgetting and model collapse. Empirically, we find that recent editors resilient over long horizons share the same core strategy: **Lifelong Normalization (LN)**, which normalizes value gradients using running statistics. Removing LN causes immediate performance collapse, and we observe a counter-intuitive **positive cumulative effect** where early edits can promote the success of future edits. Yet the mechanism of LN remains a "black box", leaving its precise role in lifelong stability poorly understood. In this work, we provide the **first** theoretical account of LN in the lifelong regime. Our analysis reveals a self-reinforcing stability loop and proves that, when combined with ridge-regularized regression, LN yields parameter updates with **asymptotic orthogonality** and **bounded norms**, directly mitigating forgetting and systemic collapse. Based on these insights, we derive STABLEEDIT, which strengthens this stability loop via an explicit warm-up stage and full whitening, improving long-horizon stability at minimal overhead. Extensive experiments validate our theory and demonstrate competitive performance. Our code is available at STABLEEDIT.

## 1. Introduction

Large Language Models (LLMs) store a vast amount of world knowledge in their parameters learned from static pre-

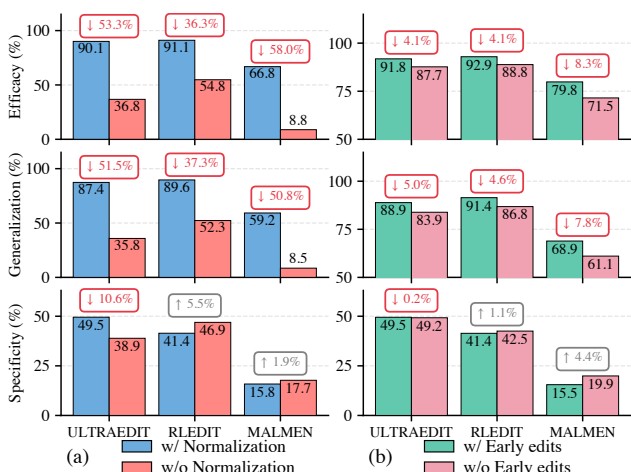

*Figure 1.* Editing results on Llama-3-8B-Instruct under *with* vs. *without* LN. (a) Performance of ULTRAEDIT and RLEDIT (20K total edits over $T$=200 steps) alongside MALMEN (2K edits over $T$=20 steps). (b) Performance on the same later segment (steps $\frac{T}{2}$ to $T$), comparing editing from the full sequence 1 to $T$ versus starting from $\frac{T}{2}$ to $T$ without early edits. Downward arrows ($\downarrow$) indicate the relative drop (%) when removing LN (w/ $\rightarrow$ w/o).

training corpora (Petroni et al., 2019; Brown et al., 2020; Zhao et al., 2023). As the world changes, however, parts of this knowledge become outdated (Lazaridou et al., 2021; Cao et al., 2021). Updating an LLM via full retraining or continual pre-training is typically prohibitively expensive (Hartvigsen et al., 2023), which motivates Lifelong Model Editing (LME): a sequential process that continuously updates targeted facts efficiently while preserving unrelated knowledge and general abilities (Sinitsin et al., 2020; Yao et al., 2023; Zhang et al., 2024). LME offers a practical mechanism to correct factual errors (Xu et al., 2024; Guo et al., 2025b), mitigate biases (Li et al., 2023; Gallegos et al., 2024), and improve safety (El-Mhamdi et al., 2022; Huang et al., 2024), ultimately achieving sustainable lifelong learning (Tao et al., 2024; Zheng et al., 2025).

This lifelong setting, however, introduces two severe challenges: (1) catastrophic forgetting, where new edits interfere with previously edited knowledge (Wang et al., 2024; Gupta et al., 2024a; Luo et al., 2025); and (2) model collapse, where the accumulation of parameter updates causes the model to drift systematically away from its original state

[1]University of Science and Technology of China [2]City University of Hong Kong. Correspondence to: Zhi Zheng <zhengzhi97@ustc.edu.cn>, Tong Xu <tongxu@ustc.edu.cn>, Enhong Chen <cheneh@ustc.edu.cn>.

*Proceedings of the 43rd International Conference on Machine Learning*, Seoul, South Korea. PMLR 306, 2026. Copyright 2026 by the author(s).

(Gupta et al., 2024b; Ma et al., 2025). Together, these challenges make long-horizon editing stability notoriously difficult. Remarkably, despite these obstacles, recent editors such as ULTRAEDIT (Gu et al., 2025) and RLEDIT (Li et al., 2025) exhibit strong resilience in LME scenarios. Examining these methods, we identify a striking commonality: they all incorporate the same normalization module, which we term **Lifelong Normalization (LN)**. LN normalizes the value gradient using running statistics (mean and variance) that are continuously updated throughout the edit sequence (see Sections 2 and 3.1 for details). Empirically, ablating LN triggers immediate performance collapse (Figure 1(a)), confirming its critical role in maintaining long-term stability.

To further investigate how LN contributes to LME, we conduct a second experiment (Figure 1(b)). We evaluate the same set of *later* edits under two settings: running the *full* edit sequence from the beginning, or starting mid-way by executing only the later edits (skipping the initial *early* edits). A natural intuition is that early edits should become a growing burden: each update consumes plasticity or accumulates constraints, making subsequent edits progressively harder. Surprisingly, we observe the opposite: running the full sequence yields better performance on later edits than starting mid-way. We term this counter-intuitive phenomenon the **positive cumulative effect**, indicating that with LN, early edits can actually stabilize the editing process and facilitate future edits. However, how LN enables such long-term stability remains poorly understood. This motivates our core research question:

> **Q1.** *In LME scenarios, what mechanism makes LN indispensable for long-term stability and produces the "positive cumulative effect" in which early edits facilitate future ones?*

In response, we provide the **first** theoretical framework to demystify LN in LME. Our analysis reveals that LN forms a **self-reinforcing stability loop** across edit steps. At a high level, LN continuously aggregates historical information to build an accurate estimate of the task-specific value gradient distribution. Using this estimate, whitening-based normalization and ridge-regularized regression yield calibrated parameter updates. These updates exhibit asymptotic orthogonality, alleviating interference (combating catastrophic forgetting) and maintain bounded norms to mitigate systemic drift (combating model collapse). Ultimately, this synergy transforms LME into a virtuous cycle where each successful edit preserves a stable environment for the next.

Based on these insights, we derive STABLEEDIT, which strengthens the stability loop via a warm-up stage and full whitening, leading to improved long-horizon stability at minimal overhead. Our contributions are threefold:

- **Empirically,** we identify LN as a cornerstone for stability in LME and uncover the counter-intuitive positive cumulative effect where early edits facilitate later ones.

- **Theoretically,** we provide the first theoretical framework to demystify LN, revealing a self-reinforcing loop that enforces asymptotic orthogonality of updates and norm stability.

- **Methodologically,** we derive STABLEEDIT, which utilizes an explicit warm-up and full whitening to maximize the stability loop. Extensive experiments validate our theory and the proposed editor.

## 2. Preliminaries

### 2.1. Lifelong Model Editing

We formulate LME as a sequential modification of LLMs, starting from an initial model $f_{\boldsymbol{W}_0}$. At each step $t$, the model receives a batch of $n_t$ edits $\mathcal{E}_t = \{\boldsymbol{e}_t^i = (\boldsymbol{x}_t^i, \boldsymbol{y}_t^i)\}_{i=1}^{n_t}$. The goal is to obtain updated parameters $\boldsymbol{W}_t = \boldsymbol{W}_{t-1} + \boldsymbol{\Delta}_t$ that satisfy three key properties: (1) *Efficacy*, which ensures the model accurately recalls the specific facts in the current batch, i.e., $f_{\boldsymbol{W}_t}(\boldsymbol{x}_t^i) = \boldsymbol{y}_t^i$; (2) *Generalization*, requiring the update to extend to semantically equivalent inputs $\boldsymbol{x}'$ such that $f_{\boldsymbol{W}_t}(\boldsymbol{x}') = \boldsymbol{y}_t$; and (3) *Specificity*, which preserves behavior on unrelated knowledge $\mathcal{D}_{\text{retain}}$, demanding minimal model drift, i.e., $f_{\boldsymbol{W}_t}(\boldsymbol{x}) \approx f_{\boldsymbol{W}_{t-1}}(\boldsymbol{x})$ for all $\boldsymbol{x} \in \mathcal{D}_{\text{retain}}$.

### 2.2. Lifelong Normalization Strategy

As highlighted in Section 1, LN serves as the cornerstone of stability for a broad class of model editors (Mitchell et al., 2022a; Tan et al., 2024; Li et al., 2025; Gu et al., 2025). This strategy is commonly motivated as stabilizing the distribution of the internal representations and gradients used to form parameter updates.

For each edit instance $i$ at step $t$, two vectors are extracted from a target module (e.g., the MLP's down_proj layer): the input hidden state $\boldsymbol{h}_t^i$ and the value gradient $\boldsymbol{v}_t^i$, which is the gradient of the loss with respect to the editable module output $\boldsymbol{u}_t^i = \boldsymbol{W}_{t-1}\boldsymbol{h}_t^i$. These are concatenated into a joint vector $\boldsymbol{z}_t^i = [\boldsymbol{h}_t^i; \boldsymbol{v}_t^i]$ and standardized element-wise: $\tilde{\boldsymbol{z}}_t^i = \frac{\boldsymbol{z}_t^i - \boldsymbol{m}_t}{\boldsymbol{\sigma}_t}$, where $\boldsymbol{m}_t$ and $\boldsymbol{\sigma}_t$ denote the running mean and standard deviation of all processed vectors. Crucially, these statistics are updated online to adapt to distribution shifts. Let $N_t = \sum_{j=1}^t n_j$ denote the cumulative sample count, $\boldsymbol{S}_t = (N_t - 1)\boldsymbol{\sigma}_t^2$ the sum of squared deviations, and $\boldsymbol{\delta}_t = \bar{\boldsymbol{z}}_t - \boldsymbol{m}_{t-1}$ the mean shift, with $\bar{\boldsymbol{z}}_t$ the current batch mean. The running statistics are updated element-wise as follows:

$$\boldsymbol{m}_t \leftarrow \boldsymbol{m}_{t-1} + \tfrac{n_t}{N_t}\boldsymbol{\delta}_t, \tag{1}$$

$$\boldsymbol{S}_t \leftarrow \boldsymbol{S}_{t-1} + n_t \operatorname{Var}(\boldsymbol{z}_t) + \tfrac{N_{t-1}n_t}{N_t}\boldsymbol{\delta}_t^2, \tag{2}$$

where $\text{Var}(\boldsymbol{z}_t)$ denotes the element-wise variance of the vectors in the current batch at step $t$. The utilization of the normalized vectors $\{\tilde{\boldsymbol{z}}_t^i\}$ diverges across architectures. ULTRAEDIT (Gu et al., 2025) uses $\tilde{\boldsymbol{z}}_t^i$ to formulate a ridge regression objective and solves for the update $\boldsymbol{\Delta}_t$ in closed form. Conversely, in hypernetwork-based methods (e.g., RLEDIT (Li et al., 2025)), $\tilde{\boldsymbol{z}}_t^i$ is first fed to an auxiliary network and the update is then obtained by solving a ridge-regression problem.

# 3. Analysis of the Lifelong Normalization Mechanism

This section demystifies LN by establishing a theoretical chain from statistical tracking to parameter updates. We first model LME as a Bayesian recursive tracking process to capture non-stationary gradient distributions (Section 3.1). We then prove that accumulated edits progressively reduce distribution estimation errors (Section 3.2). Furthermore, we demonstrate that precise estimates allow LN combined with ridge regression to enforce crucial properties on parameter updates, namely asymptotic orthogonality and bounded update norms, which directly combat catastrophic forgetting and model collapse (Section 3.3).

## 3.1. Dynamic Gradient Modeling and Bayesian Recursive Tracking

**Problem Setup**[1] At each step $t \geq 1$, we draw a batch of $n_t$ editing requests $\{\boldsymbol{e}_t^i\}_{i=1}^{n_t}$ i.i.d. from the edit distribution $\mathcal{D}$ and update editable parameters across multiple layers. Let $\boldsymbol{W}_{t-1,l} \in \mathbb{R}^{d \times d_h}$ denote the editable weight matrix at layer $l$ *before* step $t$; applying the update $\boldsymbol{\Delta}_{t,l}$ yields $\boldsymbol{W}_{t,l}$. For each edit $i$, the *input hidden state* $\boldsymbol{h}_{t,l}^i \in \mathbb{R}^{d_h}$ and the corresponding *value gradient* $\boldsymbol{v}_{t,l}^i \in \mathbb{R}^d$ at the target token position[2] are extracted, where $\boldsymbol{v}_{t,l}^i$ is defined as the gradient of the loss $\ell$ with respect to the module's output $\boldsymbol{u}_{t,l}^i = \boldsymbol{W}_{t-1,l}\boldsymbol{h}_{t,l}^i$:

$$\boldsymbol{v}_{t,l}^i := \nabla_{\boldsymbol{u}_{t,l}^i} \ell(\boldsymbol{e}_t^i; \boldsymbol{W}_{t-1,l}) \triangleq \boldsymbol{g}(\boldsymbol{e}_t^i; \boldsymbol{W}_{t-1,l}). \quad (3)$$

As the parameters evolve over steps, the induced value gradient distribution changes accordingly. We model this distribution by its first two moments:

$$\boldsymbol{\mu}_{t,l} := \mathbb{E}_{\boldsymbol{e} \sim \mathcal{D}}\left[\boldsymbol{g}\left(\boldsymbol{e}; \boldsymbol{W}_{t-1,l}\right)\right],$$
$$\boldsymbol{\Sigma}_{t,l} := \text{Cov}_{\boldsymbol{e} \sim \mathcal{D}}\left[\boldsymbol{g}\left(\boldsymbol{e}; \boldsymbol{W}_{t-1,l}\right)\right].$$

This yields a *moving target*: $(\boldsymbol{\mu}_{t,l}, \boldsymbol{\Sigma}_{t,l})$ drift with $t$. This non-stationarity is detrimental to the LN strategy: since LN

---

[1]We use bold symbols for vectors and matrices and non-bold symbols for scalars in this work.

[2]Without loss of generality, we focus on single-token examples; the analysis extends to multi-token settings.

at step $t$ requires the *current* statistics, stale estimates lead to biased centering or scaling and hence perturb the effective parameter updates. Therefore, we need a principled way to *track* $(\boldsymbol{\mu}_{t,l}, \boldsymbol{\Sigma}_{t,l})$ using only a limited number of samples.

To make this precise, we impose mild regularity conditions:

**Assumption 3.1** (Regularity Conditions). *For all editable layers $l$ and steps $t$, there exist finite constants $L, M, \sigma_- > 0, \sigma_+ > 0$ such that*

(a) **Smoothness:** $\boldsymbol{g}(\boldsymbol{e}; \boldsymbol{W})$ *is $L$-Lipschitz continuous w.r.t. $\boldsymbol{W}$, and $\sup_{\boldsymbol{W}} \mathbb{E}[\|\boldsymbol{g}(\boldsymbol{e}; \boldsymbol{W})\|_2^2] \leq M$.*

(b) **Spectrum Bounds:** *The true covariance $\boldsymbol{\Sigma}_{t,l}$ is non-degenerate: $\sigma_- \boldsymbol{I} \preceq \boldsymbol{\Sigma}_{t,l} \preceq \sigma_+ \boldsymbol{I}$.*

Under Assumption 3.1, the drift of $(\boldsymbol{\mu}_{t,l}, \boldsymbol{\Sigma}_{t,l})$ (measured by $\|\boldsymbol{\mu}_{t,l} - \boldsymbol{\mu}_{t-1,l}\|_2$ and $\|\boldsymbol{\Sigma}_{t,l} - \boldsymbol{\Sigma}_{t-1,l}\|_2$) can be bounded by the magnitude of the update $\boldsymbol{\Delta}_{t,l} = \boldsymbol{W}_{t,l} - \boldsymbol{W}_{t-1,l}$ (formal statement in Lemma D.1). Hence, if updates are controlled, then the gradient distribution *evolves slowly*; later we show LN combined with ridge-regularized regression yields such control (Section 3.3). Consequently, statistics from step $t-1$ remain a strong warm-start for step $t$.

We thus formulate distribution tracking as a *recursive Bayesian inference* task. Concretely, we model $\mathcal{D}_{t,l} = \{\boldsymbol{v}_{t,l}^i\}_{i=1}^{n_t}$ as conditionally i.i.d. samples from a multivariate Gaussian working model $\mathcal{N}_d(\boldsymbol{\mu}_{t,l}, \boldsymbol{\Sigma}_{t,l})$ to capture the critical first- and second-order statistics, and place a Normal–Inverse–Wishart (NIW) conjugate prior on $(\boldsymbol{\mu}_{t,l}, \boldsymbol{\Sigma}_{t,l})$. Conjugacy ensures the posterior remains NIW, yielding an explicit step-to-step recursion for the NIW hyperparameters. Let the sample mean be $\bar{\boldsymbol{v}}_{t,l} = \frac{1}{n_t}\sum_{i=1}^{n_t} \boldsymbol{v}_{t,l}^i$ and scatter matrix $\boldsymbol{S}_{t,l} = \sum_{i=1}^{n_t}(\boldsymbol{v}_{t,l}^i - \bar{\boldsymbol{v}}_{t,l})(\boldsymbol{v}_{t,l}^i - \bar{\boldsymbol{v}}_{t,l})^\top$. We derive the following recursive update rule.

**Theorem 3.2.** *Suppose the prior over $(\boldsymbol{\mu}_{t,l}, \boldsymbol{\Sigma}_{t,l})$ follows a NIW distribution derived from step $t - 1$,*

$$p(\boldsymbol{\mu}_{t,l}, \boldsymbol{\Sigma}_{t,l}) = NIW(\boldsymbol{m}_{t-1,l}, \kappa_{t-1,l}, \boldsymbol{\Psi}_{t-1,l}, \nu_{t-1,l}),$$

*and the likelihood is conditionally i.i.d. Gaussian,*

$$p(\mathcal{D}_{t,l} \mid \boldsymbol{\mu}_{t,l}, \boldsymbol{\Sigma}_{t,l}) = \prod_{i=1}^{n_t} \mathcal{N}_d(\boldsymbol{v}_{t,l}^i \mid \boldsymbol{\mu}_{t,l}, \boldsymbol{\Sigma}_{t,l}).$$

*Then the posterior $p(\boldsymbol{\mu}_{t,l}, \boldsymbol{\Sigma}_{t,l} \mid \mathcal{D}_{t,l})$ is also NIW, namely $NIW(\boldsymbol{m}_{t,l}, \kappa_{t,l}, \boldsymbol{\Psi}_{t,l}, \nu_{t,l})$, with updated hyperparameters:*

$$\kappa_{t,l} = \kappa_{t-1,l} + n_t, \quad \nu_{t,l} = \nu_{t-1,l} + n_t,$$
$$\boldsymbol{m}_{t,l} = \frac{\kappa_{t-1,l}\boldsymbol{m}_{t-1,l} + n_t\bar{\boldsymbol{v}}_{t,l}}{\kappa_{t,l}},$$
$$\boldsymbol{\Psi}_{t,l} = \boldsymbol{\Psi}_{t-1,l} + \boldsymbol{S}_{t,l} + \frac{\kappa_{t-1,l}n_t}{\kappa_{t,l}}\boldsymbol{Q}_{t,l},$$

*where $\boldsymbol{Q}_{t,l} = (\bar{\boldsymbol{v}}_{t,l} - \boldsymbol{m}_{t-1,l})(\bar{\boldsymbol{v}}_{t,l} - \boldsymbol{m}_{t-1,l})^\top$.*

**Corollary 3.3** (Posterior Expectations). *The posterior expectations of the distributional parameters are*

$$\hat{\boldsymbol{\mu}}_{t,l} = \boldsymbol{m}_{t,l}, \ \ \hat{\boldsymbol{\Sigma}}_{t,l} = \frac{\boldsymbol{\Psi}_{t,l}}{\nu_{t,l} - d - 1}, \ \ if \ \nu_{t,l} > d + 1. \quad (4)$$

Here, $(\boldsymbol{m}, \kappa, \boldsymbol{\Psi}, \nu)$ are NIW hyperparameters (mean, strength, scale, dof). Theorem 3.2 provides a recursive Bayesian view of LN: sequential editing continuously refines the gradient distribution, where the estimate from step $t - 1$ is carried into step $t$ and corrected using the new batch (posterior→prior→posterior). Since conjugacy guarantees the updated posterior stays in the same family, tracking the drifting gradient distribution reduces to updating $(\boldsymbol{m}, \kappa, \boldsymbol{\Psi}, \nu)$ via simple algebraic recursions. As a result, we can stably and efficiently track the *current* $(\boldsymbol{\mu}_{t,l}, \boldsymbol{\Sigma}_{t,l})$ using limited samples. Combined with Corollary 3.3, these recursions yield running estimates $(\hat{\boldsymbol{\mu}}_{t,l}, \hat{\boldsymbol{\Sigma}}_{t,l})$ for the normalization step in LN-based editors. In practice, many editors use diagonal normalization for efficiency; taking only $\text{diag}(\hat{\boldsymbol{\Sigma}}_{t,l})$ recovers running mean and variance updates in Equations (1) and (2) up to standard degrees-of-freedom corrections. The theorem is proved in Appendix D.2.

### 3.2. Statistical Foundation of Positive Cumulative Effect

Having formalized LN as a recursive tracking process in Section 3.1, we now study the statistical behavior of the Bayesian estimator in a non-stationary environment. In particular, we analyze whether early edits can improve subsequent distribution estimation relative to a cold start.

To make this comparison explicit, we split the editing process into a *previous phase* of $r \geq 1$ steps and a *current phase* of $t \geq 1$ steps (global step $r + t$). Without loss of generality, we assume a constant batch size $n_t = n$ for $t \geq 1$ and define the expected estimation errors for the mean and covariance at step $t$ of the current phase as:

$$\text{MSE}_{t,l}^{(\boldsymbol{\mu},\text{cur})} := \mathbb{E}\|\boldsymbol{m}_{t,l}^{(\text{cur})} - \boldsymbol{\mu}_{t,l}^{(\text{cur})}\|_2^2,$$

$$E_{t,l}^{(\boldsymbol{\Sigma},\text{cur})} := \mathbb{E}\|\hat{\boldsymbol{\Sigma}}_{t,l}^{(\text{cur})} - \boldsymbol{\Sigma}_{t,l}^{(\text{cur})}\|_2,$$

where $\boldsymbol{m}_{t,l}^{(\text{cur})}$ and $\hat{\boldsymbol{\Sigma}}_{t,l}^{(\text{cur})}$ are the point estimators from Corollary 3.3, with the initial condition inherited from the previous phase, i.e., $\boldsymbol{m}_{0,l}^{(\text{cur})} = \boldsymbol{m}_{r,l}^{(\text{pre})}$ and $\hat{\boldsymbol{\Sigma}}_{0,l}^{(\text{cur})} = \hat{\boldsymbol{\Sigma}}_{r,l}^{(\text{pre})}$. Here $\text{MSE}_{t,l}^{(\boldsymbol{\mu},\text{cur})}$ is the mean-squared error, while $E_{t,l}^{(\boldsymbol{\Sigma},\text{cur})}$ uses the spectral norm to control the worst-case scaling relevant to normalization stability. The expectation $\mathbb{E}[\cdot]$ is taken over all data samples observed up to the current step.

Section 3.1 relates distribution drift to the parameter update norm, suggesting that $(\boldsymbol{\mu}_{t,l}, \boldsymbol{\Sigma}_{t,l})$ evolve smoothly across steps under controlled parameter updates. To obtain long-horizon guarantees from finitely many samples per step, we

analyze a sufficient *trackable regime* in which these drift magnitudes decay.

**Assumption 3.4** (Trackable Drift Regime). For constants $\epsilon, \epsilon' > 0$:

$$D_j^{(\boldsymbol{\mu},\text{cur})} = O((r + j)^{-(1+\epsilon/2)}),$$

$$\Delta_j^{(\boldsymbol{\Sigma},\text{cur})} = O((r + j)^{-(1+\epsilon')}).$$

where $D_j^{(\boldsymbol{\mu},\text{cur})}$ and $\Delta_j^{(\boldsymbol{\Sigma},\text{cur})}$ upper-bound $\|\boldsymbol{\mu}_{j,l}^{(\text{cur})} - \boldsymbol{\mu}_{j-1,l}^{(\text{cur})}\|_2$ and $\|\boldsymbol{\Sigma}_{j,l}^{(\text{cur})} - \boldsymbol{\Sigma}_{j-1,l}^{(\text{cur})}\|_2$, respectively.

Since $(\boldsymbol{\mu}_{t,l}, \boldsymbol{\Sigma}_{t,l})$ in Assumption 3.4 are latent and cannot be verified directly, we track the consecutive-step drifts of their running estimates, $\|\hat{\boldsymbol{\mu}}_{t,l} - \hat{\boldsymbol{\mu}}_{t-1,l}\|$ and $\|\hat{\boldsymbol{\Sigma}}_{t,l} - \hat{\boldsymbol{\Sigma}}_{t-1,l}\|$, as empirical surrogates across multiple architectures and editing scales. Both quantities decay rapidly and stabilize in a low regime thereafter, consistent with the trackable-drift assumption above; full curves are in Appendix B.6.3. In Section 4, we further discuss how LN actively helps *maintain* such a trackable regime. Under Assumption 3.1 and 3.4, we establish the following error bounds:

**Theorem 3.5** (MSE Bound for Sequential Mean Estimation). *The MSE satisfies the following recursive bound:*

$$\text{MSE}_{t,l}^{(\boldsymbol{\mu},\text{cur})} \leq \underbrace{\frac{\kappa_{r,l}}{\kappa_{r+t,l}} \text{MSE}_{r,l}^{(\boldsymbol{\mu},\text{pre})}}_{\to 0 \text{ at rate } O(1/t)} + \underbrace{\frac{d\sigma_+}{\kappa_{r+t,l}} \ln\left(\frac{\kappa_{r+t,l}}{\kappa_{r,l}}\right)}_{\to 0 \text{ at rate } O(\ln t/t)}$$

$$+ \underbrace{\frac{2}{n\kappa_{r+t,l}} \sum_{j=1}^{t} \left(\kappa_{r+j,l} D_j^{(\boldsymbol{\mu},\text{cur})}\right)^2}_{\to 0 \text{ at rate } \max\{O(\ln t/t), O(t^{-\epsilon})\}}.$$

**Theorem 3.6** (Spectral Error Bound for Sequential Covariance Estimation). *Assuming $\nu_{r+t,l} > d + 1$, the spectral error $E_{t,l}^{(\boldsymbol{\Sigma},\text{cur})}$ satisfies the following recursive bound:*

$$E_{t,l}^{(\boldsymbol{\Sigma},\text{cur})} \leq \underbrace{\frac{\nu'_{r,l}}{\nu'_{r+t,l}} E_{r,l}^{(\boldsymbol{\Sigma},\text{pre})}}_{\to 0 \text{ at rate } O(1/t)} + \underbrace{\frac{(C_2(\sqrt{dn} + d) + d)\sigma_+ t}{\nu'_{r+t,l}}}_{\to \text{Const.}}$$

$$+ \frac{1}{\nu'_{r+t,l}} \sum_{j=1}^{t} \underbrace{\left(\nu'_{r+j-1,l} \Delta_j^{(\boldsymbol{\Sigma},\text{cur})} + 2n(D_j^{(\boldsymbol{\mu},\text{cur})})^2\right.}$$

$$\underbrace{\left. + 2n \text{MSE}_{j-1,l}^{(\boldsymbol{\mu},\text{cur})}\right),}_{\to 0 \text{ at rate } \max\{O((\ln t)^2/t), O(t^{-\epsilon'})\}}$$

*where $\nu'_{t,l} = \nu_{t,l} - d - 1$.*

Theorems 3.5 and 3.6 characterize how recursive tracking behaves under the trackable drift regime and why early edits help, thereby providing an estimation-level answer to the positive cumulative effect. Mathematically, the previous

phase contributes $rn$ "virtual samples", which effectively *shifts the estimation error curve of the current phase to the left* compared to a cold start (formalized in Remarks D.3 and D.5). Consequently, the current phase starts with an error level that a cold-start counterpart would only reach after roughly $r$ additional steps. This justifies an explicit *warm-up* phase: performing a small number of in-domain edits before the target sequence can precondition the running statistics and meet a target estimation error tolerance in fewer steps. See Appendices D.3 and D.4 for proofs.

### 3.3. From Accurate Tracking to Stable Updates

Section 3.2 shows that recursive tracking suppresses the estimation error. We now connect these estimates to the induced parameter updates and show how LN combined with ridge regression yields stable, weakly interfering updates.

Given the point estimates $(\hat{\boldsymbol{\mu}}_{t,l}, \hat{\boldsymbol{\Sigma}}_{t,l})$ from Corollary 3.3, LN centers and whitens the value gradient: $\tilde{\boldsymbol{v}}_{t,l}^i = \hat{\boldsymbol{\Sigma}}_{t,l}^{-1/2}(\boldsymbol{v}_{t,l}^i - \hat{\boldsymbol{\mu}}_{t,l})$, where $\hat{\boldsymbol{\Sigma}}_{t,l}$ is positive definite (hence invertible) by construction.[3] As in several LN-based editors (Tan et al., 2024; Li et al., 2025; Gu et al., 2025), the final parameter update $\boldsymbol{\Delta}_{t,l}$ is obtained by solving the following ridge regression problem:

$$\boldsymbol{\Delta}_{t,l} = \arg\min_{\boldsymbol{\Delta}} \|\boldsymbol{\Delta}\boldsymbol{H}_{t,l} - \boldsymbol{V}_{t,l}\|_F^2 + \lambda\|\boldsymbol{\Delta}\|_F^2, \quad (5)$$

where the matrix $\boldsymbol{H}_{t,l} = [\boldsymbol{h}_{t,l}^1, \ldots, \boldsymbol{h}_{t,l}^{n_t}]$ and the target matrix $\boldsymbol{V}_{t,l} = -\gamma[\|\tilde{\boldsymbol{h}}_{t,l}^1\|_2^2 \tilde{\boldsymbol{v}}_{t,l}^1, \ldots, \|\tilde{\boldsymbol{h}}_{t,l}^{n_t}\|_2^2 \tilde{\boldsymbol{v}}_{t,l}^{n_t}]$.[4] The closed-form solution to Equation (5) is given by

$$\boldsymbol{\Delta}_{t,l} = \boldsymbol{V}_{t,l}\boldsymbol{H}_{t,l}^\top(\boldsymbol{H}_{t,l}\boldsymbol{H}_{t,l}^\top + \lambda\boldsymbol{I})^{-1} = -\gamma\sum_{i=1}^{n_t}\tilde{\boldsymbol{v}}_{t,l}^i\boldsymbol{\phi}_{t,l}^i,$$
$$(6)$$

where $\boldsymbol{\phi}_{t,l}^i := \|\tilde{\boldsymbol{h}}_{t,l}^i\|_2^2(\boldsymbol{h}_{t,l}^i)^\top\left(\sum_j \boldsymbol{h}_{t,l}^j(\boldsymbol{h}_{t,l}^j)^\top + \lambda\boldsymbol{I}\right)^{-1}$.

We additionally assume bounded fourth moments for the projection factors $\boldsymbol{\phi}_{t,l}^i$ and the inverse-covariance estimate, as is standard in finite-sample analyses.

**Assumption 3.7.** The projection factor $\boldsymbol{\phi}_{t,l}^i$ and the inverse covariance estimator $\hat{\boldsymbol{\Sigma}}_{t,l}^{-1}$ possess finite fourth moments:

$$\mathbb{E}[\|\boldsymbol{\phi}_{t,l}^i\|_2^4] \leq C_{\boldsymbol{\phi}} < \infty, \quad \mathbb{E}[\|\hat{\boldsymbol{\Sigma}}_{t,l}^{-1}\|_2^4] \leq K_{\boldsymbol{\Sigma}} < \infty.$$

To expose how tracking accuracy shapes the update, decompose $\boldsymbol{v}_{t,l}^i = \boldsymbol{\mu}_{t,l} + \boldsymbol{\zeta}_{t,l}^i$, where $\boldsymbol{\mu}_{t,l}$ is the population mean,

---

[3]Initializing the NIW prior as $\boldsymbol{\Psi}_{0,l} = \epsilon_0\boldsymbol{I}$ with a small $\epsilon_0 > 0$ ensures $\hat{\boldsymbol{\Sigma}}_{t,l} \succ 0$ throughout tracking.

[4]$\tilde{\boldsymbol{h}}_{t,l}^i$ is a per-dimension standardized $\boldsymbol{h}_{t,l}^i$ used as an efficient reweighting when forming $\boldsymbol{V}_{t,l}$. See Appendix B.6.5 for more analyses.

capturing the *global shift* tendency over $\mathcal{D}$, while $\boldsymbol{\zeta}_{t,l}^i$ is the *instance-specific* direction required for the current edit with $\mathbb{E}[\boldsymbol{\zeta}_{t,l}^i] = \boldsymbol{0}$. Substituting into Equation (6) yields $\boldsymbol{\Delta}_{t,l} = \boldsymbol{\Delta}_{t,l}^{\text{spec}} + \boldsymbol{\Delta}_{t,l}^{\text{bias}}$, with $\boldsymbol{\Delta}_{t,l}^{\text{spec}} := -\gamma\sum_{i=1}^{n_t}\hat{\boldsymbol{\Sigma}}_{t,l}^{-1/2}\boldsymbol{\zeta}_{t,l}^i\boldsymbol{\phi}_{t,l}^i$ and $\boldsymbol{\Delta}_{t,l}^{\text{bias}} := -\gamma\hat{\boldsymbol{\Sigma}}_{t,l}^{-1/2}(\boldsymbol{\mu}_{t,l} - \boldsymbol{m}_{t,l})\sum_{i=1}^{n_t}\boldsymbol{\phi}_{t,l}^i$. The first term is driven by $\tilde{\boldsymbol{\zeta}}_{t,l}^i = \hat{\boldsymbol{\Sigma}}_{t,l}^{-1/2}\boldsymbol{\zeta}_{t,l}^i$ and the second is induced by the mean-estimation error. Based on this, we obtain the following results:

**Theorem 3.8** (Asymptotic Properties of Parameter Updates). *Under Assumption 3.1, 3.4 and 3.7, the parameter update $\boldsymbol{\Delta}_{t,l}$ satisfies the following properties:*

(a) *Bias Decay: There exists a constant $C_{bias} > 0$, independent of $t$, such that $\mathbb{E}[\|\boldsymbol{\Delta}_{t,l}^{bias}\|_F^2] \leq C_{bias} \cdot \text{MSE}_{t,l}^{(\boldsymbol{\mu})}$.*

(b) *Bounded Norm: There exists a constant $U_{spec} < \infty$ such that $\mathbb{E}[\|\boldsymbol{\Delta}_{t,l}^{spec}\|_F^2] \leq U_{spec}$.*

(c) *Interference Mitigation: For distinct steps $t \neq k$, let $\mathcal{G}_{t,k,l} := \sum_{i=1}^{n_t}\sum_{j=1}^{n_k}\langle\boldsymbol{\phi}_{t,l}^i, \boldsymbol{\phi}_{k,l}^j\rangle_F$. Then the expected interference $\mathbb{E}[\langle\boldsymbol{\Delta}_{t,l}, \boldsymbol{\Delta}_{k,l}\rangle_F] = \gamma^2\mathbb{E}[\tilde{\boldsymbol{s}}_{t,l}^\top\tilde{\boldsymbol{s}}_{k,l}\cdot\mathcal{G}_{t,k,l}] + o(1)$, where $\tilde{\boldsymbol{s}}_{t,l} := \mathbb{E}_{\mathcal{V}}[\tilde{\boldsymbol{\zeta}}_{t,l}^i|\mathcal{H}]$, $\mathcal{H}$ denotes the history of features $\{\tilde{\boldsymbol{h}}_{\tau,l}^i\}$ up to the current step $t$, $\mathcal{V}$ the corresponding gradients $\{\tilde{\boldsymbol{v}}_{\tau,l}^i\}$, and $o(1)$ vanishes as $t, k \to \infty$.*

**Corollary 3.9** (Bounded Update Magnitude). *The update $\boldsymbol{\Delta}_{t,l}$ satisfies $\mathbb{E}[\|\boldsymbol{\Delta}_{t,l}\|_F^2] \leq 2C_{bias} \cdot \text{MSE}_{t,l}^{(\boldsymbol{\mu})} + 2U_{spec}$.*

Theorem 3.8 explains how estimation accuracy translates into update stability. Property (a) ensures that $\boldsymbol{\Delta}_{t,l}^{\text{bias}}$ shrinks with $\text{MSE}_{t,l}^{(\boldsymbol{\mu})}$, progressively removing the shared global-shift tendency from the update, so the update is increasingly driven by instance-specific directions. Together with property (b), Corollary 3.9 yields robust and bounded updates, helping prevent explosive deviations in long-horizon editing and thereby alleviating model collapse. Finally, property (c) shows that cross-step interference is governed by the correlation between characteristic directions $\tilde{\boldsymbol{s}}_{t,l}^\top\tilde{\boldsymbol{s}}_{k,l}$; for unrelated edits (weakly correlated directions), updates become asymptotically orthogonal in expectation, mitigating catastrophic forgetting. See Appendix D.5 for formal proofs and detailed analyses.

We further show that these conclusions in Theorem 3.8 continue to hold if normalized gradients $\tilde{\boldsymbol{v}}_{t,l}^i$ are first passed through a shallow hypernetwork prior to the update computation, as long as the transformation is Lipschitz continuous (e.g., shallow MLP modules with ReLU activations and residual blocks, as used in RLEDIT (Li et al., 2025)). In essence, the update remains governed by LN and ridge-regularized regression system. See Corollary D.7 for proofs.

*Table 1.* Comparison of STABLEEDIT and baseline methods on standard-scale lifelong editing tasks (20K&17K edits, $n_t$=100 samples per step). Eff., Gen., and Spe. denote Efficacy, Generalization, and Specificity, respectively. The symbol ↑ indicates that higher values are better. The best results are highlighted in bold, while the second-best results are underlined.

| Method | ZsRE | | | FEVER | | | WikiBigEdit | | | ULTRAEDITBENCH | | |
|---|---|---|---|---|---|---|---|---|---|---|---|---|
| | Eff.↑ | Gen.↑ | Spe.↑ | Eff.↑ | Gen.↑ | Spe.↑ | Eff.↑ | Gen.↑ | Spe.↑ | Eff.↑ | Gen.↑ | Spe.↑ |
| **Mistral-7B-v0.3** | | | | | | | | | | | | |
| Pre-edited | 44.46 | 43.55 | 48.08 | 0.41 | 0.50 | 1.98 | 39.14 | 38.21 | 41.62 | 30.83 | 29.94 | 31.11 |
| FT | 13.69 | 12.43 | 19.87 | 23.80 | 23.37 | 16.30 | 13.77 | 14.86 | 11.84 | 0.06 | 0.36 | 0.07 |
| MEMIT | 0.00 | 0.00 | 0.00 | 0.00 | 0.00 | 0.00 | 0.00 | 0.00 | 0.00 | 0.00 | 0.00 | 0.00 |
| ALPHAEDIT | 0.00 | 0.00 | 0.00 | 0.00 | 0.00 | 0.00 | 0.00 | 0.00 | 0.00 | 0.00 | 0.00 | 0.00 |
| MEND | 3.33 | 3.08 | 0.20 | 0.00 | 0.00 | 0.00 | 0.01 | 0.00 | 0.00 | 0.00 | 0.00 | 0.00 |
| MALMEN | 1.15 | 1.06 | 0.02 | 18.42 | 17.43 | 12.09 | 0.00 | 0.01 | 0.00 | 2.42 | 2.48 | 3.47 |
| RLEDIT | 71.62 | 67.60 | 30.70 | 92.35 | 91.39 | 71.85 | 57.55 | 52.47 | 28.78 | 69.85 | 65.10 | 57.08 |
| ULTRAEDIT | 85.30 | 80.80 | 47.38 | 97.87 | 96.09 | **84.29** | 76.00 | 70.15 | 46.09 | 83.71 | 77.30 | 67.26 |
| **STABLEEDIT** | **87.39** | **82.93** | **50.26** | **98.38** | **96.55** | 83.89 | **78.90** | **72.57** | **47.36** | **84.98** | **78.77** | **68.36** |
| **Llama-3-8B-Instruct** | | | | | | | | | | | | |
| Pre-edited | 36.76 | 35.83 | 38.93 | 0.02 | 0.02 | 0.26 | 24.92 | 35.46 | 38.87 | 27.29 | 26.40 | 27.24 |
| FT | 18.51 | 18.11 | 5.01 | 16.21 | 13.08 | 5.01 | 13.00 | 11.70 | 6.75 | 13.50 | 11.92 | 10.18 |
| MEMIT | 0.14 | 0.14 | 1.40 | 0.02 | 0.02 | 0.02 | 0.02 | 0.02 | 0.09 | 0.74 | 0.45 | 0.21 |
| ALPHAEDIT | 86.10 | 81.15 | 26.26 | 6.39 | 6.14 | 2.72 | 63.24 | 54.68 | 20.17 | 4.51 | 3.16 | 2.78 |
| MEND | 0.00 | 0.00 | 0.00 | 36.19 | 36.19 | 24.31 | 0.26 | 0.27 | 0.18 | 0.00 | 0.00 | 0.00 |
| MALMEN | 3.86 | 3.07 | 0.26 | 95.03 | 92.44 | 69.20 | 0.00 | 0.00 | 0.00 | 40.64 | 34.18 | 37.29 |
| RLEDIT | **91.09** | **89.62** | 41.43 | 91.38 | 89.93 | 41.98 | 75.35 | 70.00 | 37.21 | 83.37 | 78.65 | 63.31 |
| ULTRAEDIT | 90.07 | 87.36 | **49.51** | 95.39 | 91.93 | 67.14 | 79.60 | 73.49 | **48.51** | 85.70 | 81.28 | 68.73 |
| **STABLEEDIT** | 90.61 | 88.38 | 48.23 | **97.89** | **94.47** | **69.24** | **81.22** | **75.98** | 45.22 | **88.88** | **85.46** | **69.06** |
| **GPT-J-6B** | | | | | | | | | | | | |
| Pre-edited | 27.22 | 26.42 | 27.33 | 9.61 | 9.68 | 15.90 | 29.97 | 29.08 | 32.58 | 22.01 | 21.47 | 22.20 |
| FT | 15.11 | 13.55 | 2.61 | 14.02 | 13.98 | 9.26 | 21.90 | 17.56 | 8.47 | 22.03 | 17.61 | 19.60 |
| MEMIT | 0.00 | 0.00 | 0.00 | 5.54 | 5.03 | 5.46 | 1.59 | 1.59 | 0.50 | 0.25 | 0.18 | 0.20 |
| ALPHAEDIT | 50.23 | 43.31 | 12.54 | 1.89 | 1.87 | 1.85 | 69.66 | 55.03 | 21.20 | 22.19 | 12.09 | 6.88 |
| MEND | 2.52 | 2.55 | 0.19 | 52.80 | 51.44 | 45.42 | 0.02 | 0.01 | 0.02 | 1.71 | 1.71 | 1.83 |
| MALMEN | 0.02 | 0.01 | 0.02 | 1.33 | 1.25 | 2.92 | 0.00 | 0.01 | 0.01 | 0.76 | 0.48 | 0.78 |
| RLEDIT | 72.65 | 68.45 | 21.79 | 97.29 | 96.01 | **79.86** | 69.18 | 63.01 | 32.75 | 81.14 | 74.88 | 61.87 |
| ULTRAEDIT | 78.03 | 72.42 | 27.05 | 97.45 | 96.37 | 79.72 | 73.84 | 66.57 | 37.17 | **84.03** | **76.62** | 64.03 |
| **STABLEEDIT** | **82.12** | **78.39** | **31.52** | **98.35** | **96.99** | 79.85 | **75.10** | **68.13** | **44.04** | 83.41 | 76.55 | **67.17** |

## 4. Closing the Loop: STABLEEDIT

We now connect the pieces in Sections 3.1 to 3.3 into a self-reinforcing stability loop that clarifies the fundamental mechanism of LN (Q1). Specifically, recursive Bayesian tracking provides an *efficient* way to estimate the drifting value-gradient distribution from limited samples (Section 3.1). As edits accumulate, the estimation error decreases (Section 3.2), making LN's centering and scaling increasingly accurate. With standardized gradients, LN together with ridge-regularized regression yields updates that are bounded and asymptotically orthogonal (Section 3.3), thereby mitigating catastrophic forgetting and model collapse. By Lemma D.1, bounded updates further constrain the drift of the distribution moments in the next step, helping maintain the slowly evolving regime required for reliable tracking. This closes the loop and explains why early edits can facilitate later ones.

Guided by this mechanism, we propose STABLEEDIT as a theory-driven refinement of ULTRAEDIT (Gu et al., 2025). It explicitly strengthens two edges of the loop. First, it adds a *warm-up* stage using a few edits to initialize reliable running estimates before entering the target edit sequence, thus skipping the high-estimation-error phase. Second, it uses *full whitening*, which acts as an adaptive preconditioner that suppresses noisy high-variance directions while amplifying weak directions, thereby improving numerical stability of updates over long sequences (see Remark D.9). This refinement incurs only modest overhead while yielding consistently improved long-horizon stability across million-scale edit streams. The pseudo-code is provided in Algorithm 1.

## 5. Experiments

In this section, we conduct extensive experiments to evaluate STABLEEDIT and validate our theoretical findings.

*Table 2.* Results on WikiBigEdit at 500K edits ($n_t$=100 samples per step) across three models.

| Method | Mistral-7B-v0.3 | | | | Llama-3-8B-Instruct | | | | GPT-J-6B | | | |
|---|---|---|---|---|---|---|---|---|---|---|---|---|
| | Eff.↑ | Gen.↑ | Spe.↑ | Personas ↑ | Eff.↑ | Gen.↑ | Spe.↑ | Personas ↑ | Eff.↑ | Gen.↑ | Spe.↑ | Personas ↑ |
| Pre-edited | 39.14 | 38.21 | 41.62 | 35.54 | 24.92 | 35.46 | 38.87 | 32.70 | 29.97 | 29.08 | 32.58 | 26.46 |
| ULTRAEDIT | 71.78 | 65.63 | 55.40 | 56.11 | 68.99 | 63.59 | 52.28 | 55.04 | 66.46 | 60.54 | 47.90 | 51.73 |
| **STABLEEDIT** | **74.54** | **67.80** | **56.49** | **58.88** | **72.44** | **66.87** | **53.83** | **60.39** | **71.47** | **65.04** | **52.84** | **57.20** |

## 5.1. Experimental Setup

We briefly outline the evaluation metrics, datasets, and baseline methods; for a comprehensive description of the experimental settings, please refer to Appendix A.

**Base LLMs & Baseline Methods.** Our experiments are conducted on four representative LLMs: GPT-J-6B, Mistral-7B-v0.3, Llama-3-8B-Instruct, and Qwen2.5-7B-Instruct. We compare STABLEEDIT against several model editing baselines, including FT (Zhu et al., 2020), MEMIT (Meng et al., 2023), ALPHAEDIT (Fang et al., 2025), MEND (Mitchell et al., 2022a), MALMEN (Tan et al., 2024), RLEDIT (Li et al., 2025), and ULTRAEDIT (Gu et al., 2025).

**Datasets & Evaluation Metrics.** We evaluate STABLEEDIT on four datasets: ZsRE (Levy et al., 2017), FEVER (Thorne et al., 2018), ULTRAEDITBENCH (Gu et al., 2025) and WikiBigEdit (Thede et al., 2025). In line with prior works, for the first three datasets, we report *Efficacy*, *Generalization*, and *Specificity*. For WikiBigEdit, we additionally assess *Personas* and *Multi-hop Reasoning capabilities*.

## 5.2. Knowledge Update and Scalability

To evaluate STABLEEDIT's effectiveness and efficiency, we report results across three settings: editing accuracy at *standard scale* (20K edits for ZsRE, FEVER, and ULTRAEDITBENCH; 17K for WikiBigEdit), editing accuracy at *ultra-large scale* (500K for WikiBigEdit; 2M for ULTRAEDITBENCH), and per-edit runtime. All runs use 100 samples per step. Extended results, including Qwen2.5-7B-Instruct, long-horizon trajectories, and covariance diagnostics, are in Appendices B.1 and B.3.

**Obs 1 (Standard-scale sequences): STABLEEDIT delivers consistently strong performance across models and datasets and demonstrates out-of-domain generalizability without dataset-specific tuning.** From Table 1, STABLEEDIT is generally top-tier on the core metrics reported for the four datasets. For example, on ULTRAEDITBENCH with Llama-3-8B-Instruct, STABLEEDIT achieves 85.46% Generalization compared to 81.28% for ULTRAEDIT. Notably, several baselines can become ineffective under these sequential editing workloads (e.g., near-zero outcomes on Mistral-7B-v0.3 for multiple prior editors), underscoring the difficulty of long-horizon editing and highlighting the robustness gained by explicitly stabilizing the LN-driven

update pipeline. We additionally evaluate on the medical benchmark MedCF (Xu et al., 2024) with no domain-specific LLM and no dataset-specific hyperparameter tuning; STABLEEDIT achieves higher scores than ULTRAEDIT on Mistral-7B-v0.3 and Llama-3-8B-Instruct across all three metrics (full table in Appendix B.2), indicating that STABLEEDIT's performance gains extend to out-of-domain settings without any dataset-specific adaptation.

**Obs 2 (Ultra-large sequences): STABLEEDIT remains robust at ultra-large scale and mitigates catastrophic forgetting.** From Table 2, on WikiBigEdit with 500K edits, STABLEEDIT consistently outperforms ULTRAEDIT across three LLMs on all reported metrics, indicating better scalability as edits accumulate (e.g., on GPT-J-6B, STABLEEDIT improves Generalization by 7.43% relative to ULTRAEDIT). This sustained advantage supports the claim that the update dynamics remain better conditioned and less mutually interfering at scale, which alleviates the catastrophic-forgetting trend when the number of edits becomes very large. Trajectory-level evidence (20K–500K on WikiBigEdit, 50K–2M on ULTRAEDITBENCH) and covariance-conditioning diagnostics that further substantiate this claim are deferred to Appendix B.3.

*Table 3.* Running time (seconds per edit) comparison of different editing methods on Llama-3-8B-Instruct.

| Method | ALPHAEDIT | MEND | MALMEN | RLEDIT | ULTRAEDIT | **Ours** |
|---|---|---|---|---|---|---|
| **Time** | 6.02 | 1.12 | 1.44 | 0.22 | 0.07 | 0.10 |

**Obs 3 (Efficiency): STABLEEDIT improves long-horizon stability with the smallest overhead over ULTRAEDIT.** On Llama-3-8B-Instruct (Table 3), STABLEEDIT is only $1.43\times$ as slow as ULTRAEDIT, while other baselines incur substantially larger slowdowns (e.g., RLEDIT $3.14\times$, MEND $16.0\times$, MALMEN $20.6\times$, ALPHAEDIT $86.0\times$). For a detailed discussion of the sources of this overhead, see Appendix B.4.

## 5.3. Evaluation of General Capability

To assess whether sequential editing preserves the base LLM's broader competence, we perform General Capability Tests using five tasks from the General Language Understanding Evaluation (GLUE) benchmark (Wang et al., 2019): SST (Socher et al., 2013), MRPC (Dolan & Brockett,

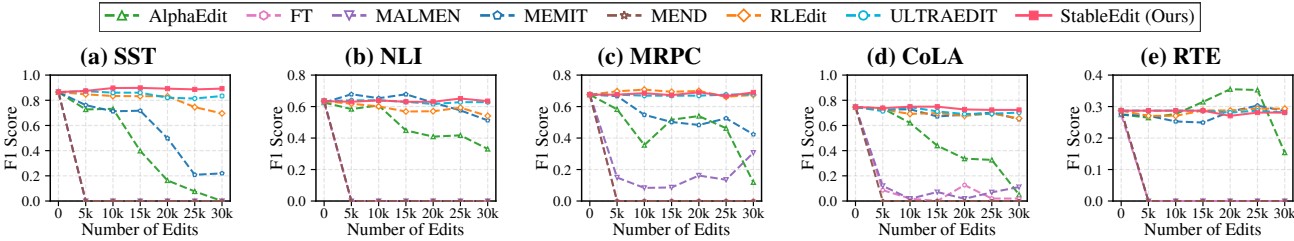

*Figure 2.* F1 scores on five GLUE tasks for general capability evaluation as the number of sequential edits increases (up to 30K) on Llama-3-8B-Instruct. STABLEEDIT, ULTRAEDIT, and RLEDIT (all LN-driven) largely sustain stable performance on the edit stream, whereas other baselines degrade rapidly as edits accumulate.

2005), RTE (Bentivogli et al., 2009), CoLA (Warstadt et al., 2019), and NLI (Williams et al., 2018). We also visualize the *final-layer* hidden-state distributions of the pre-edit and post-edit models using UMAP, computed on the same 1,000 randomly sampled factual prompts that are independent of the edit stream. Results are shown in Figures 2 and 3; detailed benchmark descriptions and extended per-backbone comparisons are in Appendices A.2 and B.5.

**Obs 4: STABLEEDIT sustains general capabilities over long edit sequences, while several baselines collapse.** From Figure 2, STABLEEDIT stays close to the pre-edit performance across all reported tasks throughout the trajectory, indicating that integrating new facts does not come at the cost of broad linguistic competence. Notably, LN-based editors such as RLEDIT and ULTRAEDIT also largely preserve these capabilities, whereas other baselines exhibit sharp degradation as edits accumulate, consistent with model-collapse behavior. This pattern aligns with our analysis: bounded, weakly interfering updates help prevent systemic drift.

**Obs 5: STABLEEDIT preserves the representation manifold and avoids explosive distributional shifts.** From Figure 3, representations produced by STABLEEDIT remain well-aligned with the pre-edit manifold, suggesting that edits are integrated via controlled, localized adjustments. In contrast, baselines tend to induce visible drift, which can propagate through the LLMs and manifest as broad performance degradation on unrelated tasks.

### 5.4. Verification of the LN Mechanism

To empirically probe the LN mechanism of Section 3 and the design choices of STABLEEDIT, we address three questions: **(i)** how does each component — LN, warm-up, and whitening — contribute to the observed stability? **(ii)** how does warm-up help stabilize the edit stream, and is this benefit sensitive to whether the warm-up data comes from the same distribution as the target? **(iii)** do the bounded-norm and near-orthogonality properties derived in Theorem 3.8 actually emerge along the edit stream, and are they absent in non-LN baselines? Additional diagnostics and extended

*(a)* STABLEEDIT on Llama-3-8B-Instruct  *(b)* ALPHAEDIT on Llama-3-8B-Instruct

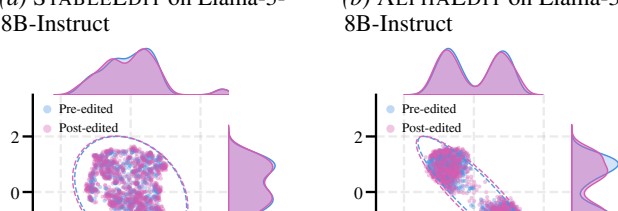

*Figure 3.* The distribution of hidden representations of pre-edited and post-edited LLMs after dimensionality reduction. The dashed lines represent the 0.95 confidence intervals. Best viewed in color.

tables are in Appendix B.6.

**Obs 6 (Component ablation): LN is the core stability mechanism; warm-up and full whitening provide further consistent gains.** From Table 4, removing LN causes a dramatic collapse across all five metrics, whereas removing warm-up or full whitening yields smaller, consistent drops; in this setting, whitening contributes a larger incremental gain than warm-up. This hierarchy aligns with our analysis: LN activates the self-reinforcing stability loop, warm-up improves early calibration of the running statistics, and full whitening produces more geometrically balanced parameter updates by normalizing gradient directions.

**Obs 7 (Warm-up dynamics): Warm-up stabilizes the edit stream by establishing reliable initial running statistics, and this benefit persists even under distribution mismatch.** Full results are in Appendix B.6.1; Table 5 provides a compact Mistral-7B-v0.3 summary. Replacing the warm-up data with the out-of-domain medical set MedCF causes an average absolute change of only 0.43 points, and the mismatched variant surpasses ULTRAEDIT on 15 out of 18 entries. These results show that replacing the warm-up distribution does not cause severe degradation, suggesting that the benefit of warm-up is not tied to strict same-distribution matching: even under substantial mismatch, the statistics accumulated during warm-up provide a reliable starting point for the running estimates and are subsequently refined on-

*Table 4.* Component ablation of STABLEEDIT on Mistral-7B-v0.3 with WikiBigEdit (20K edits).

| Variant | Eff. | Gen. | Spe. | Personas | Reasoning |
|---|---|---|---|---|---|
| STABLEEDIT (full) | 78.90 | 72.57 | 47.36 | 63.96 | 37.22 |
| w/o Warm-up | 76.96 | 71.13 | 46.84 | 62.80 | 36.08 |
| w/o Whitening | 75.12 | 69.63 | 45.50 | 62.70 | 35.30 |
| w/o LN | 39.13 | 38.18 | 41.61 | 35.55 | 29.94 |

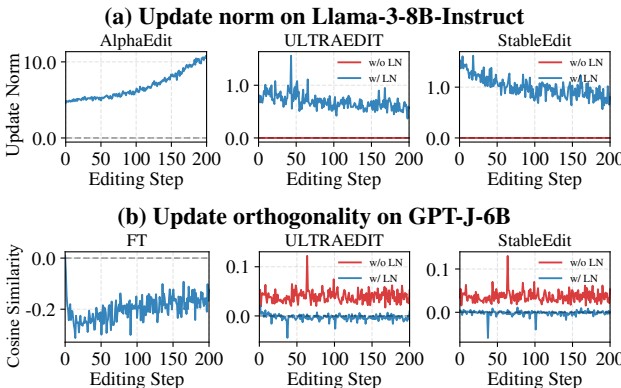

*Figure 4.* Cross-method update-geometry diagnostics (20K edits on ZsRE). **(a)** Per-step parameter update norm on Llama-3-8B-Instruct. **(b)** Cosine similarity between consecutive updates on GPT-J-6B.

line as target-stream edits arrive. Further diagnostics on warm-up size and placement, along with a statistics-only ablation that adds no parameter updates during warm-up, are in Appendix B.6.1.

**Obs 8 (Update geometry): LN yields near-orthogonal and well-scaled updates, mitigating interference and model collapse.** Figure 4 verifies the predictions of Theorem 3.8. On GPT-J-6B, STABLEEDIT and ULTRAEDIT keep adjacent-update cosine tightly around 0, and STABLEEDIT arrives earliest. Removing LN from either method drives cosine far from 0, as does the non-LN FT. On Llama-3-8B-Instruct, both LN methods keep update norms bounded, and STABLEEDIT further trends downward. Removing LN collapses the norms and triggers under-fitting, while the non-LN ALPHAEDIT grows steadily. These patterns confirm that LN with ridge regression yields weakly interfering, bounded updates; see Appendix B.6.2 for more results.

## 6. Related Work

**Parameter-Modifying Model Editing.** This line of work directly writes new facts into model weights. Early fine-tuning style approaches (Zhu et al., 2020; Hu et al., 2022) often lead to model collapse (Liu et al., 2025) in sequential tasks. More recent locate-then-edit methods (Dai et al., 2022; Meng et al., 2022; 2023; Fang et al., 2025; Pan et al., 2025) identify hidden states at subject token positions responsible for knowledge storage and apply closed-form

*Table 5.* Warm-up distribution robustness (Mistral-7B-v0.3, 20K edits). STABLEEDIT uses in-domain warm-up as the reference; Warmup-MedCF replaces warm-up data with the out-of-domain medical set MedCF while keeping the target benchmarks (ZsRE and FEVER) unchanged.

| Method | ZsRE | | | FEVER | | |
|---|---|---|---|---|---|---|
| | Eff. | Gen. | Spe. | Eff. | Gen. | Spe. |
| ULTRAEDIT | 85.30 | 80.80 | 47.38 | 97.87 | 96.09 | 84.29 |
| STABLEEDIT | 87.39 | 82.93 | 50.26 | **98.38** | **96.55** | **83.89** |
| Warmup-MedCF | **87.63** | **83.09** | **51.28** | 98.36 | 96.45 | 82.90 |

updates. However, these remain unstable in lifelong settings due to localization difficulties (Hase et al., 2023; Guo et al., 2025a). Hypernetwork-based methods (Cao et al., 2021; Mitchell et al., 2022a; Tan et al., 2024) predict updates with auxiliary networks (often using LN). While efficient, they typically struggle with model parameters that dynamically shift during the editing process, a challenge recently mitigated by RLEDIT (Li et al., 2025). Most relevant to our work, ULTRAEDIT (Gu et al., 2025) attains strong long-horizon stability largely via LN-style normalization, yet why LN works so well remains poorly understood—the focus of our mechanism analysis.

**Parameter-Preserving Model Editing.** These methods augment the model with external resources without altering its core weights, thereby minimizing unintended side effects on the base model. Strategies include utilizing external memory banks (Mitchell et al., 2022b; Hartvigsen et al., 2023; Zhong et al., 2023; Yu et al., 2024; Rizwan et al., 2025; Wang et al., 2025), injecting additional parameters through dedicated "patcher" modules (Dong et al., 2022; Huang et al., 2023), or leveraging In-Context Learning (Zheng et al., 2023; Cohen et al., 2024) to internalize new facts. While these methods avoid direct parameter corruption, they scale poorly in lifelong scenarios: growing knowledge implies an ever-larger memory footprint and roughly linear growth in inference latency from retrieval or longer context. This renders them less sustainable for long-term knowledge maintenance compared to parameter-modifying methods, which keep the model size fixed and typically offer more stable inference costs.

## 7. Conclusion

In this paper, we theoretically demystified the LN strategy in LME. We proved that LN establishes a self-reinforcing stability loop by enforcing asymptotic orthogonality and bounded update magnitudes. These properties alleviate the dual challenges of catastrophic forgetting and model collapse. Leveraging these insights, our proposed editor, STABLEEDIT, achieves competitive performance across extensive experiments on million-scale editing sequences, providing empirical support for the proposed mechanism.

## Acknowledgements

This work was supported in part by the National Science and Technology Major Project (No. 2023ZD0121104), in part by the Anhui Natural Science Foundation (No. 2508085ZD006), in part by the Postdoctoral Fellowship Program and the China Postdoctoral Science Foundation under Grant Nos. BX20250387 and 2025M781529, and in part by the Fundamental Research Funds for the Central Universities under Grant No. WK2150250042.

## Impact Statement

This paper advances lifelong model editing by providing a theoretical account of Lifelong Normalization and proposing STABLEEDIT to improve stability over long edit sequences, which may help practitioners more reliably update deployed language models (e.g., correcting outdated information) without degrading general capabilities. As with other parameter-modifying editors, these techniques could also be misused to inject false or harmful content or to undermine safety constraints; we therefore encourage responsible use with appropriate safeguards such as access control, auditing, and careful validation of edits. Overall, we do not identify ethical concerns beyond those already associated with model editing.

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

# A. Experimental Setup

In this section, we provide a detailed description of the experimental configuration, including a comprehensive explanation of the evaluation metrics, datasets, and baselines.

## A.1. Evaluation Metrics

**Efficacy (Eff.)** quantifies how successfully a language model integrates new or updated factual information. Specifically, it measures whether the model's top-1 prediction for a given edited input $x_e$ precisely matches the desired target label $y_e$. A high efficacy score indicates that the editing process has effectively updated the model's internal knowledge to produce the correct response for the specific fact that was edited. The formula for efficacy is expressed as

$$\mathbb{E}\left[\mathbb{I}\left(y_e = \arg\max_{y'} \mathbb{P}_{f_W}(y'|x_e)\right)\right].$$

**Generalization (Gen.)** assesses the robustness of the knowledge edit by evaluating whether the model can apply the newly acquired information to semantically equivalent but linguistically varied inputs. This metric checks if the model's top-1 prediction for a paraphrased version of the edited input, denoted as $x'_e$, remains consistent with the target label $y_e$. A strong generalization score indicates that the model has truly absorbed the new knowledge in a conceptual way, rather than merely memorizing the exact phrasing of the original editing instance. The formula for generalization is defined as

$$\mathbb{E}\left[\mathbb{I}\left(y_e = \arg\max_{y'} \mathbb{P}_{f_W}(y'|x'_e)\right)\right].$$

**Specificity (Spe.)** ensures that model edits do not inadvertently corrupt or degrade the model's pre-existing knowledge on unrelated topics. It measures the model's ability to retain its original behavior and predictions for inputs that are not associated with the edited knowledge. For each unrelated input $x_u$, specificity verifies that the model's top-1 prediction continues to match its original, unedited label $y_u$. A high specificity score is essential for safe and stable lifelong learning, preventing catastrophic forgetting or unintended side effects. The formula for specificity is given by

$$\mathbb{E}\left[\mathbb{I}\left(y_u = \arg\max_{y'} \mathbb{P}_{f_W}(y'|x_u)\right)\right].$$

**Personas Accuracy (Personas)** is an additional evaluation metric introduced in the WikiBigEdit benchmark to measure a stronger form of generalization beyond standard rephrasing. Specifically, given a persona-conditioned query $x_p$ and its corresponding target answer $y_p$, this metric evaluates whether the edited model can still produce the correct answer when the original fact is reformulated in the style of a specific persona. In WikiBigEdit, such examples are constructed by rewriting the original edit question under different persona styles (e.g., detective, pirate, or philosopher), while preserving the same underlying factual content. A high Personas score therefore indicates that the edit transfers not only across surface-level paraphrases but also across stylistically transformed prompts. Formally, Personas Accuracy is defined as

$$\text{Personas} = \mathbb{E}\left[\mathbb{I}\left(y_p = \arg\max_{y'} \mathbb{P}_{f_W}(y' \mid x_p)\right)\right].$$

**Multi-hop Reasoning Accuracy (Reasoning)** is another evaluation metric introduced in WikiBigEdit to assess whether an edited model can answer compositional queries that require combining multiple incorporated facts. Given a multi-hop query $x_m$ and its target answer $y_m$, this metric measures whether the model's top-1 prediction matches the correct answer. In WikiBigEdit, these evaluation examples are constructed from two related factual tuples, where the object of one fact serves as the subject of another, and the intermediate entity is omitted from the final question. A high Reasoning score thus indicates that the model can connect edited facts and perform two-step reasoning, rather than merely recalling isolated associations. Formally, Multi-hop Reasoning Accuracy is defined as

$$\text{Reasoning} = \mathbb{E}\left[\mathbb{I}\left(y_m = \arg\max_{y'} \mathbb{P}_{f_W}(y' \mid x_m)\right)\right].$$

**Discussion on Evaluation Reliability.** We adopt these standard metrics for direct comparison with prior model editing work. Recent studies (Yang et al., 2025; Gu et al., 2026) suggest that token-matching metrics may not always fully capture genuine editing effects, which we view as a complementary direction for future work.

## A.2. Datasets

**ZsRE** (Levy et al., 2017) is a widely used question-answering benchmark for evaluating factual knowledge editing in language models. Each instance centers on a question–answer pair representing a factual relation to be updated, where the answer serves as the editing target. To assess whether an edit generalizes beyond the original prompt, the dataset also provides paraphrased variants of the question generated through back-translation. In addition, it includes unrelated questions used to measure specificity, i.e., whether the edit preserves knowledge outside the target scope. This structure makes ZsRE suitable for evaluating editing quality from three complementary perspectives: edit success, generalization to equivalent queries, and specificity to unrelated facts.

**FEVER** (Thorne et al., 2018) is a fact verification dataset built from Wikipedia-derived claims annotated by their factual status. In its original form, each example is labeled as Supported, Refuted, or Not Enough Info, with evidence provided for verifiable claims. In model editing settings, it is typically converted into a binary classification problem, where each input claim is paired with a target label sampled from a Bernoulli distribution with $p = 0.5$, indicating whether it should be judged as true or false. Compared with question-answering benchmarks such as ZsRE, FEVER focuses on editing the model's judgment over factual statements rather than updating a direct answer to a question. This makes it a complementary benchmark for studying whether editing methods can handle verification-style knowledge updates.

**WikiBigEdit** (Thede et al., 2025) is a large-scale benchmark for lifelong knowledge editing built from real-world factual changes in Wikidata. Its first release contains more than 500K question-answer pairs organized into eight sequential timesteps, allowing evaluation under a realistic stream of evolving knowledge. Unlike earlier editing benchmarks that are relatively small or synthetic, WikiBigEdit is designed to reflect practical update scenarios at scale through an automated pipeline that can be continuously extended with newly observed edits. Notably, 490,519 examples are used to evaluate update efficacy, rephrasing generalization, specificity, and persona-based accuracy, while a separate set of 17,541 examples is specifically annotated for multi-hop reasoning. This design enables a more targeted assessment of the model's ability to handle compositional queries and long-range dependencies, making WikiBigEdit particularly suitable for studying whether editing methods remain effective and stable over long editing horizons.

**ULTRAEDITBENCH** (Gu et al., 2025) is introduced as a novel and expansive factual question-answering benchmark, specifically constructed from Wikidata triples to support large-scale model editing. This benchmark is composed of over 2 million comprehensive editing pairs, making it the largest dataset in its domain to date. The design principles of ULTRAEDITBENCH align with established evaluation frameworks in model editing, such as those seen in ZsRE, ensuring its relevance and utility for assessing model updates. The diversity of ULTRAEDITBENCH is a key characteristic, encompassing a broad range of subject types including persons, organizations, and geographical entities, alongside a multitude of languages. This extensive coverage and continuous expandability, owing to its Wikidata foundation, position ULTRAEDITBENCH as a robust resource for pushing the boundaries of ultra-large-scale lifelong model editing.

**The GLUE Benchmark** (Wang et al., 2019) is a widely recognized collection of diverse natural language understanding (NLU) tasks, specifically curated to assess the performance of models across a broad spectrum of linguistic phenomena. It serves as a comprehensive evaluation suite, pushing models to demonstrate a generalized understanding of language rather than excelling at a single, narrow task. We selected five tasks from this benchmark to evaluate how well different methods maintain general language capabilities.

- **SST** (Socher et al., 2013) is a sentiment classification task built from movie review sentences annotated with polarity labels. Each example contains a single sentence, and the model is required to predict whether its sentiment is positive or negative.

- **NLI** (Williams et al., 2018) evaluates whether a model can infer the semantic relation between two sentences. Given a premise and a hypothesis, the task is to determine their logical relationship, such as entailment, contradiction, or neutrality.

- **MRPC** (Dolan & Brockett, 2005) is a paraphrase identification benchmark based on sentence pairs. The goal is to decide whether the two sentences express the same underlying meaning.

- **CoLA** (Warstadt et al., 2019) focuses on linguistic acceptability judgment. It requires the model to classify whether a single sentence is grammatically acceptable according to standard English usage.

- **RTE** (Bentivogli et al., 2009) is a textual entailment task that tests whether the meaning of one sentence can be inferred from another.

**MedCF** (Xu et al., 2024) is a medical counter-fact dataset proposed for editing factual clinical and biomedical knowledge in LLMs. It is constructed from a medical knowledge graph called DRKG. Each edit instance starts from a real medical triplet that links a head entity (e.g., a drug) to a tail entity (e.g., a side effect) via some medical relation; the tail entity is then replaced by a counterfactual one so that the original fact is turned into a false one. ChatGPT is subsequently prompted to phrase the corresponding query as a natural-language medical question and to produce a rephrased variant of it, which together serve as the edit prompt and its paraphrase. The benchmark spans 17 medical relation types—including drug-drug interactions, compound-disease relations (treatment, inhibition, side effect, prevention), anatomy-disease localization, and disease-symptom associations—and contains 2,407 training, 817 validation, and 801 test instances.

## A.3. Baselines

**FT** (Zhu et al., 2020) represents a straightforward and extensively applied technique for model modification. It entails fine-tuning specific layers of the LLM through autoregressive loss on a fresh set of editing examples to integrate new information. While effective for knowledge integration, this method is characterized by its inherent susceptibility to catastrophic forgetting and model collapse.

**MEMIT** (Meng et al., 2023) is a direct model editing method designed to insert a large number of factual updates into transformer-based language models. Extending the single-edit idea of ROME (Meng et al., 2022), it performs explicit multi-layer parameter updates on a set of MLP layers identified as key mediators of factual recall. Its update rule is derived from a least-squares formulation, which enables many edits to be written into the model simultaneously rather than one at a time. As a result, MEMIT can scale to hundreds or even thousands of factual modifications while maintaining relatively strong generalization and specificity. In our experiments, we follow the original implementation and hyperparameter settings provided in the paper.

**ALPHAEDIT** (Fang et al., 2025) introduces a sophisticated approach to knowledge editing by employing a null-space constrained framework. This method operates by projecting parameter perturbations directly onto the null space of preserved knowledge, a critical mechanism that guarantees that any new updates do not inadvertently interfere with the model's existing factual information. This projection strategy eliminates the necessity for additional constraints during the optimization process, allowing the editing procedure to focus exclusively on modifying targeted knowledge without incurring trade-offs. By meticulously isolating the impact of edits, ALPHAEDIT ensures that newly injected information is integrated precisely, while the integrity and accuracy of the model's broader knowledge base remain unaffected. We implemented this baseline method using the hyperparameter configuration from their original paper.

**MEND** (Mitchell et al., 2022a) employs a meta-learning framework by training a compact hypernetwork to determine how to adjust the primary language model's parameters in response to new editing instructions. This hypernetwork is responsible for generating edit-specific gradient modifications, enabling efficient knowledge updates without necessitating a full fine-tuning process. The MEND methodology centers on learning the mechanism of editing itself, rather than directly altering the model's weights.

**MALMEN** (Tan et al., 2024) is a hypernetwork-based method for massive model editing. Its key idea is to aggregate multiple parameter shifts through a least-squares formulation, rather than naive summation, so as to better handle interference among concurrent edits. By further decoupling hypernetwork computation from base-model updates, MALMEN achieves improved memory efficiency and can scale to thousands of factual edits in a single batch.

**RLEDIT** (Li et al., 2025) is a hypernetwork-based method developed specifically for lifelong model editing. Its main idea is to cast the editing process as a reinforcement learning problem, where the hypernetwork acts as an agent that generates parameter updates according to the current model state, and editing performance is used as the reward signal. To improve long-horizon stability, RLEDIT optimizes the hypernetwork over full editing trajectories rather than isolated edits, allowing it to better adapt to continuously changing model parameters. The method further introduces memory backtracking and update regularization to reduce interference across successive edits and preserve previously edited knowledge. This design makes RLEDIT an effective approach for long sequences of sequential knowledge updates.

*Table 6.* Editable Module settings of STABLEEDIT across different models and datasets.

| Dataset | Model | Editable Module |
|---|---|---|
| ZSRE | Mistral-7B-v0.3 | `[28--31].mlp.down_proj` |
| | Llama-3-8B-Instruct | `[11--12].mlp.down_proj, [22--23].mlp.down_proj, [28--30].mlp.down_proj` |
| | GPT-J-6B | `[7--15].mlp.fc_out,[18--25].mlp.fc_out` |
| | Qwen2.5-7B-Instruct | `[8--14].mlp.down_proj, [18--25].mlp.down_proj` |
| FEVER | Mistral-7B-v0.3 | `[28-- 30].mlp.down_proj` |
| | Llama-3-8B-Instruct | `[22--26].mlp.down_proj` |
| | GPT-J-6B | `[8--14].mlp.fc_out, [18--26].mlp.fc_out` |
| | Qwen2.5-7B-Instruct | `[9--15].mlp.down_proj, [22--26].mlp.down_proj` |
| WIKIBIGEDIT | Mistral-7B-v0.3 | `[28-- 30].mlp.down_proj` |
| | Llama-3-8B-Instruct | `[11--12].mlp.down_proj, [22--23].mlp.down_proj, [28--30].mlp.down_proj` |
| | GPT-J-6B | `[8--15].mlp.fc_out, [22--24].mlp.fc_out` |
| | Qwen2.5-7B-Instruct | `[8--14].mlp.down_proj, [18--24].mlp.down_proj` |
| ULTRAEDITBENCH | Mistral-7B-v0.3 | `[28--30].mlp.down_proj` |
| | Llama-3-8B-Instruct | `[11--12].mlp.down_proj, [22--23].mlp.down_proj, [28--30].mlp.down_proj` |
| | GPT-J-6B | `[8--15].mlp.fc_out, [22--24].mlp.fc_out` |
| | Qwen2.5-7B-Instruct | `[8--14].mlp.down_proj, [18--24].mlp.down_proj` |

**ULTRAEDIT** (Gu et al., 2025) presents a highly efficient paradigm for lifelong model editing, fundamentally diverging from conventional methods by operating without the need for additional training, subject-specific assumptions, or external memory components. This approach significantly enhances scalability by calculating parameter shifts in a single step, utilizing only a hidden state and its corresponding gradient. Notably, ULTRAEDIT relies exclusively on the LN strategy and achieves dramatically faster editing speeds and requires substantially less VRAM compared to prior SOTA methods, making it uniquely capable of editing large language models on consumer-grade GPUs and supporting up to millions of edits while maintaining high accuracy and consistency.

## A.4. Implementation Details

### A.4.1. HYPERPARAMETER CONFIGURATION

Now we describe the hyperparameter configurations in our experiments. Our experiments are conducted on a single NVIDIA A6000 GPU and STABLEEDIT introduces only four hyperparameters: the learning rate $\gamma$, the ridge parameter $\lambda$, the warm-up size and the choice of editable module. We set learning rate $\gamma = 1e{-}6$ for Mistral-7B-v0.3 and Llama-3-8B-Instruct across all four datasets while $\gamma = 2e{-}6$ for GPT-J-6B and Qwen2.5-7B-Instruct. In all settings, $\lambda$ is set to 10. The warm-up size is set to 2K for standard-scale editing tasks and 20K for ultra-large-scale editing tasks. The details of the editable modules are shown in Table 6, where the numbers indicate the corresponding layer indices.

### A.4.2. ALGORITHMS

In Section 4, we derived STABLEEDIT, a more flexible and stable editor built upon the ULTRAEDIT framework. The corresponding pseudocode for STABLEEDIT editing algorithms is presented in Algorithm 1. This algorithm operates in two sequential phases: a *warm-up phase* and a *target editing phase*, both sharing the same underlying update mechanism but serving distinct purposes.

During initialization, the Normal-Inverse-Wishart (NIW) hyperparameters are set to non-informative priors for each editable layer. The warm-up phase processes a small set of in-domain edits $\{\mathcal{E}_t^{(\text{pre})}\}_{t=1}^r$ to bootstrap reliable running statistics, effectively providing "virtual samples" that shift the estimation error curve leftward (as analyzed in Section 3.2). This allows the subsequent target phase to bypass the high-estimation-error regime that would otherwise occur with a cold start.

At each editing step, the algorithm: (1) extracts hidden states and value gradients from the current model; (2) updates the NIW hyperparameters via Bayesian recursion (Theorem 3.2); (3) computes point estimates of the gradient mean and covariance; (4) applies *full whitening* using the matrix square root $\widehat{\Sigma}_{t,l}^{-1/2}$ to transform the gradients; and (5) solves the ridge regression problem to obtain the parameter update. The key distinction from ULTRAEDIT lies in the explicit warm-up stage and the use of full covariance whitening (rather than diagonal normalization), which together strengthen the self-reinforcing stability loop identified in our theoretical analysis.

---

**Algorithm 1** STABLEEDIT: Warm-up & Full Whitening for Lifelong Model Editing

---

**Input:** Pretrained model $f_{\boldsymbol{W}_0}$; editable layers $\mathcal{L}$; warm-up stream $\{\mathcal{E}_t^{(\text{pre})}\}_{t=1}^r$; target stream $\{\mathcal{E}_t^{(\text{cur})}\}_{t=1}^T$; ridge $\lambda$; step size $\gamma$; whitening ridge $\epsilon_0$.

**Output:** Edited parameters $\boldsymbol{W}_{r+T} = \{\boldsymbol{W}_{r+T,l}\}_{l \in \mathcal{L}}$.

**for all** $l \in \mathcal{L}$ **do**
$\quad (\boldsymbol{m}_{0,l}, \kappa_{0,l}, \boldsymbol{\Psi}_{0,l}, \nu_{0,l}) \leftarrow (\boldsymbol{0}, 0, \epsilon_0 \mathbf{I}, 0)$.
**end for**
$t \leftarrow 0$ // global edit step index (shared by warm-up and target phases)
**for** phase $\in \{\text{pre}, \text{cur}\}$ **do**
$\quad \mathcal{S} \leftarrow \{\mathcal{E}_j^{(\text{pre})}\}_{j=1}^r$ if phase $= \text{pre}$ else $\{\mathcal{E}_j^{(\text{cur})}\}_{j=1}^T$
$\quad$ // Warm-up stage: initialize reliable running estimates before the target stream
$\quad$ **for all** each batch $\mathcal{E} = \{(\boldsymbol{x}_t^i, \boldsymbol{y}_t^i)\}_{i=1}^{n_t}$ in $\mathcal{S}$ **do**
$\quad\quad t \leftarrow t+1, \quad \mathcal{E}_t \leftarrow \mathcal{E}$
$\quad\quad$ **for all** $l \in \mathcal{L}$ **do**
$\quad\quad\quad$ Compute $\{\boldsymbol{h}_{t,l}^i\}_{i=1}^{n_t}$ and value gradients $\{\boldsymbol{v}_{t,l}^i\}_{i=1}^{n_t}$ under current model $f_{\boldsymbol{W}_{t-1}}$ using Equation (3); set $\bar{\boldsymbol{v}}_{t,l} \leftarrow \frac{1}{n_t} \sum_{i=1}^{n_t} \boldsymbol{v}_{t,l}^i, \quad \boldsymbol{S}_{t,l} \leftarrow \sum_{i=1}^{n_t} (\boldsymbol{v}_{t,l}^i - \bar{\boldsymbol{v}}_{t,l})(\boldsymbol{v}_{t,l}^i - \bar{\boldsymbol{v}}_{t,l})^\top$. // Trace & gradients
$\quad\quad\quad$ Update $(\boldsymbol{m}_{t,l}, \kappa_{t,l}, \boldsymbol{\Psi}_{t,l}, \nu_{t,l})$ by Theorem 3.2 using $\bar{\boldsymbol{v}}_{t,l}$ and $\boldsymbol{S}_{t,l}$.
$\quad\quad\quad$ // NIW recursion: posterior→prior→post. ; warm-up posterior is inherited by the target phase
$\quad\quad\quad$ Update $\hat{\boldsymbol{\mu}}_{t,l} \leftarrow \boldsymbol{m}_{t,l}, \quad \hat{\boldsymbol{\Sigma}}_{t,l} \leftarrow \boldsymbol{\Psi}_{t,l}/(\nu_{t,l} - d - 1)$ by Corollary 3.3. // Point estimates
$\quad\quad\quad$ Compute $\tilde{\boldsymbol{v}}_{t,l}^i \leftarrow \hat{\boldsymbol{\Sigma}}_{t,l}^{-1/2}(\boldsymbol{v}_{t,l}^i - \boldsymbol{m}_{t,l})$ for $i = 1, \ldots, n_t$.
$\quad\quad\quad$ // Full whitening: $\hat{\boldsymbol{\Sigma}}_{t,l}^{-1/2}$ is computed via eigendecomposition with eigenvalue flooring (threshold) for stability
$\quad\quad\quad$ Compute $\boldsymbol{\Delta}_{t,l} \leftarrow -\gamma \sum_{i=1}^{n_t} \tilde{\boldsymbol{v}}_{t,l}^i \boldsymbol{\phi}_{t,l}^i$ with ridge projection factors $\{\boldsymbol{\phi}_{t,l}^i\}_{i=1}^{n_t}$ from Equation (6). // Ridge update
$\quad\quad\quad \boldsymbol{W}_{t,l} \leftarrow \boldsymbol{W}_{t-1,l} + \boldsymbol{\Delta}_{t,l}$
$\quad\quad$ **end for**
$\quad$ **end for**
**end for**
**return** $\boldsymbol{W}_{r+T}$

---

*Table 7.* Extended baseline comparison on standard-scale lifelong editing tasks (20K&17K edits, $n_t$=100 samples per step). Eff., Gen., and Spe. denote Efficacy, Generalization, and Specificity, respectively. The symbol ↑ indicates higher values are preferable. **Bold** and underlined entries mark the best and second-best results.

| Method | ZsRE | | | WikiBigEdit | | | | | ULTRAEDITBENCH | | |
|---|---|---|---|---|---|---|---|---|---|---|---|
| | Eff.↑ | Gen.↑ | Spe.↑ | Eff.↑ | Gen.↑ | Spe.↑ | Personas ↑ | Reasoning ↑ | Eff.↑ | Gen.↑ | Spe.↑ |
| **Mistral-7B-v0.3** | | | | | | | | | | | |
| Pre-edited | 44.46 | 43.55 | 48.08 | 39.14 | 38.21 | 41.62 | 35.54 | 29.93 | 30.83 | 29.94 | 31.11 |
| FT | 13.69 | 12.43 | 19.87 | 13.77 | 14.86 | 11.84 | 11.55 | 7.51 | 0.06 | 0.36 | 0.07 |
| MEMIT | 0.00 | 0.00 | 0.00 | 0.00 | 0.00 | 0.00 | 0.00 | 0.00 | 0.00 | 0.00 | 0.00 |
| ALPHAEDIT | 0.00 | 0.00 | 0.00 | 0.00 | 0.00 | 0.00 | 0.00 | 0.00 | 0.00 | 0.00 | 0.00 |
| MEND | 3.33 | 3.08 | 0.20 | 0.01 | 0.00 | 0.00 | 0.01 | 0.00 | 0.00 | 0.00 | 0.00 |
| MALMEN | 1.15 | 1.06 | 0.02 | 0.00 | 0.01 | 0.00 | 0.00 | 0.00 | 2.42 | 2.48 | 3.47 |
| RLEDIT | 71.62 | 67.60 | 30.70 | 57.55 | 52.47 | 28.78 | 49.21 | 22.41 | 69.85 | 65.10 | 57.08 |
| ULTRAEDIT | 85.30 | 80.80 | 47.38 | 76.00 | 70.15 | 46.09 | 62.27 | 35.80 | 83.71 | 77.30 | 67.26 |
| STABLEEDIT | **87.39** | **82.93** | **50.26** | **78.90** | **72.57** | **47.36** | **63.96** | **37.22** | **84.98** | **78.77** | **68.36** |
| **Llama-3-8B-Instruct** | | | | | | | | | | | |
| Pre-edited | 36.76 | 35.83 | 38.93 | 24.92 | 35.46 | 38.87 | 32.70 | 26.42 | 27.29 | 26.40 | 27.24 |
| FT | 18.51 | 18.11 | 5.01 | 13.00 | 11.70 | 6.75 | 11.02 | 4.38 | 13.50 | 11.92 | 10.18 |
| MEMIT | 0.14 | 0.14 | 1.40 | 0.02 | 0.02 | 0.09 | 0.02 | 0.00 | 0.74 | 0.45 | 0.21 |
| ALPHAEDIT | 86.10 | 81.15 | 26.26 | 63.24 | 54.68 | 20.17 | 42.58 | 0.01 | 4.51 | 3.16 | 2.78 |
| MEND | 0.00 | 0.00 | 0.00 | 0.26 | 0.27 | 0.18 | 0.26 | 0.34 | 0.00 | 0.00 | 0.00 |
| MALMEN | 3.86 | 3.07 | 0.26 | 0.00 | 0.00 | 0.00 | 0.00 | 0.00 | 40.64 | 34.18 | 37.29 |
| RLEDIT | **91.09** | **89.62** | 41.43 | 75.35 | 70.00 | 37.21 | 65.55 | 28.13 | 83.37 | 78.65 | 63.31 |
| ULTRAEDIT | 90.07 | 87.36 | **49.51** | 79.60 | 73.49 | **48.51** | 66.55 | **35.64** | 85.70 | 81.28 | 68.73 |
| STABLEEDIT | 90.61 | 88.38 | 48.23 | **81.22** | **75.98** | 45.22 | **70.70** | 35.03 | **88.88** | **85.46** | **69.06** |
| **GPT-J-6B** | | | | | | | | | | | |
| Pre-edited | 27.22 | 26.42 | 27.33 | 29.97 | 29.08 | 32.58 | 26.46 | 21.81 | 22.01 | 21.47 | 22.20 |
| FT | 15.11 | 13.55 | 2.61 | 21.90 | 17.56 | 8.47 | 13.91 | 9.24 | 22.03 | 17.61 | 19.60 |
| MEMIT | 0.00 | 0.00 | 0.00 | 1.59 | 1.59 | 0.50 | 0.80 | 0.00 | 0.25 | 0.18 | 0.20 |
| ALPHAEDIT | 50.23 | 43.31 | 12.54 | 69.66 | 55.03 | 21.20 | 42.60 | 0.07 | 22.19 | 12.09 | 6.88 |
| MEND | 2.52 | 2.55 | 0.19 | 0.02 | 0.01 | 0.02 | 0.02 | 0.02 | 1.71 | 1.71 | 1.83 |
| MALMEN | 0.02 | 0.01 | 0.02 | 0.00 | 0.01 | 0.01 | 0.00 | 0.00 | 0.76 | 0.48 | 0.78 |
| RLEDIT | 72.65 | 68.45 | 21.79 | 69.18 | 63.01 | 32.75 | 56.88 | 26.14 | 81.14 | 74.88 | 61.87 |
| ULTRAEDIT | 78.03 | 72.42 | 27.05 | 73.84 | 66.57 | 37.17 | 56.90 | 29.27 | **84.03** | **76.62** | 64.03 |
| STABLEEDIT | **82.12** | **78.39** | **31.52** | **75.10** | **68.13** | **44.04** | **60.53** | **37.79** | 83.41 | 76.55 | **67.17** |
| **Qwen2.5-7B-Instruct** | | | | | | | | | | | |
| Pre-edited | 34.32 | 33.39 | 38.06 | 30.97 | 30.40 | 34.50 | 28.43 | 22.09 | 26.03 | 25.22 | 26.35 |
| FT | 14.02 | 10.91 | 3.39 | 10.35 | 7.59 | 3.68 | 5.55 | 5.84 | 13.20 | 10.53 | 10.27 |
| MEMIT | 0.02 | 0.02 | 0.17 | 0.54 | 0.33 | 0.38 | 0.32 | 0.00 | 0.82 | 0.32 | 0.69 |
| ALPHAEDIT | 16.32 | 13.96 | 1.66 | 20.31 | 15.49 | 2.17 | 9.01 | 0.23 | 17.44 | 8.01 | 5.42 |
| MEND | 15.00 | 14.41 | 0.47 | 0.00 | 0.00 | 0.00 | 0.00 | 0.00 | 3.48 | 3.45 | 3.43 |
| MALMEN | 0.00 | 0.00 | 0.00 | 0.06 | 0.02 | 0.02 | 0.00 | 0.00 | 4.36 | 3.48 | 4.38 |
| RLEDIT | **84.70** | **82.79** | 38.26 | 55.88 | 47.66 | 29.48 | 41.14 | 22.21 | 80.25 | **75.93** | 64.91 |
| ULTRAEDIT | 82.03 | 77.08 | 45.51 | **73.37** | **65.86** | 45.12 | 54.65 | 32.74 | 79.01 | 71.45 | 64.10 |
| STABLEEDIT | 80.40 | 76.65 | **47.59** | 68.95 | 63.52 | **46.94** | **58.66** | **36.48** | **80.29** | 74.00 | **66.53** |

## B. More Experimental Results

### B.1. Extended Baseline Comparison

We provide comprehensive experimental results on standard-scale lifelong editing sequences in Tables 7 and 8, evaluating STABLEEDIT against seven baseline methods across four representative LLMs (Mistral-7B-v0.3, Llama-3-8B-Instruct, GPT-J-6B, and Qwen2.5-7B-Instruct) and four benchmarks (ZsRE, WikiBigEdit, ULTRAEDITBENCH, and FEVER). All experiments are conducted with 20K&17K total edits at a batch size of $n_t$=100 samples per step, measuring Efficacy, Generalization, and Specificity, with WikiBigEdit additionally assessing Personas and multi-hop Reasoning capabilities. The

*Table 8.* Extended results on FEVER under the standard-scale lifelong editing setting (20K edits, $n_t$=100 samples per step).

| Method | FEVER | | | | | | | | | | | |
|---|---|---|---|---|---|---|---|---|---|---|---|---|
| | Mistral-7B-v0.3 | | | Llama-3-8B-Instruct | | | GPT-J-6B | | | Qwen2.5-7B-Instruct | | |
| | Eff.↑ | Gen.↑ | Spe.↑ | Eff.↑ | Gen.↑ | Spe.↑ | Eff.↑ | Gen.↑ | Spe.↑ | Eff.↑ | Gen.↑ | Spe.↑ |
| Pre-edited | 0.41 | 0.50 | 1.98 | 0.02 | 0.02 | 0.26 | 9.61 | 9.68 | 15.90 | 0.57 | 0.60 | 2.17 |
| FT | 23.80 | 23.37 | 16.30 | 16.21 | 13.08 | 5.01 | 14.02 | 13.98 | 9.26 | 26.09 | 24.62 | 21.36 |
| MEMIT | 0.00 | 0.00 | 0.00 | 0.02 | 0.02 | 0.02 | 5.54 | 5.03 | 5.46 | 0.08 | 0.12 | 0.15 |
| ALPHAEDIT | 0.00 | 0.00 | 0.00 | 6.39 | 6.14 | 2.72 | 1.89 | 1.87 | 1.85 | 32.78 | 31.19 | 22.12 |
| MEND | 0.00 | 0.00 | 0.00 | 36.19 | 36.19 | 24.31 | 52.80 | 51.44 | 45.42 | 76.43 | 77.66 | 40.39 |
| MALMEN | 18.42 | 17.43 | 12.09 | 95.03 | 92.44 | 69.20 | 1.33 | 1.25 | 2.92 | 0.06 | 0.06 | 0.07 |
| RLEDIT | 92.35 | 91.39 | 71.85 | 91.38 | 89.93 | 41.98 | 97.29 | 96.01 | **79.86** | 96.93 | 93.92 | 68.28 |
| ULTRAEDIT | 97.87 | 96.09 | **84.29** | 95.39 | 91.93 | 67.14 | 97.45 | 96.37 | 79.72 | 97.97 | 93.91 | 68.86 |
| STABLEEDIT | **98.38** | **96.55** | 83.89 | **97.89** | **94.47** | 69.24 | **98.35** | **96.99** | 79.85 | **99.07** | **94.87** | **69.61** |

*Table 9.* Extended results on WikiBigEdit at **500K edits** ($n_t$=100 samples per step) across four models.

| Method | WikiBigEdit | | | | | | | |
|---|---|---|---|---|---|---|---|---|
| | Mistral-7B-v0.3 | | | | Llama-3-8B-Instruct | | | |
| | Eff.↑ | Gen.↑ | Spe.↑ | Personas ↑ | Eff.↑ | Gen.↑ | Spe.↑ | Personas ↑ |
| Pre-edited | 39.14 | 38.21 | 41.62 | 35.54 | 24.92 | 35.46 | 38.87 | 32.70 |
| ULTRAEDIT | 71.78 | 65.63 | 55.40 | 56.11 | 68.99 | 63.59 | 52.28 | 55.04 |
| STABLEEDIT | **74.54** | **67.80** | **56.49** | **58.88** | **72.44** | **66.87** | **53.83** | **60.39** |

| Method | GPT-J-6B | | | | Qwen2.5-7B-Instruct | | | |
|---|---|---|---|---|---|---|---|---|
| | Eff.↑ | Gen.↑ | Spe.↑ | Personas ↑ | Eff.↑ | Gen.↑ | Spe.↑ | Personas ↑ |
| Pre-edited | 29.97 | 29.08 | 32.58 | 26.46 | 30.97 | 30.40 | 34.50 | 28.43 |
| ULTRAEDIT | 66.46 | 60.54 | 47.90 | 51.73 | 62.73 | 57.19 | 48.52 | 49.04 |
| STABLEEDIT | **71.47** | **65.04** | **52.84** | **57.20** | **69.20** | **62.95** | **54.09** | **54.05** |

results highlight a sharp contrast between editors *without* LN and those that *incorporate* LN. Non-LN methods (FT, MEMIT, ALPHAEDIT) exhibit a consistent failure mode: performance degrades rapidly and often collapses to near-zero under long sequential updates, indicating poor robustness in lifelong settings. In contrast, LN-based methods (RLEDIT, ULTRAEDIT, STABLEEDIT) consistently maintain robust performance across all configurations. STABLEEDIT achieves best or competitive results in the vast majority of cases, with particularly notable improvements on challenging benchmarks: on GPT-J-6B with WikiBigEdit, STABLEEDIT improves Specificity from 37.17% to 44.04% and Reasoning from 29.27% to 37.79%; on Llama-3-8B-Instruct with FEVER, STABLEEDIT improves Efficacy/Generalization/Specificity by 2.62%/2.76%/3.13% relative to ULTRAEDIT. Introducing LN is necessary, yet it is not sufficient on its own: while RLEDIT, ULTRAEDIT, and our STABLEEDIT remain stable across most configurations, other LN-based approaches (e.g., MEND, MALMEN) still struggle in lifelong regimes due to their hypernetwork training being mismatched with dynamically evolving LLM parameters during editing. It underscores that LN must be paired with an update mechanism that remains calibrated under sequential distribution shift. Overall, these gains are consistent with our theory: the warm-up stage provides effective "virtual samples" that bypass the high-estimation-error phase of early updates (Section 3.2), and full whitening improves the update geometry by boosting weak eigen-directions while damping high-variance components. Together, they preserve the stability loop in Section 4—accurate tracking yields calibrated updates, which constrain drift and enable continued stable editing over long sequences.

We further validate the scalability of our approach under ultra-large-scale editing scenarios with 500K sequential edits on WikiBigEdit, as reported in Table 9. At this extreme scale, most baselines become computationally infeasible or completely collapse, leaving ULTRAEDIT as a viable comparison. The results demonstrate that STABLEEDIT consistently outperforms ULTRAEDIT across all four models on every metric, with the performance gap becoming more pronounced as edits accumulate. On GPT-J-6B, STABLEEDIT achieves 71.47%/65.04%/52.84%/57.20% (Efficacy/Generalization/Specificity/Personas) compared to ULTRAEDIT's 66.46%/60.54%/47.90%/51.73%, representing relative improvements of 7.5%, 7.4%, 10.3%, and 10.6%, respectively. Similar gains are observed on Qwen2.5-7B-Instruct, where STABLEEDIT improves Efficacy from

*Table 10.* Results on the medical benchmark MedCF.

| Model | Method | MEDCF | | |
| --- | --- | --- | --- | --- |
| | | Eff. | Gen. | Spe. |
| Mistral-7B-v0.3 | ULTRAEDIT | 60.34 | 59.67 | 47.76 |
| | STABLEEDIT | **68.83** | **63.48** | **51.61** |
| Llama-3-8B-Instruct | ULTRAEDIT | 57.64 | 54.66 | 39.37 |
| | STABLEEDIT | **63.38** | **58.62** | **42.90** |
| GPT-J-6B | ULTRAEDIT | **57.08** | **55.32** | 37.75 |
| | STABLEEDIT | 54.73 | 52.00 | **43.28** |

62.73% to 69.20% and Personas from 49.04% to 54.05%. These sustained advantages at ultra-large scale further corroborate the effectiveness of our theoretical enhancements.

### B.2. Domain Generalization

To assess whether STABLEEDIT generalizes beyond the current benchmark suite, we evaluate it on the medical knowledge editing benchmark MedCF (Xu et al., 2024). Crucially, we do *not* use a domain-specific medical LLM, do *not* perform MedCF-specific hyperparameter tuning, and do *not* adjust the editable layers for this dataset. As shown in Table 10, STABLEEDIT outperforms ULTRAEDIT on both Mistral-7B-v0.3 and Llama-3-8B-Instruct across all three metrics. On GPT-J-6B, STABLEEDIT yields higher Specificity but slightly lower Efficacy and Generalization. Overall, these results suggest STABLEEDIT is not tied to the current benchmark suite and can transfer to a new domain even without domain-specific adaptation.

### B.3. Scaling Trajectories and Long-Horizon Diagnostics

This section consolidates the evidence on how editing performance evolves as the edit stream grows, and on the covariance-side diagnostics that explain the observed trajectory shape. Part B.3.1 tracks cross-method trajectories on ULTRAEDIT-BENCH from 50K to 2M edits; Part B.3.2 tracks STABLEEDIT vs. ULTRAEDIT on WikiBigEdit from 20K to 500K edits; Part B.3.3 tracks the condition number and the largest eigenvalue of the running covariance, discussing a possible contributing factor to the weaker early-stage performance observed in certain settings and examining the practical scale of the irreducible residual in Theorem 3.6.

#### B.3.1. CROSS-METHOD TRAJECTORIES ON ULTRAEDITBENCH (50K–2M)

**Robust Stability Across Long Edit Sequences.** Figure 5 demonstrates the editing dynamics on ULTRAEDITBENCH as the number of sequential edits scales from 50K to 2M. A key observation is that all three methods—RLEDIT, ULTRAEDIT, and STABLEEDIT—maintain remarkably stable performance trajectories throughout the entire editing process. This robustness can be attributed to their shared core component: LN, which continuously tracks and calibrates the gradient distribution. The absence of catastrophic performance degradation, even at the 2M-edit scale, empirically validates our theoretical analysis that LN establishes a self-reinforcing stability loop, effectively mitigating both catastrophic forgetting and model collapse.

**Consistent Superiority of STABLEEDIT.** Across both Mistral-7B-v0.3 and Llama-3-8B-Instruct, STABLEEDIT consistently achieves the highest or competitive performance on all three metrics. These results confirm that explicitly strengthening the stability loop via warm-up and full whitening yields tangible benefits in long-horizon lifelong model editing scenarios.

#### B.3.2. LONG-HORIZON TRAJECTORIES ON WIKIBIGEDIT (20K–500K)

To provide more trajectory-level evidence for the long-horizon robustness claim, we plot the full performance trajectories on WikiBigEdit from 20K to 500K edits for both Llama-3-8B-Instruct and Qwen2.5-7B-Instruct in Figure 6. Although STABLEEDIT can be slightly behind ULTRAEDIT at earlier checkpoints on some metrics (e.g., Specificity on Llama-3-8B-Instruct), it catches up and eventually surpasses ULTRAEDIT as edits accumulate. On Qwen2.5-7B-Instruct, ULTRAEDIT drops sharply from 400K to 500K edits, while STABLEEDIT remains much more stable. This trajectory-level evidence

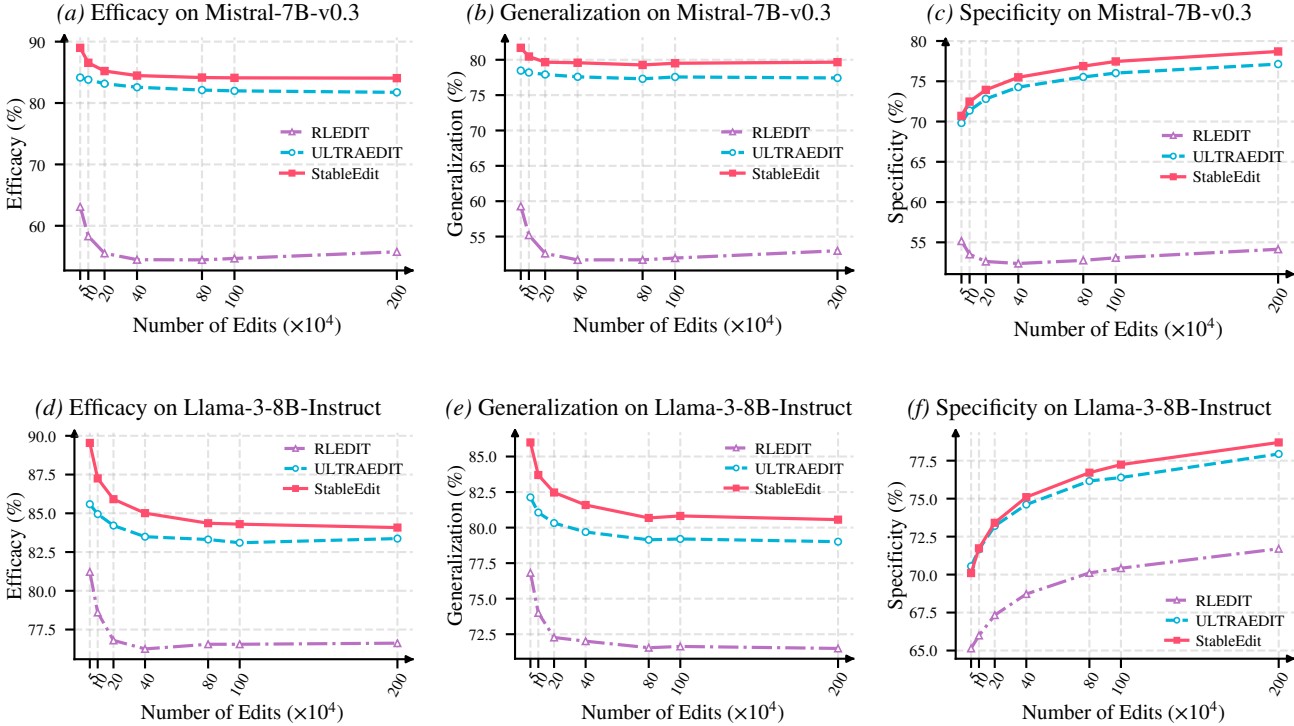

*Figure 5.* Editing dynamics on ULTRAEDITBENCH at scale. Performance trajectories of Efficacy, Generalization, and Specificity as the number of sequential edits increases (50K–2M). Each row compares RLEDIT, ULTRAEDIT, and STABLEEDIT: **top**—Mistral-7B-v0.3 on ULTRAEDITBENCH; **bottom**—Llama-3-8B-Instruct on ULTRAEDITBENCH.

aligns more faithfully with our main claim of better long-horizon robustness than a single-checkpoint comparison.

### B.3.3. COVARIANCE CONDITIONING AND EIGENVALUE DYNAMICS

We track the condition number of the running covariance across editing steps (Figure 7) and the largest eigenvalue of the covariance (Figure 8) as complementary diagnostics alongside the trajectory-level results in Figure 6. As shown in Figure 7, the condition number decreases substantially during the early editing phase and then stabilizes into a controlled range. Early in editing, the covariance is more ill-conditioned, which may cause whitening to over-amplify small-eigenvalue directions; this is a possible contributing factor to the weaker early-stage performance observed in some settings. As editing proceeds, the better-conditioned covariance leads to a more balanced whitening transform. On Qwen2.5-7B-Instruct, the condition number rises temporarily around 400K edits, yet STABLEEDIT remains stable while ULTRAEDIT degrades more noticeably at this point.

Figure 8 shows that the largest eigenvalues of the running covariance remain small in practice, typically on the order of $10^{-4}$ to $10^{-6}$, and tend to decrease over the course of editing. This is relevant to assessing the practical scale of the irreducible residual in Theorem 3.6. In our experiments, the editable-layer width is $d \approx 4,096$, the per-step batch size is $n = 100$, and the largest eigenvalue $\sigma_+$ of the running covariance is at most on the order of $10^{-4}$ in practice (often smaller, as shown in the figure). Although the residual term in Theorem 3.6 depends explicitly on $d$ and $\sigma_+$, its effective magnitude is moderate: with $d \sim 10^3$, $n = 100$, and $\sigma_+ \sim 10^{-4}$ to $10^{-6}$, the term remains small relative to the signal. These results suggest that the covariance estimation is sufficiently accurate in our experimental setting, and the resulting whitening transform remains well-conditioned throughout the editing stream.

### B.4. Extended Runtime Comparison

In Table 3, we find that compared with RLEDIT, MEND, MALMEN, and ALPHAEDIT, STABLEEDIT maintains substantially lower per-step runtime, while introducing only a small overhead relative to ULTRAEDIT. The additional

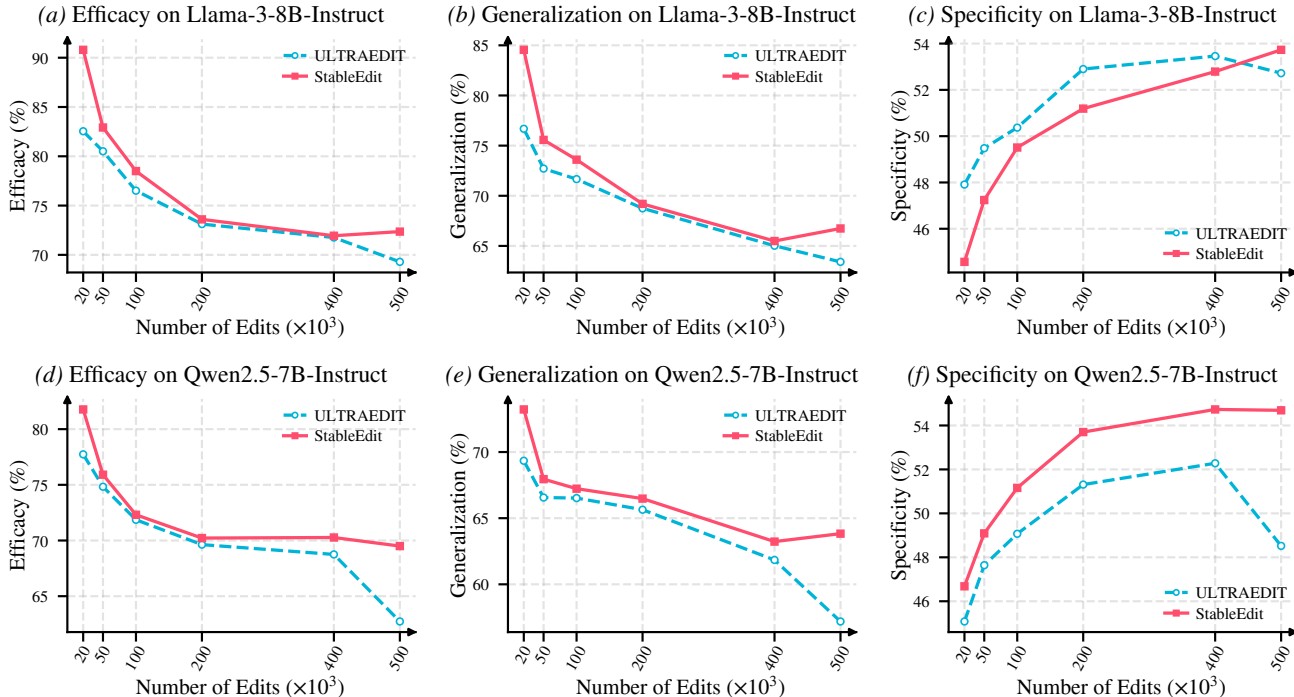

*Figure 6.* Editing dynamics on WikiBigEdit at scale. Performance trajectories of Efficacy, Generalization, and Specificity as the number of sequential edits increases (20K–500K). Each row compares ULTRAEDIT and STABLEEDIT: **top**—Llama-3-8B-Instruct; **bottom**—Qwen2.5-7B-Instruct.

cost relative to ULTRAEDIT is concentrated in computing the full whitening transform: STABLEEDIT performs a full eigendecomposition at each step, whereas ULTRAEDIT uses only diagonal normalization. For an editable layer of width $d$, the eigendecomposition costs $O(d^3)$ time with $O(d^2)$ memory, and the whitening update adds an $O(nd^2)$ term for edit batch size $n$. Crucially, this overhead scales with the *editable-layer width* rather than the total parameter count of the LLM; even for very large models (e.g., 70B parameters), the editable-layer width typically remains in the low-thousands range (around 8192). In our experiments, we edit only a small number of MLP `down_proj` layers, with $d \approx 4{,}096$ (and typically still in the low-thousands range for larger models). In this regime, the overhead is modest in practice. The reported runtime *already includes the full warm-up cost*: we measure the total wall-clock time of the entire editing procedure (warm-up plus target edits) and then report the average time per target edit.

### B.5. Hidden Representations Analysis

To further investigate the internal impact of lifelong editing on model representations, we visualize the hidden states of pre-edited and post-edited LLMs using UMAP dimensionality reduction in Figure 9. Specifically, we extract hidden representations from 1,000 factual prompts unrelated to the editing targets and project them onto a two-dimensional space, comparing the distributional alignment between the original model and the model after sequential edits. We evaluate three LN-based methods (STABLEEDIT, ULTRAEDIT, RLEDIT) on GPT-J-6B and Mistral-7B-v0.3, with dashed lines indicating the 0.95 confidence intervals of each distribution.

The visualizations reveal clear differences in representation stability across methods. STABLEEDIT and ULTRAEDIT exhibit smaller distributional shift, with the post-edited representations remaining well-aligned within the confidence region of the pre-edited distribution on both model architectures. In contrast, RLEDIT induces more pronounced distributional drift, particularly visible on Mistral-7B-v0.3 where the post-edited representations show noticeable displacement from the original manifold. Overall, these results suggest that LN-based editors (STABLEEDIT, ULTRAEDIT, and RLEDIT) tend to better preserve alignment with the pre-edited representation manifold—maintaining greater consistency with the original model state and thereby reducing the risk of overfitting to the edited requests. This observation provides direct empirical support for our theoretical analysis: the bounded update norm guarantee established in Corollary 3.9 ensures that cumulative

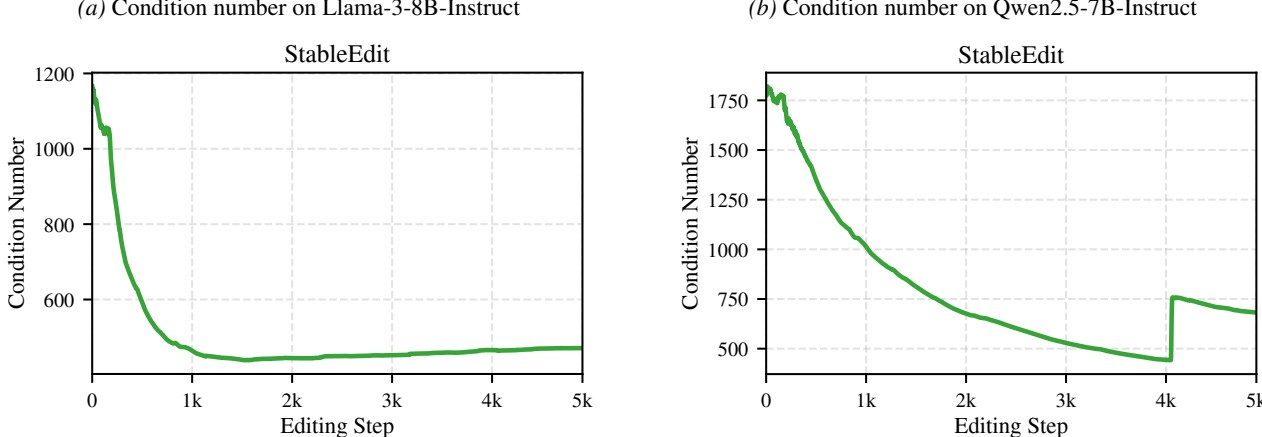

*Figure 7.* Condition number of the covariance matrix across editing steps on WikiBigEdit under the 500K-edit setting for Llama-3-8B-Instruct and Qwen2.5-7B-Instruct. The condition number decreases substantially during early editing and then stabilizes, indicating progressively better-conditioned whitening.

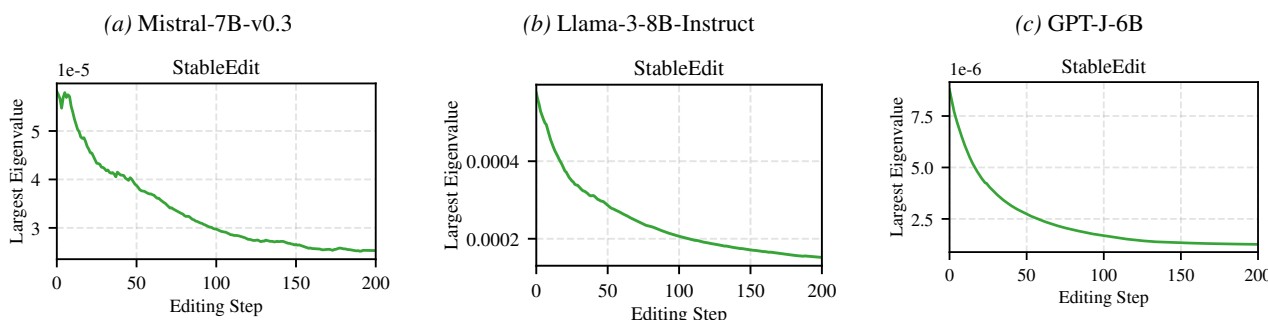

*Figure 8.* Largest eigenvalue of the covariance matrix across editing steps on ULTRAEDITBENCH under the 20K-edit setting for Mistral-7B-v0.3, Llama-3-8B-Instruct, and GPT-J-6B. The eigenvalues remain small (typically $10^{-4}$ to $10^{-6}$) and tend to decrease over time.

parameter modifications remain controlled, preventing the explosive representational drift characteristic of model collapse. The superior alignment observed with STABLEEDIT demonstrates that our enhancements—warm-up initialization and full whitening—effectively strengthen these theoretical guarantees, maintaining representation fidelity throughout extended editing sequences.

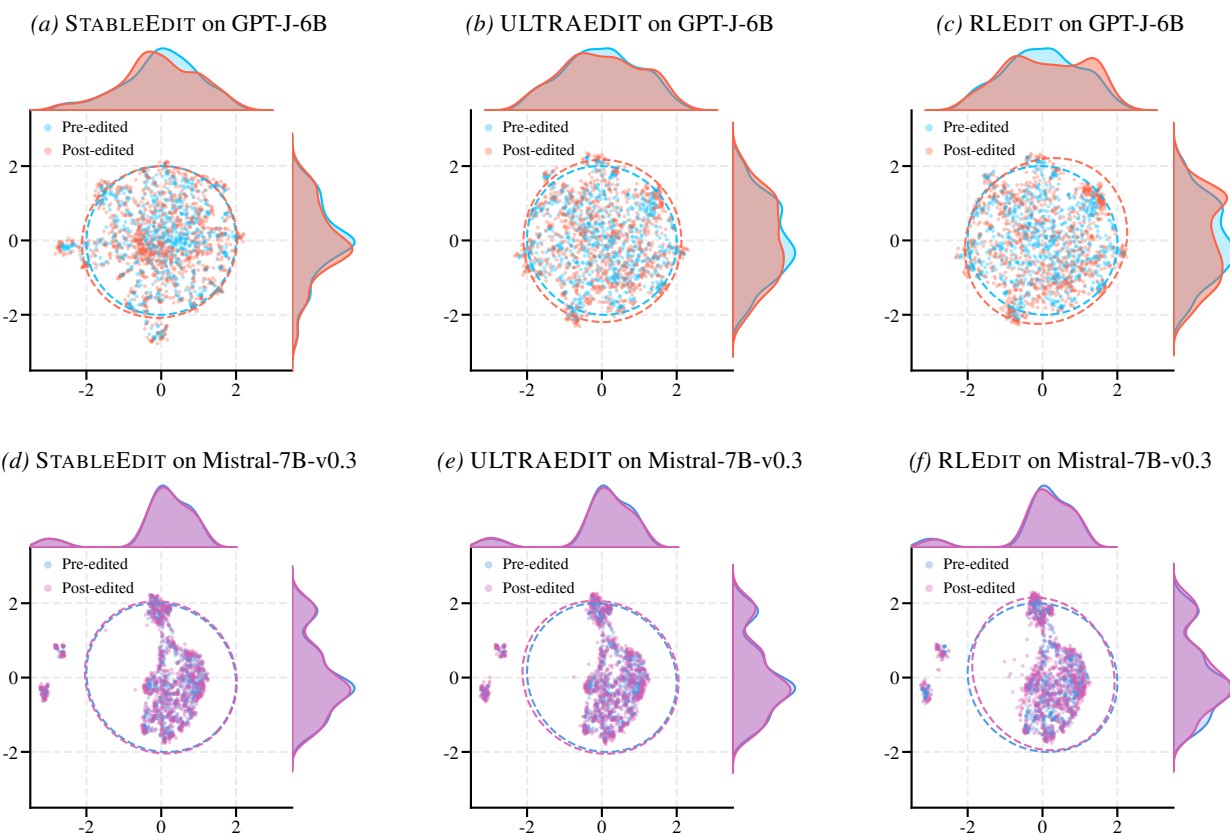

*Figure 9.* Representation shift under lifelong editing. UMAP projections of the *final-layer* output hidden states computed on the same randomly sampled set of 1,000 factual prompts (independent of the edit stream), comparing the pre-edit model and the corresponding post-edit model. For each prompt, we feed identical inputs to the LLMs and project the resulting hidden states into 2D, visualizing the joint distribution (scatter) together with marginal densities along each reduced dimension (top and right). Dashed lines indicate the 0.95 confidence intervals of the marginals. Best viewed in color.

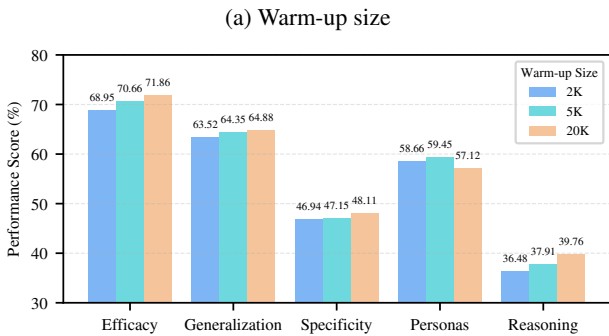
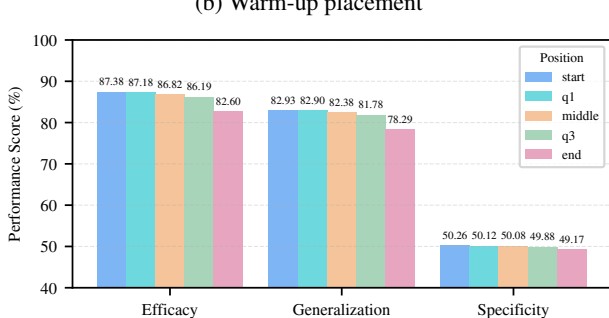

*Figure 10.* Warm-up ablations for STABLEEDIT. **Left:** Qwen2.5-7B-Instruct on WikiBigEdit. **Right:** Mistral-7B-v0.3 on ZsRE. We report final editing performance under varying warm-up sizes (2K, 5K, and 20K edits) and placements along the edit stream: *start*, *q1*, *middle*, *q3*, and *end*, where *q1/q3* denote inserting warm-up at the 1st/3rd quartile (25%/75%) of the total editing steps. Larger warm-up sizes and earlier placement consistently improve all metrics, supporting the cumulative benefit of warm-up.

## B.6. More Results on Verification of the LN Mechanism

This section consolidates the mechanism-verification experiments that support Section 3 and Section 5.4. It is organized into five parts: B.6.1 probes the four design choices of the warm-up stage; B.6.2 verifies the bounded-norm and near-orthogonality properties derived in Theorem 3.8; B.6.3 reports observable drift-surrogate curves that provide empirical evidence for Assumption 3.4; B.6.4 tests robustness under an abrupt non-i.i.d. target stream; and B.6.5 isolates the role of the hidden-state scalar $\|\tilde{h}\|_2^2$ to localize the source of the stability gains.

### B.6.1. WARM-UP DYNAMICS

The warm-up stage of STABLEEDIT involves several design choices that affect the quality of the initial running statistics. We investigate four aspects: **(i)** the number of warm-up edits used, **(ii)** where in the edit stream the warm-up is inserted, **(iii)** whether the warm-up data must come from the same distribution as the target benchmarks, and **(iv)** whether warm-up needs to perform actual parameter updates (rather than only updating the running statistics). Aspects **(i)** and **(ii)** are studied under the same-distribution setting in Figure 10; aspect **(iii)** is the distribution-mismatch test, for which Table 5 in the main text reports a compact Mistral-7B-v0.3 view and the full three-backbone tables are Tables 11 and 12 below; aspect **(iv)** is the statistics-only variant Warmup-100 in Table 13. Together, these results support the main-text claim in Obs 7 that warm-up helps primarily by establishing reliable initial running statistics and does not require the warm-up distribution to match the target stream.

**(i)-(ii) Warm-up size and placement under same-distribution warm-up.** We study how warm-up affects final performance on a target edit sequence along two axes: *warm-up size* (2K/5K/20K edits) and *warm-up placement* (inserting the warm-up block at start, q1, middle, q3, or end of the target sequence), where q1 and q3 denote insertion at the 25% and 75% quantile positions of the target edit stream, respectively. Warm-up edits are drawn i.i.d. from the same distribution as the target edits. As shown in Figure 10, increasing the warm-up size consistently improves performance across all five metrics (Efficacy, Generalization, Specificity, Personas, and Reasoning). Furthermore, placing warm-up edits earlier in the target sequence yields larger gains than inserting them mid-stream. These observations empirically validate our theoretical analysis: Theorems 3.5 and 3.6 establish that accumulated edits provide "virtual samples" that shift the estimation error curve leftward, enabling the tracker to bypass the high-error regime characteristic of cold starts. The warm-up phase effectively preconditions the running statistics, allowing subsequent edits to benefit from more accurate gradient whitening from the outset.

**(iii) Warm-up distribution robustness.** A natural concern is whether STABLEEDIT relies on in-domain warm-up data. We conduct two stress tests. First, we replace the warm-up data with the out-of-domain medical dataset MedCF (Xu et al., 2024), keeping the downstream editing benchmarks (ZsRE and FEVER) unchanged (Table 11). Across 18 reported entries, the average absolute change relative to STABLEEDIT is only 0.43 points, and the mismatched-warm-up variant still outperforms ULTRAEDIT on 15 out of 18 entries. Second, we use ZsRE as warm-up while evaluating on FEVER and ULTRAEDITBENCH (Table 12). Here the average absolute change is only 0.26 points, and the variant outperforms

*Table 11.* Ablation on the warm-up data distribution. ULTRAEDIT is reported as a baseline reference. Warmup-MedCF replaces the warm-up data of STABLEEDIT with the out-of-domain medical dataset MedCF, while leaving the downstream editing benchmarks unchanged. This comparison tests whether STABLEEDIT remains effective when the warm-up stage is performed on a mismatched distribution. For Warmup-MedCF, the value in parentheses reports the absolute change relative to STABLEEDIT. Boldfaced numbers mark the better result between STABLEEDIT and Warmup-MedCF for each model-dataset-metric entry.

| Model | Method | ZsRE | | | FEVER | | |
|---|---|---|---|---|---|---|---|
| | | Eff. | Gen. | Spe. | Eff. | Gen. | Spe. |
| Mistral-7B-v0.3 | ULTRAEDIT | 85.30 | 80.80 | 47.38 | 97.87 | 96.09 | 84.29 |
| | STABLEEDIT | 87.39 | 82.93 | 50.26 | **98.38** | **96.55** | **83.89** |
| | Warmup-MedCF | **87.63** (↑ 0.24) | **83.09** (↑ 0.16) | **51.28** (↑ 1.02) | 98.36 (↓ 0.02) | 96.45 (↓ 0.10) | 82.90 (↓ 0.99) |
| Llama-3-8B-Instruct | ULTRAEDIT | 90.07 | 87.36 | 49.51 | 95.39 | 91.93 | 67.14 |
| | STABLEEDIT | **90.61** | **88.38** | **48.23** | 97.89 | 94.47 | **69.24** |
| | Warmup-MedCF | 90.25 (↓ 0.36) | 87.97 (↓ 0.41) | 47.31 (↓ 0.92) | **98.10** (↑ 0.21) | **94.74** (↑ 0.27) | 69.00 (↓ 0.24) |
| GPT-J-6B | ULTRAEDIT | 78.03 | 72.42 | 27.05 | 97.45 | 96.37 | 79.72 |
| | STABLEEDIT | **82.12** | **78.39** | 31.52 | 98.35 | 96.99 | **79.85** |
| | Warmup-MedCF | 81.79 (↓ 0.33) | 77.76 (↓ 0.63) | **32.13** (↑ 0.61) | **99.01** (↑ 0.66) | **97.15** (↑ 0.16) | 79.38 (↓ 0.47) |

*Table 12.* Robustness to warm-up distribution shift on FEVER and ULTRAEDITBENCH under the 20K-edit setting across three backbone LLMs. ULTRAEDIT is included as a baseline reference. Warmup-ZsRE denotes the variant of STABLEEDIT that replaces the original warm-up data with ZsRE, while keeping the downstream editing benchmarks unchanged. For Warmup-ZsRE, the value in parentheses reports the absolute change relative to STABLEEDIT. Boldfaced numbers indicate the better result between the two STABLEEDIT variants (STABLEEDIT vs. Warmup-ZsRE) for each model-dataset-metric entry.

| Model | Method | FEVER | | | ULTRAEDITBENCH | | |
|---|---|---|---|---|---|---|---|
| | | Eff. | Gen. | Spe. | Eff. | Gen. | Spe. |
| Mistral-7B-v0.3 | ULTRAEDIT | 97.87 | 96.09 | 84.29 | 83.71 | 77.30 | 67.26 |
| | STABLEEDIT | **98.38** | **96.55** | **83.89** | **84.98** | **78.77** | **68.36** |
| | Warmup-ZsRE | 98.37 (↓ 0.01) | 96.38 (↓ 0.17) | 83.75 (↓ 0.14) | 84.80 (↓ 0.18) | 78.49 (↓ 0.28) | 67.89 (↓ 0.47) |
| Llama-3-8B-Instruct | ULTRAEDIT | 95.39 | 91.93 | 67.14 | 85.70 | 81.28 | 68.73 |
| | STABLEEDIT | 97.89 | 94.47 | **69.24** | **88.88** | **85.46** | **69.06** |
| | Warmup-ZsRE | **97.96** (↑ 0.07) | **94.73** (↑ 0.26) | 68.94 (↓ 0.30) | 88.53 (↓ 0.35) | 85.23 (↓ 0.23) | 68.97 (↓ 0.09) |
| GPT-J-6B | ULTRAEDIT | 97.45 | 96.37 | 79.72 | 84.03 | 76.62 | 64.03 |
| | STABLEEDIT | 98.35 | **96.99** | **79.85** | 83.41 | 76.55 | **67.17** |
| | Warmup-ZsRE | **98.76** (↑ 0.41) | 96.97 (↓ 0.02) | 79.23 (↓ 0.62) | **83.94** (↑ 0.53) | **76.75** (↑ 0.20) | 66.76 (↓ 0.41) |

ULTRAEDIT on 15 out of 18 entries. Together, these results demonstrate that the benefit of warm-up is not tied to strict same-distribution matching: even under substantial domain mismatch, the initialized statistics remain useful and are further adapted online during the target stream, directly supporting the main-text claim in Obs 7.

**(iv) Statistics-only warm-up.** Because warm-up performs actual parameter updates, the performance gains it produces could stem from two sources: (a) better-initialized running statistics used for gradient normalization, or (b) the model weights being shifted to a region where subsequent editing is easier. To isolate the effect of statistic preconditioning from the effect of extra editing updates, we construct **Warmup-100**, which uses 100 warm-up examples to update the running statistics in a one-shot manner without applying any parameter edits. All other settings remain unchanged. As shown in Table 13, Warmup-100 remains very close to the full STABLEEDIT: the average absolute change across all reported metrics is only 0.52 points (signed mean change of −0.38 points), and Warmup-100 still outperforms ULTRAEDIT on 22 out of 27 reported entries. This confirms that the primary role of warm-up is to precondition the running statistics, which is consistent with our theoretical analysis in Theorems 3.5 and 3.6.

### B.6.2. UPDATE GEOMETRY

This section provides more evidence for the bounded-norm and near-orthogonality properties derived in Theorem 3.8. We examine the geometric properties of parameter updates by measuring the cosine similarity between adjacent update increments $\langle \Delta_{t,l}, \Delta_{t-1,l} \rangle / (\|\Delta_{t,l}\|_F \|\Delta_{t-1,l}\|_F)$ and their Frobenius norms throughout the editing stream. Figure 11 verifies Theorem 3.8(c) by tracking cosine similarity with and without LN on four benchmarks; Figure 12 verifies Theorem 3.8(a)-(b)

*Table 13.* Ablation on statistics-only warm-up across ZsRE, FEVER, and ULTRAEDITBENCH. ULTRAEDIT is included as a baseline reference. Warmup-100 denotes the variant of STABLEEDIT that uses 100 warm-up examples to update the running statistics in a one-shot manner, without applying any parameter edits during the warm-up stage. For Warmup-100, the value in parentheses reports the absolute change relative to STABLEEDIT. Boldfaced numbers indicate the better result between STABLEEDIT and Warmup-100 for each model-dataset-metric entry.

| Model | Method | ZsRE | | | FEVER | | | ULTRAEDITBENCH | | |
|---|---|---|---|---|---|---|---|---|---|---|
| | | Eff. | Gen. | Spe. | Eff. | Gen. | Spe. | Eff. | Gen. | Spe. |
| Mistral-7B-v0.3 | ULTRAEDIT | 85.30 | 80.80 | 47.38 | 97.87 | 96.09 | 84.29 | 83.71 | 77.30 | 67.26 |
| | STABLEEDIT | 87.39 | 82.93 | **50.26** | 98.38 | **96.55** | **83.89** | **84.98** | **78.77** | **68.36** |
| | Warmup-100 | **87.58** (↑ 0.19) | **83.25** (↑ 0.32) | 49.91 (↓ 0.35) | **98.38** (±0.00) | 96.49 (↓ 0.06) | 83.65 (↓ 0.24) | 84.24 (↓ 0.74) | 77.94 (↓ 0.83) | 67.85 (↓ 0.51) |
| Llama-3-8B-Instruct | ULTRAEDIT | 90.07 | 87.36 | 49.51 | 95.39 | 91.93 | 67.14 | 85.70 | 81.28 | 68.73 |
| | STABLEEDIT | **90.61** | **88.38** | **48.23** | 97.89 | 94.47 | **69.24** | **88.88** | **85.46** | **69.06** |
| | Warmup-100 | 90.02 (↓ 0.59) | 87.62 (↓ 0.76) | 46.06 (↓ 2.17) | **97.96** (↑ 0.07) | **94.52** (↑ 0.05) | 68.90 (↓ 0.34) | 87.71 (↓ 1.17) | 84.00 (↓ 1.46) | 68.77 (↓ 0.29) |
| GPT-J-6B | ULTRAEDIT | 78.03 | 72.42 | 27.05 | 97.45 | 96.37 | 79.72 | 84.03 | 76.62 | 64.03 |
| | STABLEEDIT | 82.12 | **78.39** | **31.52** | 98.35 | **96.99** | **79.85** | 83.41 | 76.55 | **67.17** |
| | Warmup-100 | **82.15** (↑ 0.03) | 78.27 (↓ 0.12) | 30.12 (↓ 1.40) | **98.74** (↑ 0.39) | 96.82 (↓ 0.17) | 79.26 (↓ 0.59) | **84.18** (↑ 0.77) | **76.62** (↑ 0.07) | 66.78 (↓ 0.39) |

by tracking per-step update norms in the same four settings.

Specifically, Figures 11 and 12 present results across four datasets (ZsRE, FEVER, WikiBigEdit, and ULTRAEDITBENCH) on GPT-J-6B, comparing three editors (RLEDIT, ULTRAEDIT, and STABLEEDIT) with and without the LN module. When LN is enabled, cosine similarities between adjacent parameter increments oscillate tightly around zero, indicating that successive updates are approximately orthogonal and thus weakly interfering, while update magnitudes remain within a reasonable range throughout the sequence. In stark contrast, removing LN causes cosine similarities to drift substantially away from zero, and update norms collapse to extremely small values, leading to severe under-fitting. These empirical patterns precisely match Theorem 3.8: LN combined with ridge-regularized regression enforces *asymptotic orthogonality* (Property (c)) and *bounded update norms* (Properties (a) and (b)). The orthogonality property directly mitigates catastrophic forgetting by reducing cross-step interference, while bounded norms prevent systemic model collapse. Notably, STABLEEDIT reaches the near-orthogonal regime earlier than ULTRAEDIT, stabilizing close to zero from the initial editing steps. This accelerated stabilization stems from the explicit warm-up stage, which strengthens the self-reinforcing stability loop by providing well-calibrated distributional estimates from the beginning.

### B.6.3. EMPIRICAL SUPPORT FOR ASSUMPTION 3.4 (DRIFT SURROGATES)

Since $(\boldsymbol{\mu}_{t,l}, \boldsymbol{\Sigma}_{t,l})$ in Assumption 3.4 are latent and cannot be verified directly, we track the consecutive-step drifts of the running estimates, $\|\hat{\boldsymbol{\mu}}_{t,l} - \hat{\boldsymbol{\mu}}_{t-1,l}\|$ and $\|\hat{\boldsymbol{\Sigma}}_{t,l} - \hat{\boldsymbol{\Sigma}}_{t-1,l}\|$, as empirical surrogates across architectures and editing scales (Figures 13 and 14). On FEVER (20K edits), the mean-drift surrogate drops sharply after warm-up and typically falls below $10^{-2}$ within tens of steps. The covariance-drift surrogate is noisier but shows an overall downward trend and remains in a low, stable regime at later stages. On WikiBigEdit (500K edits), similar patterns hold across Llama-3-8B-Instruct and Qwen2.5-7B-Instruct. These curves are consistent with Assumption 3.4 across architectures and editing scales.

### B.6.4. NON-I.I.D. DISTRIBUTION SHIFT

This part examines how the self-reinforcing stability loop of STABLEEDIT behaves when the editing distribution changes abruptly mid-stream, complementing the domain-mismatch warm-up tests in Section B.6.1 (which keep the target distribution fixed). To explicitly test robustness under an abrupt domain change, we construct a non-i.i.d. editing stream on Llama-3-8B-Instruct: warm-up on ZsRE, followed by 20K edits on ULTRAEDITBENCH, and then an abrupt switch to 400 edits from ZsRE. As shown in Table 14, the shifted STABLEEDIT variant is slightly below the no-shift version but still clearly outperforms ULTRAEDIT on 5 out of 6 metrics. In the ULTRAEDITBENCH segment, it improves Efficacy and Generalization by $+1.34$ and $+1.93$ points respectively over ULTRAEDIT, with only a $-1.20$ drop in Specificity. After the abrupt switch to ZsRE, it still exceeds ULTRAEDIT by substantial margins across all three metrics. These results indicate that STABLEEDIT remains robust even under substantial non-i.i.d. distribution shift, reinforcing the interpretation that the stability benefits extend beyond the stationary regime analyzed in the theory.

*Table 14.* Non-i.i.d. distribution shift experiment on Llama-3-8B-Instruct. The editing stream consists of warm-up on ZsRE, then 20K edits on ULTRAEDITBENCH, followed by an abrupt switch to 400 ZsRE edits.

| Method | ULTRAEDITBENCH (20K) | | | ZsRE (400) | | |
|---|---|---|---|---|---|---|
| | Eff. | Gen. | Spe. | Eff. | Gen. | Spe. |
| ULTRAEDIT | 85.70 | 81.28 | 68.73 | 79.12 | 69.04 | 41.66 |
| STABLEEDIT | 88.88 | 85.46 | 69.06 | 93.32 | 86.66 | 47.90 |
| Warmup-ZsRE & Dist. shift | 87.04 | 83.21 | 67.53 | 89.45 | 84.83 | 46.44 |

*Table 15.* Ablation on removing the $\|\tilde{\boldsymbol{h}}_{t,l}^{i}\|_2^2$ term from STABLEEDIT. The $\Delta$ row reports the signed change of *w.o.* $\|\tilde{\boldsymbol{h}}_{t,l}^{i}\|_2^2$ relative to STABLEEDIT; positive and negative changes are marked in red and blue, respectively.

| Method | ZsRE | | | FEVER | | | WikiBigEdit | | | ULTRAEDITBENCH | | |
|---|---|---|---|---|---|---|---|---|---|---|---|---|
| | Eff. | Gen. | Spe. | Eff. | Gen. | Spe. | Eff. | Gen. | Spe. | Eff. | Gen. | Spe. |
| | | | | | | **Mistral-7B-v0.3** | | | | | | |
| ULTRAEDIT | 85.30 | 80.80 | 47.38 | 97.87 | 96.09 | 84.29 | 76.00 | 70.15 | 46.09 | 83.71 | 77.30 | 67.26 |
| STABLEEDIT | 87.39 | 82.93 | 50.26 | 98.38 | 96.55 | 83.89 | 78.90 | 72.57 | 47.36 | 84.98 | 78.77 | 68.36 |
| *w.o.* $\|\tilde{h}\|_2^2$ | 87.29 | 82.63 | 51.05 | 98.27 | 96.54 | 83.83 | 78.82 | 72.55 | 48.65 | 84.67 | 78.09 | 68.69 |
| $\Delta$ | -0.10 | -0.30 | +0.79 | -0.11 | -0.01 | -0.06 | -0.08 | -0.02 | +1.29 | -0.31 | -0.68 | +0.33 |
| | | | | | | **Llama-3-8B-Instruct** | | | | | | |
| ULTRAEDIT | 90.07 | 87.36 | 49.51 | 95.39 | 91.93 | 67.14 | 79.60 | 73.49 | 48.51 | 85.70 | 81.28 | 68.73 |
| STABLEEDIT | 90.61 | 88.38 | 48.23 | 97.89 | 94.47 | 69.24 | 81.22 | 75.98 | 45.22 | 88.88 | 85.46 | 69.06 |
| *w.o.* $\|\tilde{h}\|_2^2$ | 91.56 | 89.82 | 48.80 | 98.17 | 94.50 | 68.67 | 80.69 | 75.39 | 46.11 | 89.00 | 85.41 | 69.51 |
| $\Delta$ | +0.95 | +1.44 | +0.57 | +0.28 | +0.03 | -0.57 | -0.53 | -0.59 | +0.89 | +0.12 | -0.05 | +0.45 |
| | | | | | | **GPT-J-6B** | | | | | | |
| ULTRAEDIT | 78.03 | 72.42 | 27.05 | 97.45 | 96.37 | 79.72 | 73.84 | 66.57 | 37.17 | 84.03 | 76.62 | 64.03 |
| STABLEEDIT | 82.12 | 78.39 | 31.52 | 98.35 | 96.99 | 79.85 | 75.10 | 68.13 | 44.04 | 83.41 | 76.55 | 67.17 |
| *w.o.* $\|\tilde{h}\|_2^2$ | 82.00 | 78.29 | 29.08 | 98.28 | 97.03 | 80.00 | 76.04 | 68.72 | 43.21 | 83.91 | 76.87 | 66.73 |
| $\Delta$ | -0.12 | -0.10 | -2.44 | -0.07 | +0.04 | +0.15 | +0.94 | +0.59 | -0.83 | +0.50 | +0.32 | -0.44 |

### B.6.5. ABLATION ON THE $\|\tilde{\boldsymbol{h}}\|_2^2$ TERM

This part isolates the contribution of the hidden-state scalar factor $\|\tilde{\boldsymbol{h}}_{t,l}^{i}\|_2^2$ in the target matrix $\boldsymbol{V}_{t,l} = -\gamma[\|\tilde{\boldsymbol{h}}_{t,l}^{1}\|_2^2\tilde{\boldsymbol{v}}_{t,l}^{1}, \ldots, \|\tilde{\boldsymbol{h}}_{t,l}^{n_t}\|_2^2\tilde{\boldsymbol{v}}_{t,l}^{n_t}]$, to confirm that the stability benefits of STABLEEDIT originate from gradient-side normalization (centering and whitening) rather than from this inherited hidden-side rescaling. The factor acts as a per-example step-size scaling and is inherited from prior LN-based editors; our theory in Theorem 3.8 does not rely on it. To verify this, we remove the $\|\tilde{h}\|_2^2$ term from STABLEEDIT and rely solely on the learning rate $\gamma$ to control the update scale. As shown in Table 15, the resulting variant stays very close to the original: across all 36 metric entries (3 models × 4 datasets × 3 metrics), the average absolute change is only 0.47 points, and the variant still outperforms ULTRAEDIT on 32 out of 36 entries. This confirms that the normalization of the hidden state is not essential to the gains of STABLEEDIT, and that the primary stability benefits come from gradient-side normalization, consistent with the mechanism analyzed in Theorem 3.8.

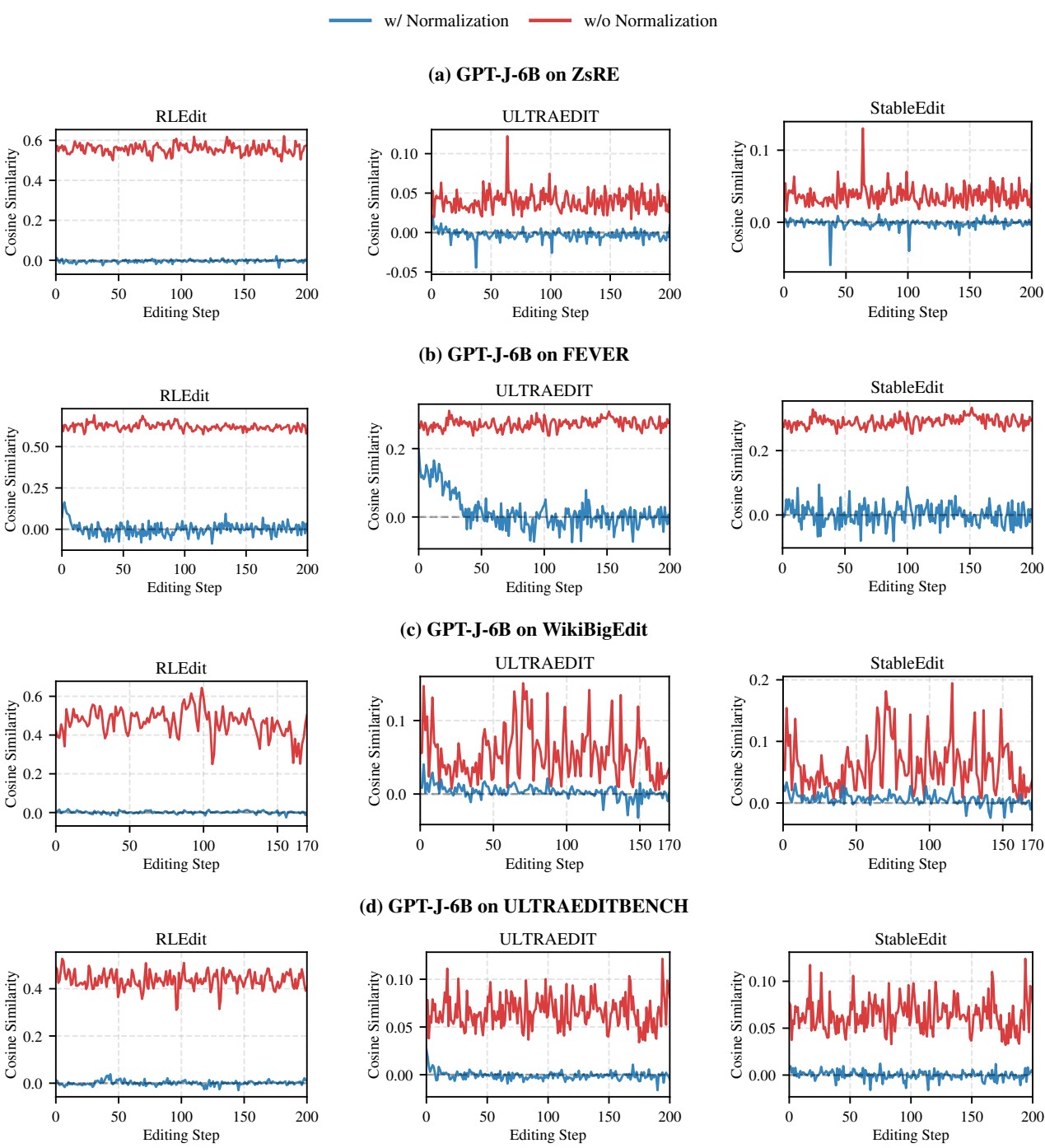

*Figure 11.* Update orthogonality with and without LN. We plot the cosine similarity between adjacent parameter increments $\mathbf{\Delta}_{t,l}$ and $\mathbf{\Delta}_{t-1,l}$ across the edit stream for three editors on GPT-J-6B. With LN enabled (blue), cosine similarities remain close to zero throughout the editing process, indicating weakly correlated (approximately orthogonal) update directions. Disabling LN (red) causes similarities to deviate from zero. These observations empirically validate Theorem 3.8(c), which proves that LN combined with ridge regression enforces asymptotic orthogonality.

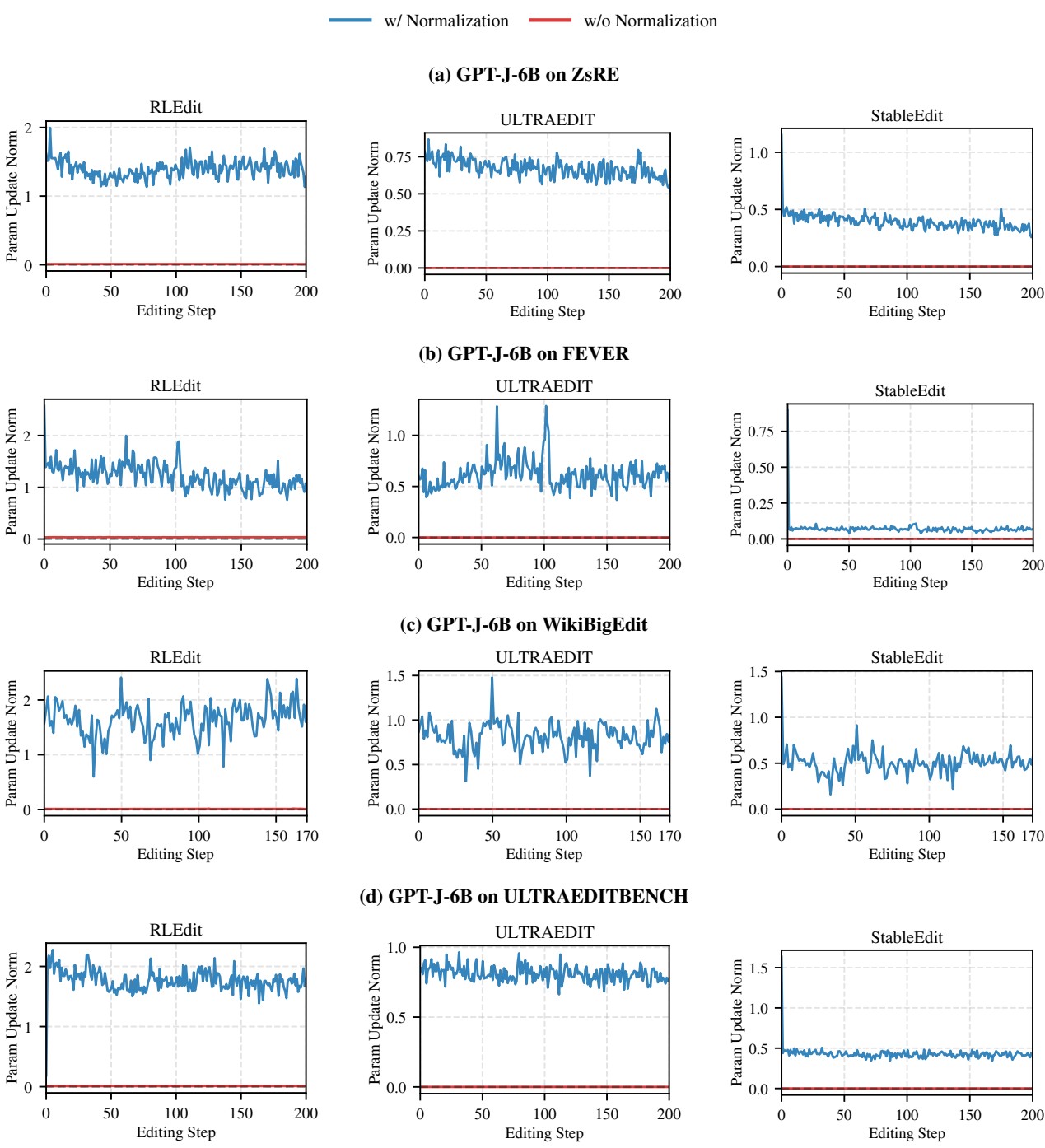

*Figure 12.* Update magnitudes with and without LN. We plot the Frobenius norm of the per-step parameter increment $\|\mathbf{\Delta}_{t,l}\|_F$ across the edit stream for three editors on GPT-J-6B. With LN enabled (blue), update magnitudes remain bounded and non-trivial, supporting effective edits throughout the sequence. Disabling LN (red) causes update norms to collapse toward zero, indicating a *vanishing-update* failure mode that leads to under-fitting. These trends are consistent with Theorem 3.8(a) and (b), and align with our mechanism: whitening boosts updates in weak eigen-directions while damping high-variance components, whereas baselines often fail to make progress along minor directions within a single step.

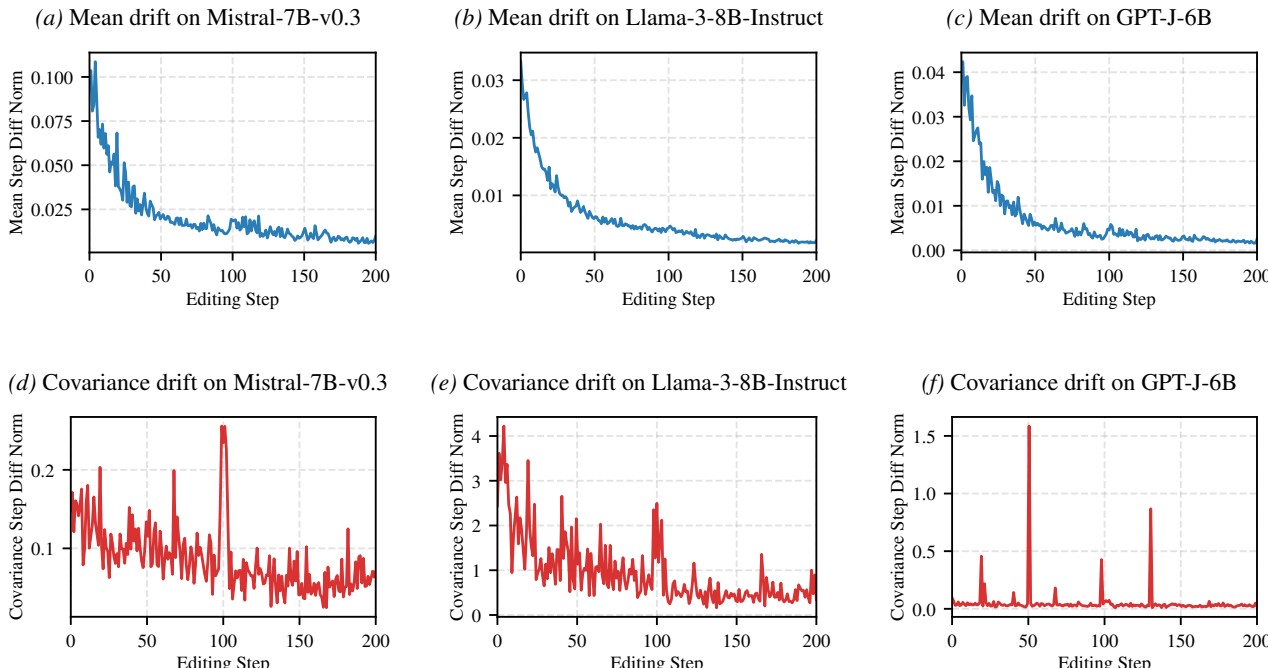

*Figure 13.* Mean drift and covariance drift across editing steps on FEVER under the 20K-edit setting for Mistral-7B-v0.3, GPT-J-6B, and Llama-3-8B-Instruct. At each editing step $t$, we plot the Frobenius norm of the change in the corresponding running statistic between two consecutive steps, i.e., $\|\hat{\boldsymbol{\mu}}_t - \hat{\boldsymbol{\mu}}_{t-1}\|_F$ for the mean drift and $\|\hat{\boldsymbol{\Sigma}}_t - \hat{\boldsymbol{\Sigma}}_{t-1}\|_F$ for the covariance drift.

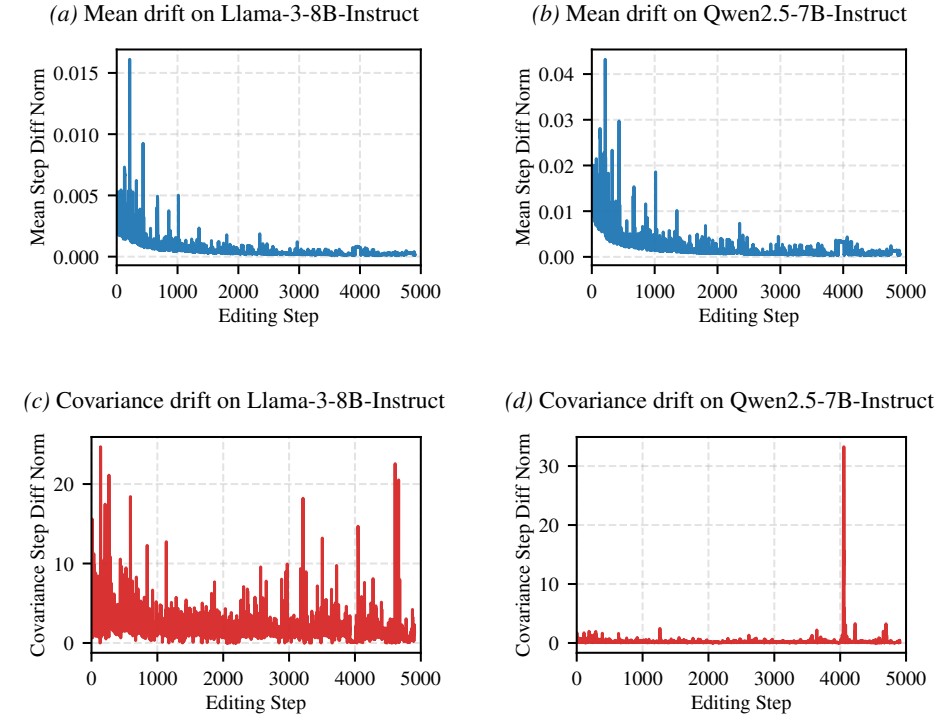

*Figure 14.* Mean drift and covariance drift across editing steps on WikiBigEdit under the 500K-edit setting for Llama-3-8B-Instruct and Qwen2.5-7B-Instruct. At each editing step $t$, we plot the Frobenius norm of the change in the corresponding running statistic between two consecutive steps, i.e., $\|\hat{\boldsymbol{\mu}}_t - \hat{\boldsymbol{\mu}}_{t-1}\|_F$ for the mean drift and $\|\hat{\boldsymbol{\Sigma}}_t - \hat{\boldsymbol{\Sigma}}_{t-1}\|_F$ for the covariance drift.

## C. Preliminaries and Assumptions

In this section, we restate the preliminaries and assumptions used in the main analysis with detailed discussions.

**Lifelong editing process.** We consider a sequential editing stream indexed by steps $t \geq 1$ and editable layers $l$. At step $t$, a batch of $n_t$ editing requests $\{e_t^i\}_{i=1}^{n_t}$ is drawn i.i.d. from an edit distribution $\mathcal{D}$. Let $\boldsymbol{W}_{t,l} \in \mathbb{R}^{d \times d_h}$ denote the editable parameters at layer $l$ after step $t$, with update $\boldsymbol{W}_{t,l} = \boldsymbol{W}_{t-1,l} + \boldsymbol{\Delta}_{t,l}$. Bold symbols denote vectors/matrices; $\|\cdot\|_2$ denotes the Euclidean norm (vectors) and spectral norm (matrices); $\|\cdot\|_F$ is the Frobenius norm.

**Value gradients and their non-stationary moments.** For edit $i$ at layer $l$, let $\boldsymbol{h}_{t,l}^i \in \mathbb{R}^{d_h}$ denote the input hidden state at the target token position, and $\boldsymbol{u}_{t,l}^i := \boldsymbol{W}_{t-1,l}\boldsymbol{h}_{t,l}^i \in \mathbb{R}^d$ the module output. Given loss $\ell(\cdot)$, define the *value gradient*

$$\boldsymbol{v}_{t,l}^i := \nabla_{\boldsymbol{u}_{t,l}^i} \ell(\boldsymbol{e}_t^i; \boldsymbol{W}_{t-1,l}) \triangleq \boldsymbol{g}(\boldsymbol{e}_t^i; \boldsymbol{W}_{t-1,l}) \in \mathbb{R}^d. \tag{7}$$

As parameters evolve, the induced distribution of $\boldsymbol{g}(\boldsymbol{e}; \boldsymbol{W}_{t-1,l})$ changes with $t$. We summarize it by the population mean and covariance:

$$\boldsymbol{\mu}_{t,l} := \mathbb{E}_{\boldsymbol{e} \sim \mathcal{D}}[\boldsymbol{g}(\boldsymbol{e}; \boldsymbol{W}_{t-1,l})], \qquad \boldsymbol{\Sigma}_{t,l} := \mathrm{Cov}_{\boldsymbol{e} \sim \mathcal{D}}[\boldsymbol{g}(\boldsymbol{e}; \boldsymbol{W}_{t-1,l})]. \tag{8}$$

**Estimation error metrics.** For analyzing warm-start effects, we split the process into a previous phase of $r \geq 1$ steps and a current phase of $t \geq 1$ steps (global step $r + t$), and (for simplicity) take constant batch size $n_t = n$ in the current phase. Let $\boldsymbol{m}_{t,l}^{(\mathrm{cur})}$ and $\widehat{\boldsymbol{\Sigma}}_{t,l}^{(\mathrm{cur})}$ be the current-phase point estimators with initialization $\boldsymbol{m}_{0,l}^{(\mathrm{cur})} = \boldsymbol{m}_{r,l}^{(\mathrm{pre})}$ and $\widehat{\boldsymbol{\Sigma}}_{0,l}^{(\mathrm{cur})} = \widehat{\boldsymbol{\Sigma}}_{r,l}^{(\mathrm{pre})}$. We measure mean estimation by MSE and covariance estimation by a spectral-norm error:

$$\mathrm{MSE}_{t,l}^{(\boldsymbol{\mu},\mathrm{cur})} := \mathbb{E}\big\|\boldsymbol{m}_{t,l}^{(\mathrm{cur})} - \boldsymbol{\mu}_{t,l}^{(\mathrm{cur})}\big\|_2^2, \qquad E_{t,l}^{(\boldsymbol{\Sigma},\mathrm{cur})} := \mathbb{E}\big\|\widehat{\boldsymbol{\Sigma}}_{t,l}^{(\mathrm{cur})} - \boldsymbol{\Sigma}_{t,l}^{(\mathrm{cur})}\big\|_2.$$

**LN whitening and ridge-regression update.** Given $(\hat{\boldsymbol{\mu}}_{t,l}, \hat{\boldsymbol{\Sigma}}_{t,l})$, LN centers and whitens gradients:

$$\tilde{\boldsymbol{v}}_{t,l}^i := \widehat{\boldsymbol{\Sigma}}_{t,l}^{-1/2}\big(\boldsymbol{v}_{t,l}^i - \hat{\boldsymbol{\mu}}_{t,l}\big). \tag{9}$$

Let $\boldsymbol{H}_{t,l} = [\boldsymbol{h}_{t,l}^1, \ldots, \boldsymbol{h}_{t,l}^{n_t}] \in \mathbb{R}^{d_h \times n_t}$. Define the target matrix

$$\boldsymbol{V}_{t,l} := -\gamma\big[\|\tilde{\boldsymbol{h}}_{t,l}^1\|_2^2 \tilde{\boldsymbol{v}}_{t,l}^1, \ldots, \|\tilde{\boldsymbol{h}}_{t,l}^{n_t}\|_2^2 \tilde{\boldsymbol{v}}_{t,l}^{n_t}\big] \in \mathbb{R}^{d \times n_t},$$

where $\tilde{\boldsymbol{h}}_{t,l}^i$ denotes a per-dimension standardized $\boldsymbol{h}_{t,l}^i$. The update $\boldsymbol{\Delta}_{t,l}$ is the ridge-regression solution

$$\boldsymbol{\Delta}_{t,l} = \arg\min_{\boldsymbol{\Delta}} \|\boldsymbol{\Delta}\boldsymbol{H}_{t,l} - \boldsymbol{V}_{t,l}\|_F^2 + \lambda\|\boldsymbol{\Delta}\|_F^2, \tag{10}$$

with closed form

$$\boldsymbol{\Delta}_{t,l} = \boldsymbol{V}_{t,l}\boldsymbol{H}_{t,l}^\top(\boldsymbol{H}_{t,l}\boldsymbol{H}_{t,l}^\top + \lambda\boldsymbol{I})^{-1} = -\gamma\sum_{i=1}^{n_t} \tilde{\boldsymbol{v}}_{t,l}^i \boldsymbol{\phi}_{t,l}^i, \tag{11}$$

where the *projection factor*

$$\boldsymbol{\phi}_{t,l}^i := \|\tilde{\boldsymbol{h}}_{t,l}^i\|_2^2 (\boldsymbol{h}_{t,l}^i)^\top \Big(\sum_{j=1}^{n_t} \boldsymbol{h}_{t,l}^j (\boldsymbol{h}_{t,l}^j)^\top + \lambda\boldsymbol{I}\Big)^{-1} \in \mathbb{R}^{1 \times d_h}.$$

**Decomposition of gradients and update components.** We decompose each value gradient into a population mean plus a zero-mean residual:

$$\boldsymbol{v}_{t,l}^i = \boldsymbol{\mu}_{t,l} + \boldsymbol{\zeta}_{t,l}^i, \qquad \mathbb{E}[\boldsymbol{\zeta}_{t,l}^i] = \boldsymbol{0}.$$

Substituting into (11) yields $\boldsymbol{\Delta}_{t,l} = \boldsymbol{\Delta}_{t,l}^{\mathrm{spec}} + \boldsymbol{\Delta}_{t,l}^{\mathrm{bias}}$ with

$$\boldsymbol{\Delta}_{t,l}^{\mathrm{spec}} := -\gamma\sum_{i=1}^{n_t} \widehat{\boldsymbol{\Sigma}}_{t,l}^{-1/2} \boldsymbol{\zeta}_{t,l}^i \boldsymbol{\phi}_{t,l}^i, \qquad \boldsymbol{\Delta}_{t,l}^{\mathrm{bias}} := -\gamma\,\widehat{\boldsymbol{\Sigma}}_{t,l}^{-1/2}(\boldsymbol{\mu}_{t,l} - \boldsymbol{m}_{t,l})\sum_{i=1}^{n_t} \boldsymbol{\phi}_{t,l}^i.$$

We also use $\tilde{\boldsymbol{\zeta}}_{t,l}^i := \widehat{\boldsymbol{\Sigma}}_{t,l}^{-1/2} \boldsymbol{\zeta}_{t,l}^i$ for whitened residual directions.

And our assumptions are

**Assumption 3.1\*.** (Regularity Conditions). *For all editable layers $l$ and steps $t$, there exist finite constants $L, M, \sigma_- > 0, \sigma_+ > 0$ such that*

*(a)* **Smoothness:** *$g(e; W)$ is $L$-Lipschitz continuous w.r.t. $W$, and $\sup_W \mathbb{E}[\|g(e; W)\|_2^2] \leq M$.*

*(b)* **Spectrum Bounds:** *The true covariance $\Sigma_{t,l}$ is non-degenerate: $\sigma_- I \preceq \Sigma_{t,l} \preceq \sigma_+ I$.*

*Remark* C.1. Assumption 3.1(a) is a standard smoothness condition in optimization analysis. It essentially states that small changes in the model parameters do not cause abrupt or unbounded jumps in the gradient values, and that the gradients themselves do not explode. Assumption 3.1(b) assume that the gradient covariance matrix $\Sigma_{t,l}$ is symmetric positive definite with bounded spectrum. The upper bound $\sigma_+$ follows from the finite gradient moments (Assumption 3.1(a)). The lower bound $\sigma_-$ is a standard non-degeneracy condition, ensuring the Gaussian likelihood is well-defined and the whitening operation is numerically stable.

**Assumption 3.4\*.** (Trackable Drift Regime). *For constants $\epsilon, \epsilon' > 0$:*

$$D_j^{(\mu, cur)} = O((r + j)^{-(1+\epsilon/2)}),$$
$$\Delta_j^{(\Sigma, cur)} = O((r + j)^{-(1+\epsilon')}).$$

*where $D_j^{(\mu, cur)}$ and $\Delta_j^{(\Sigma, cur)}$ upper-bound $\|\mu_{j,l}^{(cur)} - \mu_{j-1,l}^{(cur)}\|_2$ and $\|\Sigma_{j,l}^{(cur)} - \Sigma_{j-1,l}^{(cur)}\|_2$, respectively.*

**Assumption 3.7\*.** *For every step $t$ and layer $l$, the projection factor $\phi_{t,l}^i$ and the inverse covariance estimator $\widehat{\Sigma}_{t,l}^{-1}$ possess finite fourth moments:*

$$\mathbb{E}[\|\phi_{t,l}^i\|_2^4] \leq C_\phi < \infty, \quad \mathbb{E}[\|\widehat{\Sigma}_{t,l}^{-1}\|_2^4] \leq K_\Sigma < \infty.$$

*Remark* C.2. Assumption 3.7 is well-justified within the context of LLMs. First, input hidden states $h_{t,l}^i$ are typically processed via LayerNorm and exhibit strong concentration properties in deep networks; thus, the higher-order moments of the resulting projection factor $\phi_{t,l}^i$ are naturally bounded. Second, the stability of the inverse covariance estimator $\widehat{\Sigma}_{t,l}^{-1}$ is physically ensured by the Bayesian prior, which prevents the explosion of the spectral radius. Consequently, the boundedness of the fourth moments is a standard regularity requirement to ensure the convergence of sequential estimators.

# D. Theoretical Proofs and Derivations

This section provides the detailed mathematical proofs for the lemmas and theorems presented in the main paper.

## D.1. Proof of Lemma D.1

**Lemma D.1** (Bounds on Distributional Drift). *Under Assumption 3.1(a), the expected drift in the gradient mean (Euclidean norm $\| \cdot \|_2$) and covariance (spectral norm $\| \cdot \|_2$) at layer $l$ between consecutive steps $t-1$ and $t$ satisfies*

$$\|\mu_{t,l} - \mu_{t-1,l}\|_2 \leq L\sqrt{\mathbb{E}[\|\Delta_{t-1,l}\|_F^2]}, \tag{12}$$

$$\|\Sigma_{t,l} - \Sigma_{t-1,l}\|_2 \leq 4L\sqrt{M}\sqrt{\mathbb{E}[\|\Delta_{t-1,l}\|_F^2]}. \tag{13}$$

*Proof.* For notational convenience, let $g_{t,l}(e) := g(e; W_{t-1,l})$ denote the gradient at step $t$ given input $e$.

**Bound for the Mean Drift.** Let $e_t$ and $e_{t-1}$ denote independent samples drawn from $\mathcal{D}$ at steps $t$ and $t-1$, respectively. By definition, $\mu_{t,l} = \mathbb{E}_{e_t \sim \mathcal{D}}[g_{t,l}(e_t)]$ and $\mu_{t-1,l} = \mathbb{E}_{e_{t-1} \sim \mathcal{D}}[g_{t-1,l}(e_{t-1})]$. To isolate the parameter drift from the data sampling, we introduce a coupling term:

$$\|\mu_{t,l} - \mu_{t-1,l}\|_2 = \left\|\mathbb{E}_{e_t, e_{t-1}}[g_{t,l}(e_t) - g_{t-1,l}(e_{t-1})]\right\|_2$$
$$= \left\|\mathbb{E}_{e_t, e_{t-1}}[\underbrace{g_{t,l}(e_t) - g_{t-1,l}(e_t)}_{\text{Parameter Drift}} + \underbrace{g_{t-1,l}(e_t) - g_{t-1,l}(e_{t-1})}_{\text{Sampling Difference}}]\right\|_2.$$

Since $e_t, e_{t-1} \overset{\text{i.i.d}}{\sim} \mathcal{D}$, the expectation of the sampling difference vanishes:

$$\mathbb{E}_{e_t, e_{t-1}}[g_{t-1,l}(e_t) - g_{t-1,l}(e_{t-1})] = \mathbb{E}_{e_t}[g_{t-1,l}(e_t)] - \mathbb{E}_{e_{t-1}}[g_{t-1,l}(e_{t-1})] = \mu_{t-1,l} - \mu_{t-1,l} = \mathbf{0}.$$

Thus, the drift depends solely on the parameter change. Applying Jensen's inequality:

$$\|\boldsymbol{\mu}_{t,l} - \boldsymbol{\mu}_{t-1,l}\|_2 = \left\|\mathbb{E}_{\boldsymbol{e}\sim\mathcal{D}}[\boldsymbol{g}_{t,l}(\boldsymbol{e}) - \boldsymbol{g}_{t-1,l}(\boldsymbol{e})]\right\|_2$$
$$\leq \mathbb{E}_{\boldsymbol{e}\sim\mathcal{D}}\left[\|\boldsymbol{g}_{t,l}(\boldsymbol{e}) - \boldsymbol{g}_{t-1,l}(\boldsymbol{e})\|_2\right].$$

Invoking the Lipschitz continuity (Assumption 3.1(a)) and Cauchy-Schwarz:

$$\|\boldsymbol{\mu}_{t,l} - \boldsymbol{\mu}_{t-1,l}\|_2 \leq L\,\mathbb{E}[\|\boldsymbol{\Delta}_{t-1,l}\|_F] \leq L\sqrt{\mathbb{E}[\|\boldsymbol{\Delta}_{t-1,l}\|_F^2]}.$$

This confirms (12).

**Bound for the Covariance Drift.** Recall the definition $\boldsymbol{\Sigma}_{t,l} = \mathbb{E}_{\boldsymbol{e}_t\sim\mathcal{D}}[\boldsymbol{g}_{t,l}(\boldsymbol{e}_t)\boldsymbol{g}_{t,l}(\boldsymbol{e}_t)^\top] - \boldsymbol{\mu}_{t,l}\boldsymbol{\mu}_{t,l}^\top$. Similar to the mean drift analysis, we explicitly account for the distinct samples $\boldsymbol{e}_t$ and $\boldsymbol{e}_{t-1}$ used at each step. By the triangle inequality, the spectral norm of the drift is bounded by:

$$\|\boldsymbol{\Sigma}_{t,l} - \boldsymbol{\Sigma}_{t-1,l}\|_2 \leq \underbrace{\left\|\mathbb{E}_{\boldsymbol{e}_t}[\boldsymbol{g}_{t,l}(\boldsymbol{e}_t)\boldsymbol{g}_{t,l}(\boldsymbol{e}_t)^\top] - \mathbb{E}_{\boldsymbol{e}_{t-1}}[\boldsymbol{g}_{t-1,l}(\boldsymbol{e}_{t-1})\boldsymbol{g}_{t-1,l}(\boldsymbol{e}_{t-1})^\top]\right\|_2}_{\text{(I)}} + \underbrace{\left\|\boldsymbol{\mu}_{t,l}\boldsymbol{\mu}_{t,l}^\top - \boldsymbol{\mu}_{t-1,l}\boldsymbol{\mu}_{t-1,l}^\top\right\|_2}_{\text{(II)}}.$$

For Term (I), we apply the same coupling argument: introducing the cross-term $\boldsymbol{g}_{t-1,l}(\boldsymbol{e}_t)\boldsymbol{g}_{t-1,l}(\boldsymbol{e}_t)^\top$ allows us to cancel the sampling variance (since $\boldsymbol{e}_t, \boldsymbol{e}_{t-1} \overset{\text{i.i.d}}{\sim} \mathcal{D}$) and isolate the parameter drift. Thus, Term (I) reduces to bounding the drift over a generic sample $\boldsymbol{e} \sim \mathcal{D}$:

$$\text{(I)} = \left\|\mathbb{E}_{\boldsymbol{e}\sim\mathcal{D}}[\boldsymbol{g}_{t,l}(\boldsymbol{e})\boldsymbol{g}_{t,l}(\boldsymbol{e})^\top - \boldsymbol{g}_{t-1,l}(\boldsymbol{e})\boldsymbol{g}_{t-1,l}(\boldsymbol{e})^\top]\right\|_2.$$

We utilize the matrix norm identity $\|\boldsymbol{a}\boldsymbol{a}^\top - \boldsymbol{b}\boldsymbol{b}^\top\|_2 \leq (\|\boldsymbol{a}\|_2 + \|\boldsymbol{b}\|_2)\|\boldsymbol{a} - \boldsymbol{b}\|_2$.

*Bounding Term (I):* Using Jensen's inequality to move the norm inside the expectation, followed by the Cauchy-Schwarz inequality:

$$\text{(I)} \leq \mathbb{E}\left[(\|\boldsymbol{g}_{t,l}(\boldsymbol{e})\|_2 + \|\boldsymbol{g}_{t-1,l}(\boldsymbol{e})\|_2)\|\boldsymbol{g}_{t,l}(\boldsymbol{e}) - \boldsymbol{g}_{t-1,l}(\boldsymbol{e})\|_2\right]$$
$$\leq \sqrt{\mathbb{E}\left[(\|\boldsymbol{g}_{t,l}(\boldsymbol{e})\|_2 + \|\boldsymbol{g}_{t-1,l}(\boldsymbol{e})\|_2)^2\right]} \cdot \sqrt{\mathbb{E}\left[\|\boldsymbol{g}_{t,l}(\boldsymbol{e}) - \boldsymbol{g}_{t-1,l}(\boldsymbol{e})\|_2^2\right]}.$$

Using the inequality $(a+b)^2 \leq 2a^2 + 2b^2$ and Assumption 3.1(a):

$$\text{(I)} \leq \sqrt{2\mathbb{E}[\|\boldsymbol{g}_{t,l}(\boldsymbol{e})\|_2^2] + 2\mathbb{E}[\|\boldsymbol{g}_{t-1,l}(\boldsymbol{e})\|_2^2]} \cdot \sqrt{\mathbb{E}[L^2\|\boldsymbol{\Delta}_{t-1,l}\|_F^2]}$$
$$\leq \sqrt{4M} \cdot L\sqrt{\mathbb{E}[\|\boldsymbol{\Delta}_{t-1,l}\|_F^2]} = 2\sqrt{M}L\sqrt{\mathbb{E}[\|\boldsymbol{\Delta}_{t-1,l}\|_F^2]}.$$

*Bounding Term (II):* Similarly, applying the norm identity to the mean vectors:

$$\text{(II)} \leq (\|\boldsymbol{\mu}_{t,l}\|_2 + \|\boldsymbol{\mu}_{t-1,l}\|_2)\|\boldsymbol{\mu}_{t,l} - \boldsymbol{\mu}_{t-1,l}\|_2.$$

By Jensen's inequality, $\|\boldsymbol{\mu}_{t,l}\|_2 = \|\mathbb{E}[\boldsymbol{g}_{t,l}(\boldsymbol{e})]\|_2 \leq \sqrt{\mathbb{E}[\|\boldsymbol{g}_{t,l}(\boldsymbol{e})\|_2^2]} \leq \sqrt{M}$. Combining this with the mean drift bound derived in Eq. (12):

$$\text{(II)} \leq 2\sqrt{M} \cdot (L\sqrt{\mathbb{E}[\|\boldsymbol{\Delta}_{t-1,l}\|_F^2]}).$$

Summing terms (I) and (II) yields the final bound:

$$\|\boldsymbol{\Sigma}_{t,l} - \boldsymbol{\Sigma}_{t-1,l}\|_2 \leq 4\sqrt{M}L\sqrt{\mathbb{E}[\|\boldsymbol{\Delta}_{t-1,l}\|_F^2]}.$$

This completes the proof of (13).

$\square$

*Remark* D.2. Lemma D.1 establishes quantitative bounds on the across-step drift of the gradient statistics. This conclusion is pivotal for our **Sequential Bayesian Estimation** framework. By guaranteeing that $\boldsymbol{\mu}$ and $\boldsymbol{\Sigma}$ evolve smoothly within a **bounded-drift regime**, it validates the strategy of using the posterior from step $t-1$ as an informative prior for step $t$. This recursive "prior-posterior" mechanism enables efficient and robust online estimation, which is essential for accurate gradient whitening in streaming settings.

## D.2. Proof of Theorem 3.2

**Theorem 3.2\***. *Suppose the prior over $(\boldsymbol{\mu}_{t,l}, \boldsymbol{\Sigma}_{t,l})$ follows a NIW distribution derived from step $t-1$,*

$$p(\boldsymbol{\mu}_{t,l}, \boldsymbol{\Sigma}_{t,l}) = NIW(\boldsymbol{m}_{t-1,l}, \kappa_{t-1,l}, \boldsymbol{\Psi}_{t-1,l}, \nu_{t-1,l}),$$

*and the likelihood is conditionally i.i.d. Gaussian,*

$$p(\mathcal{D}_{t,l} \mid \boldsymbol{\mu}_{t,l}, \boldsymbol{\Sigma}_{t,l}) = \prod_{i=1}^{n_t} \mathcal{N}_d(\boldsymbol{v}_{t,l}^i \mid \boldsymbol{\mu}_{t,l}, \boldsymbol{\Sigma}_{t,l}).$$

*Then the posterior $p(\boldsymbol{\mu}_{t,l}, \boldsymbol{\Sigma}_{t,l} \mid \mathcal{D}_{t,l})$ is also NIW, namely $NIW(\boldsymbol{m}_{t,l}, \kappa_{t,l}, \boldsymbol{\Psi}_{t,l}, \nu_{t,l})$, with updated hyperparameters:*

$$\kappa_{t,l} = \kappa_{t-1,l} + n_t, \tag{14}$$

$$\nu_{t,l} = \nu_{t-1,l} + n_t, \tag{15}$$

$$\boldsymbol{m}_{t,l} = \frac{\kappa_{t-1,l}\boldsymbol{m}_{t-1,l} + n_t\bar{\boldsymbol{v}}_{t,l}}{\kappa_{t,l}}, \tag{16}$$

$$\boldsymbol{\Psi}_{t,l} = \boldsymbol{\Psi}_{t-1,l} + \boldsymbol{S}_{t,l} + \frac{\kappa_{t-1,l}n_t}{\kappa_{t,l}}\boldsymbol{Q}_{t,l}, \tag{17}$$

*where $\boldsymbol{Q}_{t,l} = (\bar{\boldsymbol{v}}_{t,l} - \boldsymbol{m}_{t-1,l})(\bar{\boldsymbol{v}}_{t,l} - \boldsymbol{m}_{t-1,l})^{\top}$.*

*Proof.* The proof proceeds by applying Bayes' theorem, where the posterior is proportional to the product of the likelihood and the prior, $p(\boldsymbol{\mu}_{t,l}, \boldsymbol{\Sigma}_{t,l} \mid \{\boldsymbol{v}_{t,l}^i\}) \propto p(\{\boldsymbol{v}_{t,l}^i\} \mid \boldsymbol{\mu}_{t,l}, \boldsymbol{\Sigma}_{t,l})p(\boldsymbol{\mu}_{t,l}, \boldsymbol{\Sigma}_{t,l})$.

**Likelihood.** The likelihood for $n_t$ i.i.d. $d$-dimensional Gaussian observations $\{\boldsymbol{v}_{t,l}^i\}_{i=1}^{n_t}$ is

$$
\begin{aligned}
p(\{\boldsymbol{v}_{t,l}^i\} \mid \boldsymbol{\mu}_{t,l}, \boldsymbol{\Sigma}_{t,l}) &= \prod_{i=1}^{n_t}(2\pi)^{-d/2}|\boldsymbol{\Sigma}_{t,l}|^{-1/2}\exp\left(-\frac{1}{2}(\boldsymbol{v}_{t,l}^i - \boldsymbol{\mu}_{t,l})^{\top}\boldsymbol{\Sigma}_{t,l}^{-1}(\boldsymbol{v}_{t,l}^i - \boldsymbol{\mu}_{t,l})\right) \\
&= (2\pi)^{-n_t d/2}|\boldsymbol{\Sigma}_{t,l}|^{-n_t/2}\exp\left(-\frac{1}{2}\sum_{i=1}^{n_t}(\boldsymbol{v}_{t,l}^i - \boldsymbol{\mu}_{t,l})^{\top}\boldsymbol{\Sigma}_{t,l}^{-1}(\boldsymbol{v}_{t,l}^i - \boldsymbol{\mu}_{t,l})\right) \\
&= (2\pi)^{-n_t d/2}|\boldsymbol{\Sigma}_{t,l}|^{-n_t/2}\exp\left(-\frac{1}{2}\operatorname{tr}\left(\boldsymbol{\Sigma}_{t,l}^{-1}\sum_{i=1}^{n_t}(\boldsymbol{v}_{t,l}^i - \boldsymbol{\mu}_{t,l})(\boldsymbol{v}_{t,l}^i - \boldsymbol{\mu}_{t,l})^{\top}\right)\right).
\end{aligned}
\tag{18}
$$

**Prior.** The Normal-Inverse-Wishart prior, $\text{NIW}(\boldsymbol{m}_{t-1,l}, \kappa_{t-1,l}, \boldsymbol{\Psi}_{t-1,l}, \nu_{t-1,l})$, is a conjugate prior for the Gaussian parameters. Its density function is the product of an Inverse-Wishart distribution for $\boldsymbol{\Sigma}_{t,l}$ and a conditional Normal distribution for $\boldsymbol{\mu}_{t,l}$:

$$
\begin{aligned}
p(\boldsymbol{\mu}_{t,l}, \boldsymbol{\Sigma}_{t,l}) &= p(\boldsymbol{\Sigma}_{t,l})p(\boldsymbol{\mu}_{t,l} \mid \boldsymbol{\Sigma}_{t,l}) \\
&\propto |\boldsymbol{\Sigma}_{t,l}|^{-(\nu_{t-1,l}+d+1)/2}\exp\left(-\frac{1}{2}\operatorname{tr}(\boldsymbol{\Psi}_{t-1,l}\boldsymbol{\Sigma}_{t,l}^{-1})\right) \\
&\quad \times |\boldsymbol{\Sigma}_{t,l}|^{-1/2}\exp\left(-\frac{\kappa_{t-1,l}}{2}(\boldsymbol{\mu}_{t,l} - \boldsymbol{m}_{t-1,l})^{\top}\boldsymbol{\Sigma}_{t,l}^{-1}(\boldsymbol{\mu}_{t,l} - \boldsymbol{m}_{t-1,l})\right).
\end{aligned}
\tag{19}
$$

**Posterior.** Combining the likelihood Equation (18) and prior Equation (19), the unnormalized posterior is proportional to

$$
\begin{aligned}
p(\boldsymbol{\mu}_{t,l}, \boldsymbol{\Sigma}_{t,l} \mid \{\boldsymbol{v}_{t,l}^i\}) &\propto p(\{\boldsymbol{v}_{t,l}^i\} \mid \boldsymbol{\mu}_{t,l}, \boldsymbol{\Sigma}_{t,l})p(\boldsymbol{\mu}_{t,l}, \boldsymbol{\Sigma}_{t,l}) \\
&\propto |\boldsymbol{\Sigma}_{t,l}|^{-n_t/2}|\boldsymbol{\Sigma}_{t,l}|^{-(\nu_{t-1,l}+d+2)/2}\exp\left(-\frac{1}{2}\operatorname{tr}\left[\boldsymbol{\Sigma}_{t,l}^{-1}\left(\boldsymbol{\Psi}_{t-1,l}\right.\right.\right. \\
&\quad \left.\left.\left. + \sum_{i=1}^{n_t}(\boldsymbol{v}_{t,l}^i - \boldsymbol{\mu}_{t,l})(\boldsymbol{v}_{t,l}^i - \boldsymbol{\mu}_{t,l})^{\top} + \kappa_{t-1,l}(\boldsymbol{\mu}_{t,l} - \boldsymbol{m}_{t-1,l})(\boldsymbol{\mu}_{t,l} - \boldsymbol{m}_{t-1,l})^{\top}\right)\right]\right).
\end{aligned}
$$

We now simplify the terms inside the exponent's trace. First, we expand the data sum of squares using the sample mean $\bar{\boldsymbol{v}}_{t,l}$:

$$\sum_{i=1}^{n_t}(\boldsymbol{v}_{t,l}^i - \boldsymbol{\mu}_{t,l})(\boldsymbol{v}_{t,l}^i - \boldsymbol{\mu}_{t,l})^\top = \boldsymbol{S}_{t,l} + n_t(\bar{\boldsymbol{v}}_{t,l} - \boldsymbol{\mu}_{t,l})(\bar{\boldsymbol{v}}_{t,l} - \boldsymbol{\mu}_{t,l})^\top,$$

where $\boldsymbol{S}_{t,l}$ is the sample scatter matrix. Substituting this back, the terms involving $\boldsymbol{\mu}_{t,l}$ are

$$n_t(\bar{\boldsymbol{v}}_{t,l} - \boldsymbol{\mu}_{t,l})(\bar{\boldsymbol{v}}_{t,l} - \boldsymbol{\mu}_{t,l})^\top + \kappa_{t-1,l}(\boldsymbol{\mu}_{t,l} - \boldsymbol{m}_{t-1,l})(\boldsymbol{\mu}_{t,l} - \boldsymbol{m}_{t-1,l})^\top. \tag{20}$$

By completing the square with respect to $\boldsymbol{\mu}_{t,l}$ and defining the posterior hyperparameters as follows:

$$\kappa_{t,l} = \kappa_{t-1,l} + n_t, \tag{21}$$

$$\boldsymbol{m}_{t,l} = \frac{\kappa_{t-1,l}\boldsymbol{m}_{t-1,l} + n_t\bar{\boldsymbol{v}}_{t,l}}{\kappa_{t,l}}, \tag{22}$$

the expression in Equation (20) can be rewritten as

$$\kappa_{t,l}(\boldsymbol{\mu}_{t,l} - \boldsymbol{m}_{t,l})(\boldsymbol{\mu}_{t,l} - \boldsymbol{m}_{t,l})^\top + \frac{\kappa_{t-1,l}n_t}{\kappa_{t,l}}(\bar{\boldsymbol{v}}_{t,l} - \boldsymbol{m}_{t-1,l})(\bar{\boldsymbol{v}}_{t,l} - \boldsymbol{m}_{t-1,l})^\top.$$

We group all terms not involving $\boldsymbol{\mu}_{t,l}$ to define the remaining posterior hyperparameters:

$$\nu_{t,l} = \nu_{t-1,l} + n_t, \tag{23}$$

$$\boldsymbol{\Psi}_{t,l} = \boldsymbol{\Psi}_{t-1,l} + \boldsymbol{S}_{t,l} + \frac{\kappa_{t-1,l}n_t}{\kappa_{t,l}}(\bar{\boldsymbol{v}}_{t,l} - \boldsymbol{m}_{t-1,l})(\bar{\boldsymbol{v}}_{t,l} - \boldsymbol{m}_{t-1,l})^\top. \tag{24}$$

With these definitions, the posterior simplifies to

$$p(\boldsymbol{\mu}_{t,l}, \boldsymbol{\Sigma}_{t,l} \mid \{\boldsymbol{v}_{t,l}^i\}) \propto |\boldsymbol{\Sigma}_{t,l}|^{-(\nu_{t,l}+d+2)/2} \exp\left(-\frac{1}{2}\operatorname{tr}\left(\boldsymbol{\Psi}_{t,l}\boldsymbol{\Sigma}_{t,l}^{-1}\right)\right)$$

$$\times \exp\left(-\frac{\kappa_{t,l}}{2}(\boldsymbol{\mu}_{t,l} - \boldsymbol{m}_{t,l})^\top\boldsymbol{\Sigma}_{t,l}^{-1}(\boldsymbol{\mu}_{t,l} - \boldsymbol{m}_{t,l})\right).$$

This is the kernel of a Normal–Inverse–Wishart distribution $\mathrm{NIW}(\boldsymbol{m}_{t,l}, \kappa_{t,l}, \boldsymbol{\Psi}_{t,l}, \nu_{t,l})$, with the posterior hyperparameters updated as in Equations (21) to (24), which completes the proof. $\square$

### D.3. Proof of Theorem 3.5

**Theorem 3.5\***. (MSE Bound for Sequential Gradient Estimation, Detailed Version). *Consider a sequential editing process where a **previous phase** of $r \geq 1$ steps is followed by a **current phase** of $t \geq 1$ steps. Let $\mathrm{MSE}_{t,l}^{(\boldsymbol{\mu},cur)}$ denote the expected squared error at step $t$ of the current phase. We explicitly link this to the global indexing (where step $t$ corresponds to global step $r+t$) as follows:*

$$\mathrm{MSE}_{t,l}^{(\boldsymbol{\mu},cur)} := \mathbb{E}\left[\|\boldsymbol{m}_{t,l}^{(cur)} - \boldsymbol{\mu}_{t,l}^{(cur)}\|_2^2\right] \equiv \mathbb{E}\left[\|\boldsymbol{m}_{r+t,l} - \boldsymbol{\mu}_{r+t,l}\|_2^2\right],$$

*where the expectation $\mathbb{E}[\cdot]$ is taken over all data samples observed up to the current step. Similarly, let $D_j^{(\boldsymbol{\mu},cur)}$ denote the **upper bounds** on the mean drift at step $j$ of the current phase. It satisfies*

$$\|\boldsymbol{\mu}_{j-1,l}^{(cur)} - \boldsymbol{\mu}_{j,l}^{(cur)}\|_2 \equiv \|\boldsymbol{\mu}_{r+j-1,l} - \boldsymbol{\mu}_{r+j,l}\|_2 \leq D_j^{(\boldsymbol{\mu},cur)},$$

*with the boundary condition $\boldsymbol{\mu}_{0,l}^{(cur)} = \boldsymbol{\mu}_{r,l} = \boldsymbol{\mu}_{r,l}^{(pre)}$. Assume $n_t = n$ for $t \geq 1$ without loss of generality. Under Assumption 3.1(b), the MSE satisfies the following recursive bound:*

$$\mathrm{MSE}_{t,l}^{(\boldsymbol{\mu},cur)} \leq \underbrace{\frac{\kappa_{r,l}}{\kappa_{r+t,l}}\mathrm{MSE}_{r,l}^{(\boldsymbol{\mu},pre)}}_{\text{(I) Inherited History}\to 0 \ at \ rate \ O(1/t)} + \underbrace{\frac{d\sigma_+}{\kappa_{r+t,l}}\ln\left(\frac{\kappa_{r+t,l}}{\kappa_{r,l}}\right)}_{\text{(II) Current Variance}\to 0 \ at \ rate \ O(\ln t/t)} + \underbrace{\frac{2}{n\kappa_{r+t,l}}\sum_{j=1}^{t}\left(\kappa_{r+j,l}D_j^{(\boldsymbol{\mu},cur)}\right)^2}_{\text{(III) Current Drift}}. \tag{25}$$

*Furthermore, under the additional* Assumption 3.4 *that the mean drift asymptotically satisfies* $D_j^{(\boldsymbol{\mu},cur)} = O((r+j)^{-(1+\epsilon/2)})$ *for some* $\epsilon > 0$, *then the third term converges at rate* $\max\{O(\ln(r+t)/(r+t)), O((r+t)^{-\epsilon})\}$.

*Proof.* For clarity and brevity, all variables without superscripts (e.g., $\boldsymbol{m}_{k,l}, \boldsymbol{\mu}_{k,l}, \kappa_{k,l}$) refer to quantities at the **global step** $k$. Variables with the superscript cur (pre) refer to the current (previous) editing phase at local step $t$, which corresponds to the global step $k = r + t$ ($k = t$).

Let $\boldsymbol{e}_{r+t,l} := \boldsymbol{m}_{r+t,l} - \boldsymbol{\mu}_{r+t,l}$ be the estimation error and $\boldsymbol{\delta}_{r+t,l} := \boldsymbol{\mu}_{r+t-1,l} - \boldsymbol{\mu}_{r+t,l}$ be the deterministic mean drift at step $t$. Using the recursive update rule Equation (16):

$$\boldsymbol{m}_{r+t,l} = \frac{\kappa_{r+t-1,l}}{\kappa_{r+t,l}} \boldsymbol{m}_{r+t-1,l} + \frac{n}{\kappa_{r+t,l}} \overline{\boldsymbol{v}}_{r+t,l},$$

we express the error recursion by adding and subtracting the previous mean $\boldsymbol{\mu}_{r+t-1,l}$:

$$
\begin{aligned}
\boldsymbol{e}_{r+t,l} &= \frac{\kappa_{r+t-1,l}}{\kappa_{r+t,l}} \boldsymbol{m}_{r+t-1,l} + \frac{n}{\kappa_{r+t,l}} \overline{\boldsymbol{v}}_{r+t,l} - \boldsymbol{\mu}_{r+t,l} \\
&= \frac{\kappa_{r+t-1,l}}{\kappa_{r+t,l}} (\boldsymbol{m}_{r+t-1,l} - \boldsymbol{\mu}_{r+t-1,l}) + \frac{\kappa_{r+t-1,l}}{\kappa_{r+t,l}} (\boldsymbol{\mu}_{r+t-1,l} - \boldsymbol{\mu}_{r+t,l}) + \frac{n}{\kappa_{r+t,l}} (\overline{\boldsymbol{v}}_{r+t,l} - \boldsymbol{\mu}_{r+t,l}) \\
&= \frac{\kappa_{r+t-1,l}}{\kappa_{r+t,l}} \boldsymbol{e}_{r+t-1,l} + \frac{\kappa_{r+t-1,l}}{\kappa_{r+t,l}} \boldsymbol{\delta}_{r+t,l} + \frac{n}{\kappa_{r+t,l}} (\overline{\boldsymbol{v}}_{r+t,l} - \boldsymbol{\mu}_{r+t,l}),
\end{aligned}
$$

where we utilized the identity $\frac{\kappa_{r+t-1,l}}{\kappa_{r+t,l}} \boldsymbol{\mu}_{r+t,l} + \frac{n}{\kappa_{r+t,l}} \boldsymbol{\mu}_{r+t,l} = \boldsymbol{\mu}_{r+t,l}$.

To simplify the analysis of the squared norm, we define two auxiliary terms:

$$\boldsymbol{X}_{r+t,l} := \frac{\kappa_{r+t-1,l}}{\kappa_{r+t,l}} \boldsymbol{e}_{r+t-1,l}, \quad \text{and} \quad \boldsymbol{Y}_{r+t,l} := \frac{\kappa_{r+t-1,l}}{\kappa_{r+t,l}} \boldsymbol{\delta}_{r+t,l} + \frac{n}{\kappa_{r+t,l}} (\overline{\boldsymbol{v}}_{r+t,l} - \boldsymbol{\mu}_{r+t,l}).$$

Thus, $\boldsymbol{e}_{r+t,l} = \boldsymbol{X}_{r+t,l} + \boldsymbol{Y}_{r+t,l}$. Taking the expectation of the squared norm over sample randomness yields

$$\text{MSE}_{t,l}^{(\boldsymbol{\mu},\text{cur})} = \text{MSE}_{r+t,l}^{(\boldsymbol{\mu})} = \mathbb{E}[\|\boldsymbol{X}_{r+t,l}\|_2^2] + \mathbb{E}[\|\boldsymbol{Y}_{r+t,l}\|_2^2] + 2\mathbb{E}[\langle \boldsymbol{X}_{r+t,l}, \boldsymbol{Y}_{r+t,l} \rangle]. \tag{26}$$

We now bound each term on the right-hand side of Equation (26).

The first term captures the contribution from the previous error:

$$\mathbb{E}[\|\boldsymbol{X}_{r+t,l}\|_2^2] = \left(\frac{\kappa_{r+t-1,l}}{\kappa_{r+t,l}}\right)^2 \mathbb{E}[\|\boldsymbol{e}_{r+t-1,l}\|_2^2] = \left(\frac{\kappa_{r+t-1,l}}{\kappa_{r+t,l}}\right)^2 \text{MSE}_{t-1,l}^{(\boldsymbol{\mu},\text{cur})}. \tag{27}$$

The term $\boldsymbol{Y}_{r+t,l}$ contains deterministic drift and stochastic noise. Since $\overline{\boldsymbol{v}}_{r+t,l}$ is an unbiased estimator of $\boldsymbol{\mu}_{r+t,l}$, the cross-term within $\|\boldsymbol{Y}_{r+t,l}\|_2^2$ vanishes. Using $\mathbb{E}[\|\overline{\boldsymbol{v}}_{r+t,l} - \boldsymbol{\mu}_{r+t,l}\|_2^2] = \frac{1}{n} \text{tr}(\boldsymbol{\Sigma}_{r+t,l})$, we obtain

$$
\begin{aligned}
\mathbb{E}[\|\boldsymbol{Y}_{r+t,l}\|_2^2] &= \left(\frac{\kappa_{r+t-1,l}}{\kappa_{r+t,l}}\right)^2 \|\boldsymbol{\delta}_{r+t,l}\|_2^2 + \left(\frac{n}{\kappa_{r+t,l}}\right)^2 \mathbb{E}[\|\overline{\boldsymbol{v}}_{r+t,l} - \boldsymbol{\mu}_{r+t,l}\|_2^2] \\
&= \left(\frac{\kappa_{r+t-1,l}}{\kappa_{r+t,l}}\right)^2 \|\boldsymbol{\delta}_{r+t,l}\|_2^2 + \frac{n}{\kappa_{r+t,l}^2} \text{tr}(\boldsymbol{\Sigma}_{r+t,l}).
\end{aligned}
\tag{28}
$$

And for the cross term $2\mathbb{E}[\langle \boldsymbol{X}_{r+t,l}, \boldsymbol{Y}_{r+t,l} \rangle]$, independence between the current batch noise and the past error $\boldsymbol{e}_{r+t-1,l}$ implies that only the drift component of $\boldsymbol{Y}_{r+t,l}$ contributes to the inner product. Applying the weighted Young's inequality $(2\langle a, b \rangle \le \frac{1}{\eta}\|a\|_2^2 + \eta\|b\|_2^2)$ with parameter $\eta_{r+t} > 0$:

$$2\mathbb{E}[\langle \boldsymbol{X}_{r+t,l}, \boldsymbol{Y}_{r+t,l} \rangle] = 2\left(\frac{\kappa_{r+t-1,l}}{\kappa_{r+t,l}}\right)^2 \mathbb{E}[\langle \boldsymbol{e}_{r+t-1,l}, \boldsymbol{\delta}_{r+t,l} \rangle] \le \left(\frac{\kappa_{r+t-1,l}}{\kappa_{r+t,l}}\right)^2 \left(\frac{1}{\eta_{r+t}} \text{MSE}_{t-1,l}^{(\boldsymbol{\mu},\text{cur})} + \eta_{r+t}\|\boldsymbol{\delta}_{r+t,l}\|_2^2\right). \tag{29}$$

Combining these bounds (Equations (27) to (29)) into Equation (26), and substituting the bounds $\|\boldsymbol{\delta}_{r+t,l}\|_2^2 \leq \left(D_t^{(\boldsymbol{\mu},\mathrm{cur})}\right)^2$ and $\mathrm{tr}(\boldsymbol{\Sigma}_{t,l}) \leq d\sigma_+$ (Assumption 3.1(b)), we arrive at the recursive inequality:

$$\mathrm{MSE}_{t,l}^{(\boldsymbol{\mu},\mathrm{cur})} \leq \underbrace{\left(1 - \frac{n}{\kappa_{r+t,l}}\right)^2 \left(1 + \frac{1}{\eta_{r+t}}\right)}_{\alpha_{r+t}} \mathrm{MSE}_{t-1,l}^{(\boldsymbol{\mu},\mathrm{cur})} + \underbrace{\underbrace{\left(1 - \frac{n}{\kappa_{r+t,l}}\right)^2 (1 + \eta_{r+t})(D_t^{(\mathrm{cur})})^2}_{B_{r+t}^{(\mathrm{drift})}} + \underbrace{\frac{n}{\kappa_{r+t,l}^2} d\sigma_+}_{B_{r+t}^{(\mathrm{var})}}}_{B_{r+t}} . \tag{30}$$

Next we begin by unrolling the recursive bound from Equation (30). The recursion $\mathrm{MSE}_{t,l}^{(\boldsymbol{\mu},\mathrm{cur})} \leq \alpha_{r+t} \mathrm{MSE}_{t-1,l}^{(\boldsymbol{\mu},\mathrm{cur})} + B_{r+t}$ expands to

$$\mathrm{MSE}_{t,l}^{(\boldsymbol{\mu},\mathrm{cur})} \leq \left(\prod_{j=1}^{t} \alpha_{r+j}\right) \mathrm{MSE}_{0,l}^{(\boldsymbol{\mu},\mathrm{cur})} + \sum_{j=1}^{t} \left(\prod_{k=j+1}^{t} \alpha_{r+k}\right) B_{r+j}, \tag{31}$$

where $\prod_{k=t+1}^{t} \alpha_{r+k} = 1$ and $\mathrm{MSE}_{0,l}^{(\boldsymbol{\mu},\mathrm{cur})} = \mathrm{MSE}_{r,l}^{(\boldsymbol{\mu})} = \mathrm{MSE}_{r,l}^{(\boldsymbol{\mu},\mathrm{pre})}$. Our analysis proceeds by bounding the two terms on the right-hand side separately. To obtain a tight bound, we choose free parameter $\eta_{r+j} = \kappa_{r+j,l}/n$.

First, we analyze the contraction factor $\alpha_{r+j}$:

$$\alpha_{r+j} = \left(1 - \frac{n}{\kappa_{r+j,l}}\right)^2 \left(1 + \frac{n}{\kappa_{r+j,l}}\right) = 1 - \frac{n}{\kappa_{r+j,l}} - \frac{n^2}{\kappa_{r+j,l}^2}\left(1 - \frac{n}{\kappa_{r+j,l}}\right).$$

Since $\kappa_{r+j,l} \geq n > 0$, the last term is non-negative, yielding the strict upper bound $\alpha_{r+j} \leq 1 - \frac{n}{\kappa_{r+j,l}} = \frac{\kappa_{r+j-1,l}}{\kappa_{r+j,l}}$. Consequently, the cumulative product of $\alpha_{r+j}$ telescopes:

$$\prod_{j=1}^{t} \alpha_{r+j} \leq \prod_{j=1}^{t} \frac{\kappa_{r+j-1,l}}{\kappa_{r+j,l}} = \left(\frac{\kappa_{r,l}}{\kappa_{r+1,l}}\right)\left(\frac{\kappa_{r+1,l}}{\kappa_{r+2,l}}\right) \cdots \left(\frac{\kappa_{r+t-1,l}}{\kappa_{r+t,l}}\right) = \frac{\kappa_{r,l}}{\kappa_{r+t,l}}.$$

This directly bound the historical error term in Equation (31):

$$\left(\prod_{j=1}^{t} \alpha_j\right) \mathrm{MSE}_{0,l}^{(\boldsymbol{\mu},\mathrm{cur})} \leq \frac{\kappa_{r,l}}{\kappa_{r+t,l}} \mathrm{MSE}_{r,l}^{(\boldsymbol{\mu},\mathrm{pre})}. \tag{32}$$

We now bound the accumulated error term $\sum_{j=1}^{t}(\prod_{k=j+1}^{t} \alpha_{r+k})B_{r+j}$ in Equation (31) by analyzing the variance and drift components separately. Note that any partial product satisfies $\prod_{k=j+1}^{t} \alpha_{r+k} \leq \kappa_{r+j,l}/\kappa_{r+t,l}$.

For the term related to $B_{r+j}^{(\mathrm{var})}$, using the product bound, the accumulated variance is

$$\sum_{j=1}^{t} \left(\prod_{k=j+1}^{t} \alpha_{r+k}\right) B_{r+j}^{(\mathrm{var})} \leq \sum_{j=1}^{t} \left(\frac{\kappa_{r+j,l}}{\kappa_{r+t,l}}\right)\left(\frac{nd\sigma_+}{\kappa_{r+j,l}^2}\right) = \frac{nd\sigma_+}{\kappa_{r+t,l}} \sum_{j=1}^{t} \frac{1}{\kappa_{r+j,l}} = \frac{nd\sigma_+}{\kappa_{r+t,l}} \sum_{j=1}^{t} \frac{1}{\kappa_{r,l} + jn}.$$

We bound the sum using a standard integral bound for a decreasing function:

$$\sum_{j=1}^{t} \frac{1}{\kappa_{r,l} + jn} \leq \int_{0}^{t} \frac{dx}{\kappa_{r,l} + xn} = \left[\frac{1}{n} \ln(\kappa_{r,l} + xn)\right]_{0}^{t} = \frac{1}{n} \ln\left(\frac{\kappa_{r+t,l}}{\kappa_{r,l}}\right).$$

Substituting this back, the accumulated sampling variance is bounded by

$$\sum_{j=1}^{t} \left(\prod_{k=j+1}^{t} \alpha_{r+k}\right) B_{r+j}^{(\mathrm{var})} \leq \frac{d\sigma_+}{\kappa_{r+t,l}} \ln\left(\frac{\kappa_{r+t,l}}{\kappa_{r,l}}\right). \tag{33}$$

For the term related to $B_{r+j}^{(\text{drift})}$, substituting $\eta_{r+j} = \kappa_{r+j,l}/n$ and $(1 - n/\kappa_{r+j,l})^2 \leq 1$, the coefficient in $B_{r+j}^{(\text{drift})}$ satisfies $\kappa_{r+j,l}(1 + \eta_{r+j}) \leq \kappa_{r+j,l} + \kappa_{r+j,l}^2/n \leq 2\kappa_{r+j,l}^2/n$. Thus,

$$\sum_{j=1}^{t} \left( \prod_{k=j+1}^{t} \alpha_{r+k} \right) B_{r+j}^{(\text{drift})} \leq \sum_{j=1}^{t} \left( \frac{\kappa_{r+j,l}}{\kappa_{r+t,l}} \right) \left[ \left( 1 - \frac{n}{\kappa_{r+j,l}} \right)^2 \left( 1 + \frac{\kappa_{r+j,l}}{n} \right) (D_j^{(\boldsymbol{\mu},\text{cur})})^2 \right]$$

$$\leq \frac{2}{n\kappa_{r+t,l}} \sum_{j=1}^{t} \kappa_{r+j,l}^2 (D_j^{(\boldsymbol{\mu},\text{cur})})^2. \tag{34}$$

Combining the bounds from Equations (32) to (34), we arrive at the final decomposed upper bound for the MSE:

$$\text{MSE}_{t,l}^{(\boldsymbol{\mu},\text{cur})} \leq \frac{\kappa_{r,l}}{\kappa_{r+t,l}} \text{MSE}_{r,l}^{(\boldsymbol{\mu},\text{pre})} + \frac{d\sigma_+}{\kappa_{r+t,l}} \ln\left( \frac{\kappa_{r+t,l}}{\kappa_{r,l}} \right) + \frac{2}{n\kappa_{r+t,l}} \sum_{j=1}^{t} \kappa_{r+j,l}^2 (D_j^{(\boldsymbol{\mu},\text{cur})})^2.$$

The first two terms converge to 0 as $t \to \infty$, at rates of $O(1/t)$ and $O(\ln t/t)$ respectively (since $\kappa_{r+t,l} = \kappa_{r,l} + tn \sim O(t)$). The convergence of the MSE is thus dominated by the third term, the accumulated mean drift.

We now analyze this third term under the additional Assumption 3.4 that the mean drift satisfies $D_j^{(\boldsymbol{\mu},\text{cur})} = O((r + j)^{-(1+\epsilon/2)})$ as $j \to \infty$. This asymptotic condition implies the existence of a step index $J_0$ and a constant $C_{tail}$ such that $D_j^{(\boldsymbol{\mu},\text{cur})} \leq C_{tail}(r + j)^{-(1+\epsilon/2)}$ for all $j > J_0$. For the initial phase $1 \leq j \leq J_0$, the drift terms are finite (bounded by model constraints). We can therefore define a global covering constant $C_D$ as:

$$C_D = \max \left\{ \max_{1 \leq j \leq J_0} \left( D_j^{(\boldsymbol{\mu},\text{cur})}(r + j)^{1+\epsilon/2} \right), C_{tail} \right\}.$$

With this constant, the bound $D_j^{(\boldsymbol{\mu},\text{cur})} \leq C_D(r + j)^{-(1+\epsilon/2)}$ holds for all $j \geq 1$. Using the identity $\kappa_{r+j,l} = (r + j)n$, we have

$$\frac{2}{n\kappa_{r+t,l}} \sum_{j=1}^{t} \kappa_{r+j,l}^2 (D_j^{(\boldsymbol{\mu},\text{cur})})^2 \leq \frac{2}{n\kappa_{r+t,l}} \sum_{j=1}^{t} n^2(r + j)^2 \frac{C_D^2}{(r + j)^{2+\epsilon}} = \frac{2C_D^2}{r + t} \sum_{j=1}^{t} \frac{(r + j)^2}{(r + j)^{2+\epsilon}} = \frac{2C_D^2}{r + t} \sum_{j=1}^{t} \frac{1}{(r + j)^{\epsilon}}. \tag{35}$$

The convergence of this drift term now depends on the $p$-series $\sum 1/(r + j)^{\epsilon}$, where $p = \epsilon$. We analyze all possibilities for $\epsilon > 0$:

- **Case 1** ($\epsilon > 1$): The $p$-series $\sum_{j=1}^{\infty} 1/j^{\epsilon}$ converges to a finite constant $S < \infty$. The accumulated drift term is therefore bounded by

$$\frac{2}{n\kappa_{r+t,l}} \sum_{j=1}^{t} \kappa_{r+j,l}^2 (D_j^{(\boldsymbol{\mu},\text{cur})})^2 \leq \frac{2C_D^2 S}{r + t} \sim O(1/(r + t)).$$

- **Case 2** ($\epsilon = 1$): The $p$-series $\sum_{j=1}^{t} 1/(r + j)$ is the harmonic series, which grows as $O(\ln(r + t))$:

$$\sum_{j=1}^{t} \frac{1}{r + j} \leq \int_0^t \frac{1}{r + x} dx = \ln(r + t) - \ln(r) = \ln\left( 1 + \frac{t}{r} \right).$$

The accumulated drift term decays as

$$\frac{2}{n\kappa_{r+t,l}} \sum_{j=1}^{t} \kappa_{r+j,l}^2 (D_j^{(\boldsymbol{\mu},\text{cur})})^2 \leq \frac{O(\ln(r + t))}{O(r + t)} \sim O(\ln(r + t)/(r + t)).$$

- **Case 3** ($0 < \epsilon < 1$): The sum grows polynomially. Using integral approximation:

$$\sum_{j=1}^{t}(r+j)^{-\epsilon} \leq \int_{0}^{t}(r+x)^{-\epsilon}dx = \frac{(r+t)^{1-\epsilon} - r^{1-\epsilon}}{1-\epsilon} \leq \frac{(r+t)^{1-\epsilon}}{1-\epsilon}.$$

Substituting this back into Equation (35):

$$\frac{2}{n\kappa_{r+t,l}}\sum_{j=1}^{t}\kappa_{r+j,l}^{2}(D_{j}^{(\boldsymbol{\mu},\mathrm{cur})})^{2} \leq \frac{2C_{D}^{2}}{r+t} \cdot \frac{(r+t)^{1-\epsilon}}{1-\epsilon} = \frac{2C_{D}^{2}}{1-\epsilon}(r+t)^{-\epsilon}.$$

Thus, the term decays as $O((r+t)^{-\epsilon})$.

In all cases, for any $\epsilon > 0$, the accumulated mean drift term converges to 0. The overall $\mathrm{MSE}_{t,l}^{(\boldsymbol{\mu},\mathrm{cur})}$ thus converges to 0. The final convergence rate is determined by the slowest-converging term, $\max(O(\ln(r+t)/(r+t)), O(1/(r+t)^{\epsilon}))$. This completes the proof.

$\square$

*Remark* D.3 (The Benefit of Sequential History in Mean Estimation). To rigorously quantify the advantage of the sequential strategy, we first establish the cold start performance of the base case estimator, and then use this logic to analyze the sequential estimator.

In the base case with $\kappa_{0,l}^{(\mathrm{base})} = 0$, we must re-derive the bound starting from $t = 1$, using the original recursion Equation (30). At step $t = 1$, $\kappa_{1,l}^{(\mathrm{base})} = n$. The recursive bound collapses to the variance term:

$$\mathrm{MSE}_{1,l}^{(\boldsymbol{\mu},\mathrm{base})} \leq 0 + 0 + \frac{n}{(\kappa_{1,l}^{(\mathrm{base})})^{2}}d\sigma_{+} = \frac{n}{n^{2}}d\sigma_{+} = \frac{d\sigma_{+}}{n}.$$

When $t > 1$, we unroll the recursion from $t$ down to $j = 2$, and this yields the full bound for base case, where $\kappa_{t,l}^{(\mathrm{base})} = tn$:

$$\begin{aligned} \mathrm{MSE}_{t,l}^{(\boldsymbol{\mu},\mathrm{base})} &\leq \frac{\kappa_{1,l}^{(\mathrm{base})}}{\kappa_{t,l}^{(\mathrm{base})}}\mathrm{MSE}_{1,l}^{(\boldsymbol{\mu},\mathrm{base})} + \frac{d\sigma_{+}}{\kappa_{t,l}^{(\mathrm{base})}}\ln\left(\frac{\kappa_{t,l}^{(\mathrm{base})}}{\kappa_{1,l}^{(\mathrm{base})}}\right) + \frac{2}{n\kappa_{t,l}^{(\mathrm{base})}}\sum_{j=2}^{t}(\kappa_{j,l}^{(\mathrm{base})}D_{j}^{(\boldsymbol{\mu})})^{2} \\ &= \frac{d\sigma_{+}}{tn} + \frac{d\sigma_{+}}{tn}\ln(t) + O\left(\frac{\sum_{j=2}^{t}(jD_{j}^{(\boldsymbol{\mu})})^{2}}{t}\right). \end{aligned} \tag{36}$$

Now we return to the sequential estimator. The final bound derived in Equation (25) provides insight into the role of the previous phase (which yields $\kappa_{r,l} \gg n$). The bound is

$$\begin{aligned} \mathrm{MSE}_{t,l}^{(\boldsymbol{\mu},\mathrm{cur})} &\leq \frac{\kappa_{r,l}}{\kappa_{r+t,l}}\mathrm{MSE}_{r,l}^{(\boldsymbol{\mu},\mathrm{pre})} + \frac{d\sigma_{+}}{\kappa_{r+t,l}}\ln\left(\frac{\kappa_{r+t,l}}{\kappa_{r,l}}\right) + \frac{2}{n\kappa_{r+t,l}}\sum_{j=1}^{t}(\kappa_{r+j,l}D_{j}^{(\boldsymbol{\mu},\mathrm{cur})})^{2} \\ &= \frac{\kappa_{r,l}}{\kappa_{r,l}+tn}\mathrm{MSE}_{r,l}^{(\boldsymbol{\mu},\mathrm{pre})} + \frac{d\sigma_{+}}{\kappa_{r,l}+tn}\ln\left(\frac{\kappa_{r,l}+tn}{\kappa_{r,l}}\right) + \frac{2}{n(\kappa_{r,l}+tn)}\sum_{j=1}^{t}(\kappa_{r+j,l}D_{j}^{(\boldsymbol{\mu},\mathrm{cur})})^{2}. \end{aligned}$$

The previous phase (steps $1...r$) is effectively a base case process of length $r$. Using the logic from Equation (36), we expand the inherited term $\mathrm{MSE}_{r,l}^{(\boldsymbol{\mu},\mathrm{pre})}$:

$$\mathrm{MSE}_{r,l}^{(\boldsymbol{\mu},\mathrm{pre})} \leq \frac{\kappa_{1,l}}{\kappa_{r,l}}\mathrm{MSE}_{1,l}^{(\boldsymbol{\mu},\mathrm{pre})} + \frac{d\sigma_{+}}{\kappa_{r,l}}\ln\left(\frac{\kappa_{r,l}}{\kappa_{1,l}}\right) + \frac{2}{n\kappa_{r,l}}\sum_{j=2}^{r}(\kappa_{j,l}D_{j}^{(\boldsymbol{\mu},\mathrm{pre})})^{2}. \tag{37}$$

Substituting Equation (37) into Equation (25), the terms merge naturally into a global view. Specifically, the variance terms combine as $\frac{d\sigma_{+}}{\kappa_{r+t,l}}[\ln(\kappa_{r,l}/\kappa_{1,l}) + \ln(\kappa_{r+t,l}/\kappa_{r,l})] = \frac{d\sigma_{+}}{\kappa_{r+t,l}}\ln(r+t)$, and the drift sums concatenate:

$$
\begin{aligned}
\mathrm{MSE}_{t,l}^{(\boldsymbol{\mu},\mathrm{cur})} &\leq \frac{\kappa_{r,l}}{\kappa_{r,l}+tn}\left(\frac{\kappa_{1,l}}{\kappa_{r,l}}\mathrm{MSE}_{1,l}^{(\boldsymbol{\mu},\mathrm{pre})}+\frac{d\sigma_{+}}{\kappa_{r,l}}\ln\left(\frac{\kappa_{r,l}}{\kappa_{1,l}}\right)+\frac{2}{n\kappa_{r,l}}\sum_{j=2}^{r}(\kappa_{j,l}D_{j}^{(\boldsymbol{\mu},\mathrm{pre})})^{2}\right) \\
&\quad +\frac{d\sigma_{+}}{\kappa_{r,l}+tn}\ln\left(\frac{\kappa_{r,l}+tn}{\kappa_{r,l}}\right)+\frac{2}{n(\kappa_{r,l}+tn)}\sum_{j=1}^{t}(\kappa_{r+j,l}D_{j}^{(\boldsymbol{\mu},\mathrm{cur})})^{2} \\
&\leq \frac{d\sigma_{+}}{\kappa_{r,l}+tn}+\frac{d\sigma_{+}}{\kappa_{r,l}+tn}\ln(r+t)+\frac{2}{n(\kappa_{r,l}+tn)}\left[\sum_{j=2}^{r}(\kappa_{j,l}D_{j}^{(\boldsymbol{\mu},\mathrm{pre})})^{2}+\sum_{j=1}^{t}(\kappa_{j,l}D_{j}^{(\boldsymbol{\mu},\mathrm{cur})})^{2}\right] \\
&\leq \frac{d\sigma_{+}}{\kappa_{r,l}+tn}+\frac{d\sigma_{+}}{\kappa_{r,l}+tn}\ln(r+t)+O\left(\frac{\sum_{k=2}^{r+t}(kD_{k}^{(\boldsymbol{\mu})})^{2}}{r+t}\right).
\end{aligned}
\tag{38}
$$

Note: In the final step, we unified the indices into a global step $k$, where $\kappa_{r+t,l}=n(r+t)$.

Comparing the base case (Equation (36)) and the sequential case (Equation (38)), the only structural difference is the denominator: $tn$ versus $\kappa_{r+t,l}=(r+t)n$. The previous phase is thus equivalent to providing $\kappa_{r,l}=rn$ "virtual samples". This **shifts the entire convergence curve to the left**, allowing the estimator at step $t=1$ to immediately inherit the stability that a cold start estimator would require $t=\kappa_{r,l}/n=r$ steps to reach. This mathematical property manifests physically as **"Inertia"**. The large accumulated weight $\kappa_{r,l}$ acts like mass, resisting the high variance typically seen in the early steps of a cold start. While this inertia theoretically introduces a **tracking lag** (increasing the penalty from the drift term in Equation (25)), as long as the drift is controlled, the benefit of suppressing the early-stage variance significantly outweighs the cost of the lag, leading to a more stable estimation process.

### D.4. Proof of Theorem 3.6

**Lemma D.4** (Theorem 4.6.1 in (Vershynin, 2018)). *Let $\boldsymbol{A}$ be an $m\times n$ matrix with independent, mean-zero, isotropic, and sub-gaussian rows $\boldsymbol{A}_i$. Let $K=\max_i\|\boldsymbol{A}_i\|_{\psi_2}$, where $\|\cdot\|_{\psi_2}$ denotes the sub-Gaussian norm. Then there exists a constant $C>0$ such that*

$$
\mathbb{E}\left\|\frac{1}{m}\boldsymbol{A}^{\top}\boldsymbol{A}-\boldsymbol{I}_n\right\|_2\leq CK^2\left(\sqrt{\frac{n}{m}}+\frac{n}{m}\right).
$$

**Theorem 3.6\*.** (Spectral Error Bound for Sequential Covariance Estimation, Detailed Version). *Let $\widehat{\boldsymbol{\Sigma}}_{t,l}^{(cur)}=\boldsymbol{\Psi}_{r+t,l}/(\nu_{r+t,l}-d-1)$ be the NIW posterior mean estimator (as derived in Corollary 3.3) at step $t$ of the current phase. Define the expected spectral norm error $E_{t,l}^{(\boldsymbol{\Sigma},cur)}$ as*

$$
E_{t,l}^{(\boldsymbol{\Sigma},cur)}=\mathbb{E}\left\|\widehat{\boldsymbol{\Sigma}}_{t,l}^{(cur)}-\boldsymbol{\Sigma}_{t,l}^{(cur)}\right\|_2\equiv\mathbb{E}\left\|\widehat{\boldsymbol{\Sigma}}_{r+t,l}-\boldsymbol{\Sigma}_{r+t,l}\right\|_2,
$$

*where the expectation $\mathbb{E}[\cdot]$ is taken over all samples up to the current step. Let $\Delta_j^{(\boldsymbol{\Sigma},cur)}$ denote the **upper bounds** on the covariance drift at step $j$ of the current phase. It satisfies*

$$
\|\boldsymbol{\Sigma}_{j-1,l}^{(cur)}-\boldsymbol{\Sigma}_{j,l}^{(cur)}\|_2\equiv\|\boldsymbol{\Sigma}_{r+j-1,l}-\boldsymbol{\Sigma}_{r+j,l}\|_2\leq\Delta_j^{(\boldsymbol{\Sigma},cur)},
$$

*with the boundary condition $\boldsymbol{\Sigma}_{0,l}^{(cur)}=\boldsymbol{\Sigma}_{r,l}=\boldsymbol{\Sigma}_{r,l}^{(pre)}$. Assume $n_t=n$ for $t\geq 1$ without loss of generality. Under Assumption 3.1(b) and assuming $\nu_{r+t,l}>d+1$, the error $E_{t,l}^{(\boldsymbol{\Sigma},cur)}$ satisfies the following recursive bound:*

$$
\begin{aligned}
E_{t,l}^{(\boldsymbol{\Sigma},cur)} &\leq \underbrace{\frac{\nu_{r,l}'}{\nu_{r+t,l}'}E_{r,l}^{(\boldsymbol{\Sigma},pre)}}_{\to 0\ at\ rate\ O(1/t)}+\underbrace{\frac{(C_2(\sqrt{dn}+d)+d)\sigma_{+}t}{\nu_{r+t,l}'}}_{\to Const.} \\
&\quad +\frac{1}{\nu_{r+t,l}'}\sum_{j=1}^{t}\nu_{r+j-1,l}'\Delta_j^{(\boldsymbol{\Sigma},cur)}+\frac{2n}{\nu_{r+t,l}'}\sum_{j=1}^{t}(D_j^{(\boldsymbol{\mu},cur)})^{2}+\frac{2n}{\nu_{r+t,l}'}\sum_{j=1}^{t}\mathrm{MSE}_{j-1,l}^{(\boldsymbol{\mu},cur)},
\end{aligned}
\tag{39}
$$

where $\nu'_{t,l} = \nu_{t,l} - d - 1$. *Furthermore, under the additional* Assumption 3.4 *that the drifts asymptotically satisfy* $(D_j^{(\boldsymbol{\mu},cur)})^2 \leq C_D j^{-(2+\epsilon)}$ *and* $\Delta_j^{(\boldsymbol{\Sigma},cur)} \leq C_\Delta j^{-(1+\epsilon')}$ *for constants* $\epsilon, \epsilon' > 0$, *then the last three terms converge at rates* $\max\{O(\ln(r+t)/(r+t)), O((r+t)^{-\epsilon'})\}$, $O(1/(r+t))$, *and* $\max\{O((\ln(r+t))^2/(r+t)), O((r+t)^{-\epsilon})\}$ *respectively.*

*Proof.* For brevity, we omit superscripts for quantities defined at the global step $k$ (e.g., $\boldsymbol{\Psi}_{k,l}, \boldsymbol{\Sigma}_{k,l}$), while employing the superscripts cur (pre) to denote the current (previous) editing phase at local step $t$, corresponding to global step $k = r + t$ ($k = t$).

Recall the recursive update rule $\boldsymbol{\Psi}_{r+t,l} = \boldsymbol{\Psi}_{r+t-1,l} + \boldsymbol{S}_{r+t,l} + \frac{\kappa_{r+t-1,l} n}{\kappa_{r+t,l}} \boldsymbol{Q}_{r+t,l}$ with $\boldsymbol{Q}_{r+t,l} = (\overline{\boldsymbol{v}}_{r+t,l} - \boldsymbol{m}_{r+t-1,l})(\overline{\boldsymbol{v}}_{r+t,l} - \boldsymbol{m}_{r+t-1,l})^\top$ and $\nu_{r+t,l} = \nu_{r+t-1,l} + n$ (refer to Equation (17) and Equation (15)). By substituting this into the definition of the estimator $\widehat{\boldsymbol{\Sigma}}_{r+t,l}$ and applying the triangle inequality along with the linearity of expectation, we decompose the total expected error into three distinct components:

$$
\begin{aligned}
E_{t,l}^{(\boldsymbol{\Sigma},cur)} &= \mathbb{E}\left\|\widehat{\boldsymbol{\Sigma}}_{r+t,l} - \boldsymbol{\Sigma}_{r+t,l}\right\|_2 \\
&\leq \underbrace{\frac{\mathbb{E}\left\|\boldsymbol{S}_{r+t,l} - n\boldsymbol{\Sigma}_{r+t,l}\right\|_2}{\nu_{r+t,l} - d - 1}}_{T_{var}} + \underbrace{\frac{\mathbb{E}\left\|\boldsymbol{\Psi}_{r+t-1,l} - (\nu_{r+t-1,l} - d - 1)\boldsymbol{\Sigma}_{r+t,l}\right\|_2}{\nu_{r+t,l} - d - 1}}_{T_{shr}} \\
&\quad + \underbrace{\frac{\kappa_{r+t-1,l} n}{\kappa_{r+t,l}(\nu_{r+t,l} - d - 1)} \mathbb{E}\left\|(\overline{\boldsymbol{v}}_{r+t,l} - \boldsymbol{m}_{r+t-1,l})(\overline{\boldsymbol{v}}_{r+t,l} - \boldsymbol{m}_{r+t-1,l})^\top\right\|_2}_{T_{mean}}.
\end{aligned}
$$

We now proceed to bound each of these three terms. First, we address the variance term $T_{var}$ by invoking covariance concentration results from Lemma D.4. Let $\boldsymbol{z}_{r+t,l}^i := \boldsymbol{\Sigma}_{r+t,l}^{-1/2}(\boldsymbol{v}_{r+t,l}^i - \boldsymbol{\mu}_{r+t,l}) \sim \mathcal{N}_d(\boldsymbol{0}, \boldsymbol{I}_d)$ denote the whitened sample vectors. And the sample mean then satisfies $\bar{\boldsymbol{z}}_{r+t,l} = \boldsymbol{\Sigma}_{r+t,l}^{-1/2}(\overline{\boldsymbol{v}}_{r+t,l} - \boldsymbol{\mu}_{r+t,l}) \sim \mathcal{N}_d\left(0, \frac{1}{N}\boldsymbol{I}_d\right)$. The scatter matrix $\boldsymbol{S}_{r+t,l} = \sum_{i=1}^n (\boldsymbol{v}_{r+t,l}^i - \bar{\boldsymbol{v}}_{r+t,l})(\boldsymbol{v}_{r+t,l}^i - \bar{\boldsymbol{v}}_{r+t,l})^\top$ can be expanded and rewritten in terms of $\boldsymbol{z}_{r+t,l}^i$ as

$$
\begin{aligned}
\frac{1}{n}\boldsymbol{S}_{r+t,l} &= \frac{1}{n}\sum_{i=1}^n (\boldsymbol{v}_{r+t,l}^i - \overline{\boldsymbol{v}}_{r+t,l})(\boldsymbol{v}_{r+t,l}^i - \overline{\boldsymbol{v}}_{r+t,l})^\top \\
&= \frac{1}{n}\sum_{i=1}^n \left[(\boldsymbol{v}_{r+t,l}^i - \boldsymbol{\mu}_{r+t,l}) + (\boldsymbol{\mu}_{r+t,l} - \overline{\boldsymbol{v}}_{r+t,l})\right]\left[(\boldsymbol{v}_{r+t,l}^i - \boldsymbol{\mu}_{r+t,l})^\top + (\boldsymbol{\mu}_{r+t,l} - \overline{\boldsymbol{v}}_{r+t,l})^\top\right] \\
&= \frac{1}{n}\sum_{i=1}^n \left[(\boldsymbol{v}_{r+t,l}^i - \boldsymbol{\mu}_{r+t,l})(\boldsymbol{v}_{r+t,l}^i - \boldsymbol{\mu}_{r+t,l})^\top\right] + (\boldsymbol{\mu}_{r+t,l} - \bar{\boldsymbol{v}}_{r+t,l})(\bar{\boldsymbol{v}}_{r+t,l} - \boldsymbol{\mu}_{r+t,l})^\top \\
&= \boldsymbol{\Sigma}_{r+t,l}^{1/2}\left(\frac{1}{n}\sum_{i=1}^n \boldsymbol{z}_{r+t,l}^i(\boldsymbol{z}_{r+t,l}^i)^\top - \bar{\boldsymbol{z}}_{r+t,l}\bar{\boldsymbol{z}}_{r+t,l}^\top\right)\boldsymbol{\Sigma}_{r+t,l}^{1/2}.
\end{aligned}
$$

This allows us to express the error matrix as

$$
\frac{1}{n}\boldsymbol{S}_{r+t,l} - \boldsymbol{\Sigma}_{r+t,l} = \boldsymbol{\Sigma}_{r+t,l}^{1/2}\left(\frac{1}{n}\sum_{i=1}^n \boldsymbol{z}_{r+t,l}^i(\boldsymbol{z}_{r+t,l}^i)^\top - \boldsymbol{I}_d - \bar{\boldsymbol{z}}_{r+t,l}\bar{\boldsymbol{z}}_{r+t,l}^\top\right)\boldsymbol{\Sigma}_{r+t,l}^{1/2}.
$$

Taking the spectral norm and expectation and noting that $\|\boldsymbol{\Sigma}_{r+t,l}\|_2 \leq \sigma_+$ by Assumption 3.1(b), we obtain

$$
\begin{aligned}
\mathbb{E}\left\|\frac{1}{n}\boldsymbol{S}_{r+t,l} - \boldsymbol{\Sigma}_{r+t,l}\right\|_2 &\leq \|\boldsymbol{\Sigma}_{r+t,l}\|_2 \mathbb{E}\left\|\frac{1}{n}\sum_{i=1}^n \boldsymbol{z}_{r+t,l}^i(\boldsymbol{z}_{r+t,l}^i)^\top - \boldsymbol{I}_d - \bar{\boldsymbol{z}}_{r+t,l}\bar{\boldsymbol{z}}_{r+t,l}^\top\right\|_2 \\
&\leq \sigma_+\left(\mathbb{E}\left\|\frac{1}{n}\sum_{i=1}^n \boldsymbol{z}_{r+t,l}^i(\boldsymbol{z}_{r+t,l}^i)^\top - \boldsymbol{I}_d\right\|_2 + \mathbb{E}\|\bar{\boldsymbol{z}}_{r+t,l}\bar{\boldsymbol{z}}_{r+t,l}^\top\|_2\right).
\end{aligned}
$$

By Lemma D.4, the first term is bounded by $C_1\left(\sqrt{d/n} + d/n\right)$ for some constant $C_1$. For the second term, observe that the spectral norm of the rank-one matrix $\bar{z}_{r+t,l}\bar{z}_{r+t,l}^\top$ reduces to $\|\bar{z}_{r+t,l}\|^2$. Since $\mathbb{E}[\|\bar{z}_{r+t,l}\|_2^2] = \text{tr}(\text{Cov}(\bar{z}_{r+t,l})) = \text{tr}(\frac{1}{n}\boldsymbol{I}_d) = d/n$, for some constant $C_2$, we have

$$\mathbb{E}\left\|\frac{1}{n}\boldsymbol{S}_{r+t,l} - \boldsymbol{\Sigma}_{r+t,l}\right\|_2 \leq \sigma_+\left(C_1\left(\sqrt{\frac{d}{n}} + \frac{d}{n}\right) + \frac{d}{n}\right) \leq C_2\sigma_+\left(\sqrt{\frac{d}{n}} + \frac{d}{n}\right).$$

Consequently,

$$T_{\text{var}} \leq \frac{n}{\nu_{r+t,l} - d - 1}C_2\sigma_+\left(\sqrt{\frac{d}{n}} + \frac{d}{n}\right) = \frac{C_2\sigma_+}{\nu_{r+t,l} - d - 1}\left(\sqrt{dn} + d\right). \tag{40}$$

Next, consider the shrinkage term $T_{\text{shr}}$. Using the triangle inequality and the definition of the previous estimator $\widehat{\boldsymbol{\Sigma}}_{r+t-1}$, we have

$$\begin{aligned}
\mathbb{E}\left\|\boldsymbol{\Psi}_{r+t-1} - (\nu_{r+t-1} - d - 1)\boldsymbol{\Sigma}_{r+t,l}\right\|_2 &\leq \mathbb{E}\left\|\boldsymbol{\Psi}_{r+t-1} - (\nu_{r+t-1} - d - 1)\boldsymbol{\Sigma}_{r+t-1}\right\|_2 \\
&\quad + (\nu_{r+t-1} - d - 1)\mathbb{E}\left\|\boldsymbol{\Sigma}_{r+t-1} - \boldsymbol{\Sigma}_{r+t,l}\right\|_2 \\
&= (\nu_{r+t-1} - d - 1)\mathbb{E}\left\|\widehat{\boldsymbol{\Sigma}}_{r+t-1} - \boldsymbol{\Sigma}_{r+t-1}\right\|_2 \\
&\quad + (\nu_{r+t-1} - d - 1)\left\|\boldsymbol{\Sigma}_{r+t-1} - \boldsymbol{\Sigma}_{r+t,l}\right\|_2.
\end{aligned}$$

Since $\|\boldsymbol{\Sigma}_{r+t-1} - \boldsymbol{\Sigma}_{r+t,l}\|_2 \leq \Delta_t^{(\boldsymbol{\Sigma},\text{cur})}$, it follows immediately that

$$T_{\text{shr}} \leq \frac{\nu_{r+t-1} - d - 1}{\nu_{r+t,l} - d - 1}\left(E_{t-1,l}^{(\boldsymbol{\Sigma},\text{cur})} + \Delta_t^{(\boldsymbol{\Sigma},\text{cur})}\right). \tag{41}$$

Finally, we analyze the mean correction term $T_{\text{mean}}$. Since the spectral norm of the rank-one matrix $\boldsymbol{Q}_{r+t,l}$ is simply $\|\bar{\boldsymbol{v}}_{r+t,l} - \boldsymbol{m}_{r+t-1}\|_2^2$, we decompose the error vector as

$$\bar{\boldsymbol{v}}_{r+t,l} - \boldsymbol{m}_{r+t-1} = (\bar{\boldsymbol{v}}_{r+t,l} - \boldsymbol{\mu}_{r+t,l}) + (\boldsymbol{\mu}_{r+t,l} - \boldsymbol{\mu}_{r+t-1}) + (\boldsymbol{\mu}_{r+t-1} - \boldsymbol{m}_{r+t-1}).$$

Applying the triangle inequality for norms, we have

$$\begin{aligned}
\mathbb{E}[\|\bar{\boldsymbol{v}}_{r+t,l} - \boldsymbol{m}_{r+t-1}\|_2^2] &= \mathbb{E}[\|\bar{\boldsymbol{v}}_{r+t,l} - \boldsymbol{\mu}_{r+t,l}\|_2^2] + \|\boldsymbol{\mu}_{r+t,l} - \boldsymbol{\mu}_{r+t-1}\|_2^2 + \text{MSE}_{t-1,l}^{(\boldsymbol{\mu},\text{cur})} \\
&\quad + 2\mathbb{E}[\langle\bar{\boldsymbol{v}}_{r+t,l} - \boldsymbol{\mu}_{r+t,l}, \boldsymbol{\mu}_{r+t,l} - \boldsymbol{\mu}_{r+t-1}\rangle] \\
&\quad + 2\mathbb{E}[\langle\bar{\boldsymbol{v}}_{r+t,l} - \boldsymbol{\mu}_{r+t,l}, \boldsymbol{\mu}_{r+t-1} - \boldsymbol{m}_{r+t-1}\rangle] \\
&\quad + 2\mathbb{E}[\langle\boldsymbol{\mu}_{r+t,l} - \boldsymbol{\mu}_{r+t-1}, \boldsymbol{\mu}_{r+t-1} - \boldsymbol{m}_{r+t-1}\rangle].
\end{aligned}$$

The cross-terms involving $(\bar{\boldsymbol{v}}_{r+t,l} - \boldsymbol{\mu}_{r+t,l})$ vanish in expectation. Specifically, for $2\mathbb{E}[\langle\bar{\boldsymbol{v}}_{r+t,l} - \boldsymbol{\mu}_{r+t,l}, \boldsymbol{\mu}_{r+t,l} - \boldsymbol{\mu}_{r+t-1}\rangle]$, the term $(\boldsymbol{\mu}_{r+t,l} - \boldsymbol{\mu}_{r+t-1})$ is a fixed (deterministic) quantity, so the expectation operates only on $(\bar{\boldsymbol{v}}_{r+t,l} - \boldsymbol{\mu}_{r+t,l})$, which has zero expectation. For $2\mathbb{E}[\langle\bar{\boldsymbol{v}}_{r+t,l} - \boldsymbol{\mu}_{r+t,l}, \boldsymbol{\mu}_{r+t-1} - \boldsymbol{m}_{r+t-1}\rangle]$, the current batch noise $(\bar{\boldsymbol{v}}_{r+t,l} - \boldsymbol{\mu}_{r+t,l})$ is independent of the previous estimation error $(\boldsymbol{\mu}_{r+t-1} - \boldsymbol{m}_{r+t-1})$, allowing the expectation to factorize into a product of expectations, where $\mathbb{E}[\bar{\boldsymbol{v}}_{r+t,l} - \boldsymbol{\mu}_{r+t,l}] = \boldsymbol{0}$. For the last cross-term $2\mathbb{E}[\langle\boldsymbol{\mu}_{r+t,l} - \boldsymbol{\mu}_{r+t-1}, \boldsymbol{\mu}_{r+t-1} - \boldsymbol{m}_{r+t-1}\rangle]$, we apply Young's inequality with parameter $\xi_{r+t} > 0$: $2\langle a, b\rangle \leq \frac{1}{\xi_{r+t}}\|a\|_2^2 + \xi_{r+t}\|b\|_2^2$. This yields

$$2\mathbb{E}[\langle\boldsymbol{\mu}_{r+t,l} - \boldsymbol{\mu}_{r+t-1}, \boldsymbol{\mu}_{r+t-1} - \boldsymbol{m}_{r+t-1}\rangle] \leq \frac{1}{\xi_{r+t}}\|\boldsymbol{\mu}_{r+t,l} - \boldsymbol{\mu}_{r+t-1}\|_2^2 + \xi_{r+t}\mathbb{E}[\|\boldsymbol{\mu}_{r+t-1} - \boldsymbol{m}_{r+t-1}\|_2^2].$$

Incorporating this result, and applying the drift bound $\|\boldsymbol{\mu}_{r+t,l} - \boldsymbol{\mu}_{r+t-1}\|_2^2 \leq (D_t^{(\boldsymbol{\mu},\text{cur})})^2$ alongside the trace bound

$\text{tr}(\boldsymbol{\Sigma}_{r+t,l}) \le d\sigma_+$, we arrive at

$$
\begin{aligned}
\mathbb{E}[\|\overline{\boldsymbol{v}}_{r+t,l} - \boldsymbol{m}_{r+t-1}\|_2^2] &\le \frac{\text{tr}(\boldsymbol{\Sigma}_{r+t,l})}{n} + \|\boldsymbol{\mu}_{r+t,l} - \boldsymbol{\mu}_{r+t-1}\|_2^2 + \text{MSE}_{t-1,l}^{(\boldsymbol{\mu},\text{cur})} \\
&\quad + \frac{1}{\xi_{r+t}}\|\boldsymbol{\mu}_{r+t,l} - \boldsymbol{\mu}_{r+t-1}\|_2^2 + \xi_{r+t}\,\text{MSE}_{t-1,l}^{(\boldsymbol{\mu},\text{cur})} \\
&= \frac{\text{tr}(\boldsymbol{\Sigma}_{r+t,l})}{n} + (1 + \frac{1}{\xi_{r+t}})\|\boldsymbol{\mu}_{r+t,l} - \boldsymbol{\mu}_{r+t-1}\|_2^2 + (1 + \xi_{r+t})\,\text{MSE}_{t-1,l}^{(\boldsymbol{\mu},\text{cur})} \\
&\le \frac{d\sigma_+}{n} + (1 + \frac{1}{\xi_{r+t}})(D_t^{(\boldsymbol{\mu},\text{cur})})^2 + (1 + \xi_{r+t})\,\text{MSE}_{t-1,l}^{(\boldsymbol{\mu},\text{cur})}\,.
\end{aligned}
$$

Thus, $T_{\text{mean}}$ is bounded by

$$
T_{\text{mean}} \le \frac{\kappa_{r+t-1}n}{\kappa_{r+t,l}(\nu_{r+t,l} - d - 1)}\left(\frac{d\sigma_+}{n} + (1 + \frac{1}{\xi_{r+t}})(D_t^{(\boldsymbol{\mu},\text{cur})})^2 + (1 + \xi_{r+t})\,\text{MSE}_{t-1,l}^{(\boldsymbol{\mu},\text{cur})}\right). \tag{42}
$$

Combining the bounds for $T_{\text{var}}$ (Equation (40)), $T_{\text{shr}}$ (Equation (41)) and $T_{\text{mean}}$ (Equation (42)) yields the recursive inequality:

$$
\begin{aligned}
E_{t,l}^{(\boldsymbol{\Sigma},\text{cur})} &\le \underbrace{\left(1 - \frac{n}{\nu_{r+t,l} - d - 1}\right)}_{\beta_{r+t}} E_{t-1,l}^{(\boldsymbol{\Sigma},\text{cur})} \\
&\quad + \frac{C_2\sigma_+}{\nu_{r+t,l} - d - 1}\left(\sqrt{dn} + d\right) + \left(1 - \frac{n}{\nu_{r+t,l} - d - 1}\right)\Delta_t^{(\boldsymbol{\Sigma},\text{cur})} \\
&\quad + \left(1 - \frac{n}{\kappa_{r+t,l}}\right)\frac{n}{(\nu_{r+t,l} - d - 1)}\left(\frac{d\sigma_+}{n} + (1 + \frac{1}{\xi_{r+t}})(D_t^{(\boldsymbol{\mu},\text{cur})})^2 + (1 + \xi_{r+t})\,\text{MSE}_{t-1,l}^{(\boldsymbol{\mu},\text{cur})}\right).
\end{aligned} \tag{43}
$$

We now analyze the convergence of the spectral norm error by unrolling the recursion $E_{t,l}^{(\boldsymbol{\Sigma},\text{cur})} \le \beta_{r+t}E_{t-1,l}^{(\boldsymbol{\Sigma},\text{cur})} + B_{r+t}$ from Equation (43), where we define the contraction factor $\beta_{r+t} := 1 - n/\nu'_{r+t,l}$ with effective degrees of freedom $\nu'_{k,l} := \nu_{k,l} - d - 1$. The update rule $\nu_{r+t,l} = \nu_{r+t-1} + n$ implies $\nu'_{r+t,l} = \nu'_{r,l} + tn$. The non-recursive term $B_{r+t}$ decomposes into variance, drift, and coupled components:

$$
\begin{aligned}
B_{r+t} &= \underbrace{\frac{C_2\sigma_+(\sqrt{dn} + d)}{\nu'_{r+t}}}_{B_{r+t}^{(\text{var})}} + \underbrace{\beta_{r+t}\Delta_t^{(\boldsymbol{\Sigma},\text{cur})}}_{B_{r+t}^{(\text{drift})}} \\
&\quad + \underbrace{\left(1 - \frac{n}{\kappa_{r+t,l}}\right)\frac{n}{\nu'_{r+t,l}}\left(\frac{d\sigma_+}{n} + (1 + \frac{1}{\xi_{r+t}})(D_t^{(\boldsymbol{\mu},\text{cur})})^2 + (1 + \xi_{r+t})\,\text{MSE}_{t-1,l}^{(\boldsymbol{\mu},\text{cur})}\right)}_{B_{r+t}^{(\text{coupled})}}.
\end{aligned}
$$

Unrolling the recursion $E_{t,l}^{(\boldsymbol{\Sigma},\text{cur})} \le \beta_{r+t}E_{t-1}^{(\boldsymbol{\Sigma},\text{cur})} + B_{r+t}$ from $t$ down to 0 yields

$$
E_{t,l}^{(\boldsymbol{\Sigma},\text{cur})} \le \left(\prod_{j=1}^{t}\beta_{r+j}\right)E_{r,l}^{(\boldsymbol{\Sigma},\text{pre})} + \sum_{j=1}^{t}\left(\prod_{k=j+1}^{t}\beta_{r+k}\right)B_{r+j},
$$

where $\prod_{k=t+1}^{t}\beta_{r+k} = 1$. Observing that $\beta_{r+j} = \nu'_{r+j-1,l}/\nu'_{r+j,l}$, the product term telescopes as $\prod_{j=1}^{t}\beta_{r+j} = \nu'_{r,l}/\nu'_{r+t,l}$. Similarly, the partial product $\prod_{k=j+1}^{t}\beta_{r+k}$ simplifies to $\nu'_{r+j,l}/\nu'_{r+t,l}$. Substituting these into the unrolled sum, we obtain the explicit bound:

$$
E_{t,l}^{(\boldsymbol{\Sigma},\text{cur})} \le \underbrace{\frac{\nu'_{r,l}}{\nu'_{r+t,l}}E_{r,l}^{(\boldsymbol{\Sigma},\text{pre})}}_{\mathcal{T}_{\text{init}}} + \underbrace{\sum_{j=1}^{t}\frac{\nu'_{r+j,l}}{\nu'_{r+t,l}}B_{r+j}}_{\mathcal{T}_{\text{accum}}}. \tag{44}
$$

We now analyze the asymptotic behavior of these terms. The initial error term $\mathcal{T}_{\text{init}}$ decays as $O(1/t)$ since $\nu'_{r+t,l} \sim O(t)$, and thus vanishes as $t \to \infty$. The accumulated error $\mathcal{T}_{\text{accum}}$ is analyzed by decomposing $B_{r+j}$ into its three constituents.

First, the accumulated intrinsic variance (arising from $B_{r+j}^{(\text{var})}$) is given by

$$\sum_{j=1}^{t} \frac{\nu'_{r+j,l}}{\nu'_{r+t,l}} B_{r+j}^{(\text{var})} = \sum_{j=1}^{t} \frac{\nu'_{r+j,l}}{\nu'_{r+t,l}} \left( \frac{C_2 \sigma_+(\sqrt{dn}+d)}{\nu'_{r+j,l}} \right) = \sum_{j=1}^{t} \frac{C_2 \sigma_+(\sqrt{dn}+d)}{\nu'_{r+t,l}} = \frac{(C_2 \sigma_+(\sqrt{dn}+d))t}{\nu'_{r,l}+tn}.$$

As $t \to \infty$, this term converges to the constant $C_2 \sigma_+(\sqrt{dn}+d)/n$. This represents the irreducible sampling noise inherent to the estimator.

Second, regarding the covariance drift term, we have

$$\sum_{j=1}^{t} \frac{\nu'_{r+j,l}}{\nu'_{r+t,l}} B_{r+j}^{(\text{drift})} = \sum_{j=1}^{t} \frac{\nu'_{r+j,l}}{\nu'_{r+t,l}} \left( \frac{\nu'_{r+j-1,l}}{\nu'_{r+j,l}} \right) \Delta_j^{(\Sigma,\text{cur})} = \frac{1}{\nu'_{r+t,l}} \sum_{j=1}^{t} \nu'_{r+j-1,l} \Delta_j^{(\Sigma,\text{cur})}.$$

Assumption 3.4 posits that the drifts decay asymptotically. Specifically, there exist constants $C_{D,\text{tail}}, C_{\Delta,\text{tail}}$ and a step index $J_0$ such that for all $j > J_0$, the bounds $D_j \leq C_{D,\text{tail}}(r+j)^{-(1+\epsilon/2)}$ and $\Delta_j \leq C_{\Delta,\text{tail}}(r+j)^{-(1+\epsilon')}$ hold. Since the drifts are finite for the initial finite steps $1 \leq j \leq J_0$, we can construct global covering constants $C_D$ and $C_\Delta$ (as done in Theorem 3.5) such that for all $j \geq 1$:

$$D_j^{(\boldsymbol{\mu},\text{cur})} \leq C_D(r+j)^{-(1+\epsilon/2)} \quad \text{and} \quad \Delta_j^{(\Sigma,\text{cur})} \leq C_\Delta(r+j)^{-(1+\epsilon')}.$$

Recall that $\nu'_{k,l} = \nu_{0,l} + kn - d - 1$. For the index corresponding to step $r+j-1$, we have

$$\nu'_{r+j-1,l} = \nu_{0,l} + (r+j-1)n - d - 1 \leq \nu_{0,l} + n(r+j).$$

Since $r+j \geq 1$, we trivially have $\nu_{0,l} \leq \nu_{0,l}(r+j)$. Consequently,

$$\nu'_{r+j-1,l} \leq (\nu_{0,l}+n)(r+j).$$

Defining the constant $C_\nu := \nu_{0,l} + n$, we have the rigorous bound $\nu'_{r+j-1,l} \leq C_\nu(r+j)$. Substituting this and the uniform bound for $\Delta_j$, we have

$$\nu'_{r+j-1,l} \Delta_j^{(\Sigma,\text{cur})} \leq C_\nu(r+j) \cdot C_\Delta(r+j)^{-(1+\epsilon')} = C'(r+j)^{-\epsilon'},$$

where $C' = C_\nu C_\Delta$. Thus, the term is bounded by

$$\frac{1}{\nu'_{r+t,l}} \sum_{j=1}^{t} \nu'_{r+j-1,l} \Delta_j^{(\Sigma,\text{cur})} \leq \frac{C'}{\nu'_{r+t,l}} \sum_{j=1}^{t} (r+j)^{-\epsilon'}.$$

Since $\nu'_{r+t,l} \sim O(r+t)$, we analyze the ratio $\frac{\sum_{j=1}^{t}(r+j)^{-\epsilon'}}{r+t}$ for $\epsilon' > 0$:

- **Case 1** ($\epsilon' > 1$): The sum $\sum_{j=1}^{\infty} 1/(r+j)^{\epsilon'}$ converges to a constant $S_\Sigma$. The term is $O(1/\nu'_{r+t,l}) \sim O(1/(r+t)) \to 0$.

- **Case 2** ($\epsilon' = 1$): The sum $\sum_{j=1}^{t} 1/(r+j) \sim O(\ln(r+t))$. The term is $O(\ln(r+t)/\nu'_{r+t,l}) \sim O(\ln(r+t)/(r+t)) \to 0$.

- **Case 3** ($0 < \epsilon' < 1$): The sum $\sum_{j=1}^{t} 1/(r+j)^{\epsilon'} \sim O((r+t)^{1-\epsilon'})$. The term is $O((r+t)^{1-\epsilon'}/\nu'_{r+t,l}) \sim O((r+t)^{-\epsilon'}) \to 0$.

In all plausible cases ($\epsilon' > 0$), the accumulated covariance drift converges to 0 at the rate $\max\{O((r+t)^{-\epsilon'}), O(\ln(r+t)/(r+t))\}$.

Finally, we consider the coupled error from mean estimation. Setting $\xi_{r+j} = 1$ and noting that $(1 - n/\kappa_{r+j,l}) \leq 1$, the term simplifies via cancellation of $\nu'_{r+j,l}$:

$$\sum_{j=1}^{t} \frac{\nu'_{r+j,l}}{\nu'_{r+t,l}} B_{r+j}^{(\text{coupled})} \leq \frac{n}{\nu'_{r+t,l}} \sum_{j=1}^{t} \left( \frac{d\sigma_+}{n} + 2(D_j^{(\boldsymbol{\mu},\text{cur})})^2 + 2\,\text{MSE}_{j-1,l}^{(\boldsymbol{\mu},\text{cur})} \right)$$

$$= \frac{d\sigma_+}{\nu'_{r+t,l}} t + \frac{2n}{\nu'_{r+t,l}} \sum_{j=1}^{t} (D_j^{(\boldsymbol{\mu},\text{cur})})^2 + \frac{2n}{\nu'_{r+t,l}} \sum_{j=1}^{t} \text{MSE}_{j-1,l}^{(\boldsymbol{\mu},\text{cur})}.$$

This sum is dominated by the constant term $d\sigma_+/n$, leading to a limit of $d\sigma_+/n$. The remaining components involve the mean drift squared and the mean MSE. Since $(D_j^{(\boldsymbol{\mu},\text{cur})})^2 \leq C_D^2(r+j)^{-(2+\epsilon)}$ and utilizing Theorem 3.5 (where $\text{MSE}_{t,l}^{(\boldsymbol{\mu},\text{cur})} \sim \max\{O(\ln(r+j)/(r+j)), O(1/(r+j)^\epsilon)\}$), the sums $\sum_j (D_j^{(\boldsymbol{\mu},\text{cur})})^2$ and $\sum_j \text{MSE}$ grow sub-linearly (converging or logarithmic), causing their contribution to the average to vanish as $O(1/(r+t))$ or $\max\{O((\ln(r+t))^2/(r+t)), O(1/(r+t)^\epsilon)\}$ respectively.

Combining these results into Equation (44), we arrive at the final decomposed upper bound for $E_{t,l}^{(\boldsymbol{\Sigma},\text{cur})}$ in Equation (39). Therefore, the spectral norm error $E_{t,l}^{(\boldsymbol{\Sigma},\text{cur})}$ is bounded by

$$
E_{t,l}^{(\boldsymbol{\Sigma},\text{cur})} \leq \underbrace{\frac{\nu'_{r,l}}{\nu'_{r,l} + tn} E_{r,l}^{(\boldsymbol{\Sigma},\text{pre})}}_{O(1/t)} + \underbrace{\frac{C_2 \sigma_+(\sqrt{dn} + d)t}{\nu'_{r+t,l}}}_{O(1)} + \underbrace{\frac{1}{\nu'_{r+t,l}} \sum_{j=1}^{t} \nu'_{r+j-1,l} \Delta_j^{(\boldsymbol{\Sigma},\text{cur})}}_{\max\{O(1/t^{\epsilon'}), O(\ln t/t)\}}
$$
$$
+ \underbrace{\sum_{j=1}^{t} \frac{\nu'_{r+j,l}}{\nu'_{r+t,l}} \left[ \left(1 - \frac{n}{\kappa_{r+j,l}}\right) \frac{n}{\nu'_{r+j,l}} \left( \frac{d\sigma_+}{n} + (1 + \frac{1}{\xi_{r+j}})(D_j^{(\boldsymbol{\mu},\text{cur})})^2 + (1 + \xi_{r+j})\,\text{MSE}_{j-1,l}^{(\boldsymbol{\mu},\text{cur})} \right) \right]}_{O(1)}.
\tag{45}
$$

The spectral norm error $E_{t,l}^{(\boldsymbol{\Sigma},\text{cur})}$ does not vanish but remains bounded within a constant radius determined by the sampling noise and dimension:

$$\limsup_{t \to \infty} E_{t,l}^{(\boldsymbol{\Sigma},\text{cur})} \leq \frac{C_2 \sigma_+(\sqrt{dn} + d)}{n} + \frac{d\sigma_+}{n} = \frac{C_2 \sigma_+(\sqrt{dn} + d) + d\sigma_+}{n} \sim O(1).$$

This confirms that the estimator is stable and bounded.

$\square$

*Remark* D.5 (The Role of Sequential History in Covariance Estimation). Similar to the mean estimation analysis, we provide a rigorous comparison between the base case and the sequential case for covariance estimation.

First, we derive the bound for a cold-start estimator starting at $t = 1$ (where $\nu'^{(\text{base})}_{1,l} = n - d - 1$). The spectral error at the first step is bounded by sampling noise:

$$E_{1,l}^{(\boldsymbol{\Sigma},\text{base})} = \mathbb{E}\left\| \frac{1}{n} \boldsymbol{S}_{1,l}^{(\text{cur})} - \boldsymbol{\Sigma}_{1,l}^{(\text{cur})} \right\|_2 \leq \frac{C_2 \sigma_+(\sqrt{dn} + d)}{\nu'^{(\text{base})}_{1,l}}.$$

For $t > 1$, unrolling the recursion from $t$ down to $j = 2$ yields the full bound for base case:

$$
\begin{aligned}
E_{t,l}^{(\boldsymbol{\Sigma},\text{base})} &\leq (\Pi_{j=2}^{t}\beta_{j}^{(\text{base})})E_{1,l}^{(\boldsymbol{\Sigma},\text{base})} + \sum_{j=2}^{t}(\Pi_{k=j+1}^{t}\beta_{k}^{(\text{base})})B_{j}^{(\text{base})} \\
&\leq \frac{\nu_{1,l}^{\prime(\text{base})}}{\nu_{t,l}^{\prime(\text{base})}}E_{1,l}^{(\boldsymbol{\Sigma},\text{base})} + \frac{C_2\sigma_+(\sqrt{dn}+d)(t-1)}{\nu_{t,l}^{\prime(\text{base})}} + \frac{1}{\nu_{t,l}^{\prime(\text{base})}}\sum_{j=2}^{t}\nu_{j-1,l}^{\prime(\text{base})}\Delta_j^{(\boldsymbol{\Sigma})} \\
&\quad + \frac{d\sigma_+(t-1)}{\nu_{t,l}^{\prime(\text{base})}} + \frac{2n}{\nu_{t,l}^{\prime(\text{base})}}\sum_{j=2}^{t}(D_j^{(\boldsymbol{\mu})})^2 + \frac{2n}{\nu_{t,l}^{\prime(\text{base})}}\sum_{j=2}^{t}(\text{MSE}_{j-1,l}^{(\boldsymbol{\mu},\text{base})})^2 \\
&\leq \frac{\nu_{1,l}^{\prime(\text{base})}}{\nu_{t,l}^{\prime(\text{base})}}\cdot\frac{C_2\sigma_+(\sqrt{dn}+d)}{\nu_{1,l}^{\prime(\text{base})}} + \frac{C_2\sigma_+(\sqrt{dn}+d)(t-1)}{\nu_{t,l}^{\prime(\text{base})}} + \frac{1}{\nu_{t,l}^{\prime(\text{base})}}\sum_{j=2}^{t}\nu_{j-1,l}^{\prime(\text{base})}\Delta_j^{(\boldsymbol{\Sigma})} \\
&\quad + \frac{d\sigma_+(t-1)}{\nu_{t,l}^{\prime(\text{base})}} + \frac{2n}{\nu_{t,l}^{\prime(\text{base})}}\sum_{j=2}^{t}(D_j^{(\boldsymbol{\mu})})^2 + \frac{2n}{\nu_{t,l}^{\prime(\text{base})}}\sum_{j=2}^{t}(\text{MSE}_{j-1,l}^{(\boldsymbol{\mu},\text{base})})^2 \\
&\leq \frac{C_2\sigma_+(\sqrt{dn}+d)t}{tn-d-1} + \frac{d\sigma_+(t-1)}{tn-d-1} \\
&\quad + O\left(\frac{\sum_{j=2}^{t}(j-1)\Delta_j^{(\boldsymbol{\Sigma})}}{t}\right) + O\left(\frac{\sum_{j=2}^{t}(D_j^{(\boldsymbol{\mu})})^2}{t}\right) + O\left(\frac{\sum_{j=2}^{t}(\text{MSE}_{j-1,l}^{(\boldsymbol{\mu},\text{base})})^2}{t}\right).
\end{aligned}
\tag{46}
$$

For the sequential estimator, the bound derived in Equation (39) is

$$
\begin{aligned}
E_{t,l}^{(\boldsymbol{\Sigma},\text{cur})} &\leq (\Pi_{j=1}^{t}\beta_{r+j})E_{r,l}^{(\boldsymbol{\Sigma},\text{pre})} + \sum_{j=1}^{t}(\Pi_{k=j+1}^{t}\beta_{r+k})B_j \\
&\leq \frac{\nu_{r,l}^{\prime}}{\nu_{r+t,l}^{\prime}}E_{r,l}^{(\boldsymbol{\Sigma},\text{pre})} + \frac{C_2\sigma_+(\sqrt{dn}+d)t}{\nu_{r+t,l}^{\prime}} + \frac{1}{\nu_{r+t,l}^{\prime}}\sum_{j=1}^{t}\nu_{r+j-1,l}^{\prime}\Delta_j^{(\boldsymbol{\Sigma},\text{cur})} \\
&\quad + \frac{d\sigma_+t}{\nu_{r+t,l}^{\prime}} + \frac{2n}{\nu_{r+t,l}^{\prime}}\sum_{j=1}^{t}(D_j^{(\boldsymbol{\mu},\text{cur})})^2 + \frac{2n}{\nu_{r+t,l}^{\prime}}\sum_{j=1}^{t}(\text{MSE}_{j-1,l}^{(\boldsymbol{\mu},\text{cur})})^2.
\end{aligned}
$$

And the previous phase (length $r$) is treated as a base case process. We expand the inherited term $E_{r,l}^{(\boldsymbol{\Sigma},\text{pre})}$ using the logic derived above:

$$
\begin{aligned}
E_{r,l}^{(\boldsymbol{\Sigma},\text{pre})} &\leq \frac{C_2\sigma_+(\sqrt{dn}+d)r}{\nu_{r,l}^{\prime}} + \frac{d\sigma_+(r-1)}{\nu_{r,l}^{\prime}} \\
&\quad + \frac{1}{\nu_{r,l}^{\prime}}\sum_{j=2}^{r}\nu_{j-1,l}^{\prime}\Delta_j^{(\boldsymbol{\Sigma},\text{pre})} + \frac{2n}{\nu_{r,l}^{\prime}}\sum_{j=2}^{r}(D_j^{(\boldsymbol{\mu},\text{pre})})^2 + \frac{2n}{\nu_{r,l}^{\prime}}\sum_{j=2}^{r}(\text{MSE}_{j-1,l}^{(\boldsymbol{\mu},\text{pre})})^2.
\end{aligned}
\tag{47}
$$

Substituting Equation (47) into the recursive bound Equation (39), the terms merge naturally. Specifically, the constant terms from the previous and current phases combine as $\frac{\nu_{r,l}^{\prime}}{\nu_{r+t,l}^{\prime}}\cdot\frac{C_3\cdot r}{\nu_{r,l}^{\prime}} + \frac{C_3\cdot t}{\nu_{r+t,l}^{\prime}} = \frac{C_3(r+t)}{\nu_{r+t,l}^{\prime}}$, where $C_3 = C_2\sigma_+(\sqrt{dn}+d)r$.

The drift terms similarly concatenate into global sums:

$$
\begin{aligned}
E_{t,l}^{(\boldsymbol{\Sigma},\text{cur})} &\leq \frac{\nu'_{r,l}}{\nu'_{r+t,l}} \cdot \left( \frac{C_2\sigma_+(\sqrt{dn}+d)r}{\nu'_{r,l}} + \frac{d\sigma_+(r-1)}{\nu'_{r,l}} \right) + \frac{C_2\sigma_+(\sqrt{dn}+d)t}{\nu'_{r+t,l}} + \frac{d\sigma_+ t}{\nu'_{r+t,l}} \\
&\quad + \frac{\nu'_{r,l}}{\nu'_{r+t,l}} \cdot \left( \frac{1}{\nu'_{r,l}} \sum_{j=2}^{r} \nu'_{j-1,l}\Delta_j^{(\boldsymbol{\Sigma},\text{pre})} + \frac{2n}{\nu'_{r,l}} \sum_{j=2}^{r} (D_j^{(\boldsymbol{\mu},\text{pre})})^2 + \frac{2n}{\nu'_{r,l}} \sum_{j=2}^{r} (\text{MSE}_{j-1,l}^{(\boldsymbol{\mu},\text{pre})})^2 \right) \\
&\quad + \frac{1}{\nu'_{r+t,l}} \sum_{j=1}^{t} \nu'_{r+j-1,l}\Delta_j^{(\boldsymbol{\Sigma},\text{cur})} + \frac{2n}{\nu'_{r+t,l}} \sum_{j=1}^{t} (D_j^{(\boldsymbol{\mu},\text{cur})})^2 + \frac{2n}{\nu'_{r+t,l}} \sum_{j=1}^{t} (\text{MSE}_{j-1,l}^{(\boldsymbol{\mu},\text{cur})})^2 \\
&= \frac{C_2\sigma_+(\sqrt{dn}+d)(r+t)}{\nu'_{r+t,l}} + \frac{d\sigma_+(r+t-1)}{\nu'_{r+t,l}} \\
&\quad + \frac{1}{\nu'_{r+t,l}} \sum_{k=2}^{r+t} \nu'_{k-1}\Delta_k^{(\boldsymbol{\Sigma})} + \frac{2n}{\nu'_{r+t,l}} \sum_{k=2}^{r+t} (D_k^{(\boldsymbol{\mu})})^2 + \frac{2n}{\nu'_{r+t,l}} \sum_{k=2}^{r+t} (\text{MSE}_{k-1,l}^{(\boldsymbol{\mu})}) \\
&\leq \frac{C_2\sigma_+(\sqrt{dn}+d)(r+t)}{(r+t)n-d-1} + \frac{d\sigma_+(r+t-1)}{(r+t)n-d-1} \\
&\quad + O\left( \frac{\sum_{k=2}^{r+t}(k-1)\Delta_k^{(\boldsymbol{\Sigma})}}{r+t} \right) + O\left( \frac{\sum_{k=2}^{r+t}(D_k^{(\boldsymbol{\mu})})^2}{r+t} \right) + O\left( \frac{\sum_{k=2}^{r+t}(\text{MSE}_{k-1,l}^{(\boldsymbol{\mu})})^2}{r+t} \right).
\end{aligned}
\tag{48}
$$

Note: In the final step, we unified the indices into a global step $k$, where $\nu'_{r+t,l} = n(r+t) - d - 1$.

**Conclusion: Curve Shift and Trade-off.** Comparing the base case (Equation (46)) and sequential case (Equation (48)) bounds, the dominant error terms share the functional form $f(N) \approx \frac{C \cdot N}{Nn-K}$, where $N$ is the accumulated sample count ($t$ for base case, $r + t$ for sequential). Consider the function $f(x) = \frac{x}{ax-b}$ for $a, b > 0$. Since $f'(x) = \frac{-b}{(ax-b)^2} < 0$, this function is strictly decreasing for $ax > b$. The sequential estimator effectively replaces the argument $t$ with $r + t$. Since $r + t > t$, we have $f(r + t) < f(t)$. Thus, the history $\nu_{r,l}$ not only **shifts the convergence curve to the left** but also guarantees a strictly lower error bound due to the monotonic decay of the variance term with respect to degrees of freedom. This "Inertia" allows the estimator to bypass the high-error region of the curve (where degrees of freedom are low), ensuring stability from the very first step of the new phase.

However, inheriting prior hyperparameters (e.g., $\nu_{r,l}, \kappa_{r,l}, \boldsymbol{m}_{r,l}, \boldsymbol{\Sigma}_{r,l}$) from previous phase also introduces a trade-off: the sequential estimator inherits the accumulated **mean and covariance drifts** from the previous phase. These drift terms, representing the tracking lag, could theoretically increase the error if the distribution shifts rapidly. Nevertheless, provided the global drifts decay as $(D_j^{(\boldsymbol{\mu},\text{cur})})^2 \leq C_D j^{-(2+\epsilon)}$ and $\Delta_j^{(\boldsymbol{\Sigma},\text{cur})} \leq C_\Delta j^{-(1+\epsilon')}$ for constants $\epsilon, \epsilon' > 0$, these drift penalties remain bounded. Consequently, similar to the mean estimation case, the benefit of suppressing the early-stage variance significantly outweighs the cost of the tracking lag, leading to a more stable and robust estimation process.

Furthermore, the covariance bound in Equation (45) includes an $O(1)$ spectral-norm finite-sample floor, but it remains bounded, and warm-start still reduces the remaining terms via a larger effective sample size. Unlike static estimation, the term $\mathbb{E}\|\cdot\|_2$ bounds the **worst-case fluctuation** of the estimator due to the stochasticity of the incoming mini-batches. It essentially characterizes a "noise ball" within which the estimator oscillates around the true covariance matrix.

## D.5. Proof of Theorem 3.8

**Lemma D.6** (Hypercontractivity; Theorem 5.10 in (Janson, 1997)). *Let $\boldsymbol{z} \sim \mathcal{N}(\boldsymbol{0}, \boldsymbol{I}_d)$ be a standard Gaussian random vector in $\mathbb{R}^d$. Let $P_{\leq n} : \mathbb{R}^d \to \mathbb{R}$ be a polynomial of degree at most $n$. For any $2 \leq q < \infty$, the following moment inequality holds:*

$$
\|P_{\leq n}(\boldsymbol{z})\|_{L^q} \leq (q-1)^{n/2}\|P_{\leq n}(\boldsymbol{z})\|_{L^2},
\tag{49}
$$

*where $\|P_{\leq n}(\boldsymbol{z})\|_{L^q} := (\mathbb{E}[|P_{\leq n}(\boldsymbol{z})|^q])^{1/q}$ denotes the $L^q$-norm of the random variable.*

**Theorem 3.8\*.** (Asymptotic Properties of Parameter Updates, Detailed Version). *Under Assumption 3.1, 3.4 and 3.7, the parameter update $\boldsymbol{\Delta}_{t,l}$ satisfies the following properties:*

*(a)* ***Bias Convergence:*** *There exists a constant $C_{bias} > 0$, independent of $t$, such that*

$$\mathbb{E}[\|\boldsymbol{\Delta}_{t,l}^{bias}\|_F^2] \leq C_{bias} \cdot \mathrm{MSE}_{t,l}^{(\boldsymbol{\mu})}.$$

*Consequently, the systematic bias component vanishes asymptotically as $\mathrm{MSE}_{t,l}^{(\boldsymbol{\mu})} \to 0$.*

*(b)* ***Bounded Instance-Specific Component Norm:*** *There exists a constant $U_{spec} < \infty$ such that*

$$\mathbb{E}[\|\boldsymbol{\Delta}_{t,l}^{spec}\|_F^2] \leq U_{spec}.$$

*This condition ensures that the magnitude of the instance-specific update component is well-conditioned: it is strictly bounded away from infinity (ensuring stability), effectively regularizing the effective step size.*

*(c)* ***Interference Mitigation:*** *For distinct steps $t \neq k$, let $\mathcal{G}_{t,k,l} := \sum_{i=1}^{n_t} \sum_{j=1}^{n_k} \langle \phi_{t,l}^i, \phi_{k,l}^j \rangle_F$. Then the expected interference $\mathbb{E}[\langle \boldsymbol{\Delta}_{t,l}, \boldsymbol{\Delta}_{k,l} \rangle_F] = \gamma^2 \, \mathbb{E}[\tilde{s}_{t,l}^\top \tilde{s}_{k,l} \cdot \mathcal{G}_{t,k,l}] + o(1)$, where $\tilde{s}_{t,l} := \mathbb{E}_{\mathcal{V}}[\tilde{\zeta}_{t,l}^i | \mathcal{H}]$, $\mathcal{H}$ denotes the history of features $\{\tilde{h}_{\tau,l}^i\}$ up to the current step $t$, $\mathcal{V}$ the corresponding gradients $\{\tilde{v}_{\tau,l}^i\}$, and $o(1)$ vanishes as $t, k \to \infty$. Specifically, the cross-term contributions arising from global mean shifts and stochastic noise vanish asymptotically.*

*Proof.* The update rule at step $t$ is explicitly given by

$$\boldsymbol{\Delta}_{t,l} = -\gamma \sum_{i=1}^{n_t} \tilde{v}_{t,l}^i \phi_{t,l}^i, \quad \text{where} \quad \phi_{t,l}^i := \|\tilde{h}_{t,l}^i\|_2^2 (h_{t,l}^i)^\top \left( \sum_{j=1}^{n_t} h_{t,l}^j (h_{t,l}^j)^\top + \lambda I \right)^{-1}.$$

Decomposing the gradient vector as $\boldsymbol{v}_{t,l}^i = \boldsymbol{\mu}_{t,l} + \boldsymbol{\zeta}_{t,l}^i$, we partition the update into the *instance-specific component* and the *systematic bias component*:

$$\boldsymbol{\Delta}_{t,l} = -\gamma \sum_{i=1}^{n_t} \widehat{\boldsymbol{\Sigma}}_{t,l}^{-1/2} (\boldsymbol{\zeta}_{t,l}^i + \boldsymbol{\mu}_{t,l} - \boldsymbol{m}_{t,l}) \phi_{t,l}^i$$

$$= \underbrace{\left( -\gamma \sum_{i=1}^{n_t} \widehat{\boldsymbol{\Sigma}}_{t,l}^{-1/2} \boldsymbol{\zeta}_{t,l}^i \phi_{t,l}^i \right)}_{\boldsymbol{\Delta}_{t,l}^{spec}} + \underbrace{\left( -\gamma \widehat{\boldsymbol{\Sigma}}_{t,l}^{-1/2} (\boldsymbol{\mu}_{t,l} - \boldsymbol{m}_{t,l}) \sum_{i=1}^{n_t} \phi_{t,l}^i \right)}_{\boldsymbol{\Delta}_{t,l}^{bias}}.$$

### Proof of Part (a): Bias Convergence.

Taking the Frobenius norm of the systematic bias term $\boldsymbol{\Delta}_{t,l}^{bias}$ and applying the rank-1 matrix norm identity $\|\boldsymbol{u}\boldsymbol{v}^\top\|_F^2 = \|\boldsymbol{u}\|_2^2 \|\boldsymbol{v}\|_2^2$, we obtain

$$\left\| \boldsymbol{\Delta}_{t,l}^{bias} \right\|_F^2 = \gamma^2 \left\| \widehat{\boldsymbol{\Sigma}}_{t,l}^{-1/2} (\boldsymbol{\mu}_{t,l} - \boldsymbol{m}_{t,l}) \right\|_2^2 \left\| \sum_{i=1}^{n_t} \phi_{t,l}^i \right\|_2^2.$$

Taking expectations and applying the Cauchy-Schwarz inequality yields

$$\mathbb{E}\big[\|\boldsymbol{\Delta}_{t,l}^{bias}\|_F^2\big] \leq \gamma^2 \sqrt{\mathbb{E}\left[ \left\| \sum_{i=1}^{n_t} \phi_{t,l}^i \right\|_2^4 \right]} \sqrt{\mathbb{E}\left[ \left\| \widehat{\boldsymbol{\Sigma}}_{t,l}^{-1/2} (\boldsymbol{\mu}_{t,l} - \boldsymbol{m}_{t,l}) \right\|_2^4 \right]}.$$

We analyze the two factors on the right-hand side separately. For the first term involving the projection vectors, we apply the inequality $\|\sum_{i=1}^n \boldsymbol{x}_i\|_2^4 \leq n^3 \sum_{i=1}^n \|\boldsymbol{x}_i\|_2^4$ (derived from Jensen's inequality on the convex function $x^4$):

$$\mathbb{E}\left[ \left\| \sum_{i=1}^{n_t} \phi_{t,l}^i \right\|_2^4 \right] \leq n_t^3 \sum_{i=1}^{n_t} \mathbb{E}\left[ \|\phi_{t,l}^i\|_2^4 \right] \leq n_t^4 C_\phi,$$

where the last inequality follows from Assumption 3.7. Thus, the first square root term is bounded by $n_t^2 \sqrt{C_\phi}$.

For the second term, we expand it using the sub-multiplicativity of the spectral norm and Cauchy-Schwarz again:

$$\mathbb{E}\left[\left\|\widehat{\mathbf{\Sigma}}_{t,l}^{-1/2}(\boldsymbol{\mu}_{t,l} - \boldsymbol{m}_{t,l})\right\|_2^4\right] \leq \mathbb{E}\left[\left\|\widehat{\mathbf{\Sigma}}_{t,l}^{-1}\right\|_2^2 \|\boldsymbol{\mu}_{t,l} - \boldsymbol{m}_{t,l}\|_2^4\right] \leq \sqrt{\mathbb{E}\left[\left\|\widehat{\mathbf{\Sigma}}_{t,l}^{-1}\right\|_2^4\right]}\sqrt{\mathbb{E}\left[\|\boldsymbol{\mu}_{t,l} - \boldsymbol{m}_{t,l}\|_2^8\right]}.$$

Under Assumption 3.7, the first factor is bounded by $\sqrt{K_{\mathbf{\Sigma}}}$. To bound the second factor, define the estimation error vector at step $t$ as $\boldsymbol{e}_{t,l} := \boldsymbol{m}_{t,l} - \boldsymbol{\mu}_{t,l}$, where we treat the (unknown) target moment $\boldsymbol{\mu}_{t,l}$ as fixed. Since $\boldsymbol{m}_{t,l}$ is a linear combination of Gaussian gradients, $\boldsymbol{e}_{t,l}$ follows a Gaussian distribution $\mathcal{N}(\boldsymbol{\eta}_{t,l}, \boldsymbol{V}_{t,l})$. We first control the fourth moment of the error norm $\mathbb{E}[\|\boldsymbol{e}_{t,l}\|_2^4]$. Leveraging the standard fourth moment identity for a Gaussian vector, we have $\mathbb{E}[\|\boldsymbol{e}_{t,l}\|_2^4] = (\operatorname{tr}\boldsymbol{V}_{t,l} + \|\boldsymbol{\eta}_{t,l}\|_2^2)^2 + 2\operatorname{tr}(\boldsymbol{V}_{t,l}^2) + 4\boldsymbol{\eta}_{t,l}^\top\boldsymbol{V}_{t,l}\boldsymbol{\eta}_{t,l}$. Recognizing that $\mathrm{MSE}_{t,l}^{(\boldsymbol{\mu})} = \operatorname{tr}\boldsymbol{V}_{t,l} + \|\boldsymbol{\eta}_{t,l}\|_2^2$, and applying the inequalities $\operatorname{tr}(\boldsymbol{V}_{t,l}^2) \leq (\operatorname{tr}\boldsymbol{V}_{t,l})^2$ and $\boldsymbol{\eta}_{t,l}^\top\boldsymbol{V}_{t,l}\boldsymbol{\eta}_{t,l} \leq \|\boldsymbol{\eta}_{t,l}\|_2^2\operatorname{tr}\boldsymbol{V}_{t,l} \leq \frac{1}{2}(\|\boldsymbol{\eta}_{t,l}\|_2^4 + (\operatorname{tr}\boldsymbol{V}_{t,l})^2)$, we obtain

$$\mathbb{E}[\|\boldsymbol{e}_{t,l}\|_2^4] \leq (\mathrm{MSE}_{t,l}^{(\boldsymbol{\mu})})^2 + 2(\mathrm{MSE}_{t,l}^{(\boldsymbol{\mu})})^2 + 2(\mathrm{MSE}_{t,l}^{(\boldsymbol{\mu})})^2 = 5(\mathrm{MSE}_{t,l}^{(\boldsymbol{\mu})})^2.$$

Next, we bound the eighth moment $\mathbb{E}[\|\boldsymbol{e}_{t,l}\|_2^8]$. Since $\boldsymbol{e}_{t,l}$ is a Gaussian vector, it can be represented as an affine transformation of a standard Gaussian vector. Let $\boldsymbol{e}_{t,l} = \boldsymbol{V}_{t,l}^{1/2}\boldsymbol{z} + \boldsymbol{\eta}_{t,l}$, where $\boldsymbol{z} \sim \mathcal{N}(\boldsymbol{0}, \boldsymbol{I}_d)$. Consider a polynomial of degree 2 in the standard Gaussian vector $P_{\leq 2}(\boldsymbol{z}) = \|\boldsymbol{e}_{t,l}\|_2^2 = \|\boldsymbol{V}_{t,l}^{1/2}\boldsymbol{z} + \boldsymbol{\eta}_{t,l}\|_2^2$, we apply the Hypercontractivity inequality (Lemma D.6) with $q = 4$:
$$\|P_{\leq 2}\|_{L^4} \leq (4-1)^{2/2}\|P_{\leq 2}\|_{L^2} = 3\|P_{\leq 2}\|_{L^2}.$$

Converting back to expectations:

$$(\mathbb{E}[\|\boldsymbol{e}_{t,l}\|_2^8])^{1/4} \leq 3(\mathbb{E}[\|\boldsymbol{e}_{t,l}\|_2^4])^{1/2} \implies \mathbb{E}[\|\boldsymbol{e}_{t,l}\|_2^8] \leq 81(\mathbb{E}[\|\boldsymbol{e}_{t,l}\|_2^4])^2.$$

Substituting the bound for the fourth moment:

$$\sqrt{\mathbb{E}[\|\boldsymbol{e}_{t,l}\|_2^8]} \leq 9 \cdot \mathbb{E}[\|\boldsymbol{e}_{t,l}\|_2^4] \leq 9 \cdot 5(\mathrm{MSE}_{t,l}^{(\boldsymbol{\mu})})^2 = 45(\mathrm{MSE}_{t,l}^{(\boldsymbol{\mu})})^2.$$

Combining all terms, we obtain the final bound for the bias term:

$$\mathbb{E}\left[\|\boldsymbol{\Delta}_{t,l}^{\mathrm{bias}}\|_F^2\right] \leq \gamma^2(n_t^2\sqrt{C_{\boldsymbol{\phi}}})(K_\Sigma)^{1/4}(45(\mathrm{MSE}_{t,l}^{(\boldsymbol{\mu})})^2)^{1/2}.$$

Defining $C_{\mathrm{bias}} := 3\gamma^2 n_t^2\sqrt{5C_{\boldsymbol{\phi}}}(K_{\mathbf{\Sigma}})^{1/4}$ concludes the proof for part (a).

**Proof of Part (b): Bounded Instance-Specific Component Norm.**

We analyze the expected squared Frobenius norm of the instance-specific component $\boldsymbol{\Delta}_{t,l}^{\mathrm{spec}} = -\gamma\sum_{i=1}^{n_t}\tilde{\boldsymbol{\zeta}}_{t,l}^i\boldsymbol{\phi}_{t,l}^i$, where $\tilde{\boldsymbol{\zeta}}_{t,l}^i := \widehat{\mathbf{\Sigma}}_{t,l}^{-1/2}\boldsymbol{\zeta}_{t,l}^i$. To establish the stability bound $U_{\mathrm{spec}}$, we bound the expected magnitude $\mathbb{E}[\|\boldsymbol{\Delta}_{t,l}^{\mathrm{spec}}\|_F^2]$. Applying the Cauchy-Schwarz inequality to the expectation separates the whitened gradient from the projection factor:

$$\mathbb{E}\left[\left\|\boldsymbol{\Delta}_{t,l}^{\mathrm{spec}}\right\|_F^2\right] \leq \gamma^2 n_t\sum_{i=1}^{n_t}\mathbb{E}\left[\left\|\tilde{\boldsymbol{\zeta}}_{t,l}^i\right\|_2^2\|\boldsymbol{\phi}_{t,l}^i\|_2^2\right] \leq \gamma^2 n_t\sum_{i=1}^{n_t}\sqrt{\mathbb{E}\left[\left\|\tilde{\boldsymbol{\zeta}}_{t,l}^i\right\|_2^4\right]\mathbb{E}\left[\|\boldsymbol{\phi}_{t,l}^i\|_2^4\right]}.$$

The projection factor $\mathbb{E}[\|\boldsymbol{\phi}_{t,l}^i\|_2^4] \leq C_{\boldsymbol{\phi}}$ is bounded by Assumption 3.7. For the whitened vector $\tilde{\boldsymbol{\zeta}}_{t,l}^i = \widehat{\mathbf{\Sigma}}_{t,l}^{-1/2}\boldsymbol{\zeta}_{t,l}^i$, we bound its fourth moment. Using $\|\boldsymbol{A}\boldsymbol{x}\|_2 \leq \|\boldsymbol{A}\|_2\|\boldsymbol{x}\|_2$ and Cauchy-Schwarz:

$$\mathbb{E}[\|\tilde{\boldsymbol{\zeta}}_{t,l}^i\|_2^4] \leq \mathbb{E}[\|\widehat{\mathbf{\Sigma}}_{t,l}^{-1}\|_2^2\|\boldsymbol{\zeta}_{t,l}^i\|_2^4] \leq \sqrt{\mathbb{E}[\|\widehat{\mathbf{\Sigma}}_{t,l}^{-1}\|_2^4]}\sqrt{\mathbb{E}[\|\boldsymbol{\zeta}_{t,l}^i\|_2^8]}.$$

The first term is bounded by $\sqrt{K_{\mathbf{\Sigma}}}$ (Assumption 3.7). For the second term, we recall that $\boldsymbol{\zeta}_{t,l}^i = \boldsymbol{v}_{t,l}^i - \boldsymbol{\mu}_{t,l}$ follows a zero-mean Gaussian distribution $\mathcal{N}(\boldsymbol{0}, \mathbf{\Sigma}_{t,l})$. The fourth moment of its Euclidean norm satisfies the standard identity:

$$\mathbb{E}[\|\boldsymbol{\zeta}_{t,l}^i\|_2^4] = (\operatorname{tr}(\mathbf{\Sigma}_{t,l}))^2 + 2\operatorname{tr}(\mathbf{\Sigma}_{t,l}^2).$$

Under Assumption 3.1(b), the eigenvalues of $\boldsymbol{\Sigma}_{t,l}$ are bounded by $\sigma_+$. Consequently, $\operatorname{tr}(\boldsymbol{\Sigma}_{t,l}) \leq d\sigma_+$ and $\operatorname{tr}(\boldsymbol{\Sigma}_{t,l}^2) \leq (\operatorname{tr}(\boldsymbol{\Sigma}_{t,l}))^2 \leq (d\sigma_+)^2$. Substituting these bounds yields $\mathbb{E}[\|\boldsymbol{\zeta}_{t,l}^i\|_2^4] \leq 3(d\sigma_+)^2$.

Next, to bound the eighth moment, we apply the Hypercontractivity inequality (Lemma D.6) with $q = 4$ to the polynomial of degree 2 in the standard Gaussian vector $Q_{\leq 2}(\boldsymbol{z}) = \|\boldsymbol{\zeta}_{t,l}\|_2^2 = \|\boldsymbol{\Sigma}_{t,l}^{1/2}\boldsymbol{z}\|_2^2$:

$$\|Q_{\leq 2}\|_{L^4} \leq 3\|Q_{\leq 2}\|_{L^2} \implies \mathbb{E}[\|\boldsymbol{\zeta}_{t,l}^i\|_2^8] \leq 81(\mathbb{E}[\|\boldsymbol{\zeta}_{t,l}^i\|_2^4])^2 \leq 81(3(d\sigma_+)^2)^2.$$

Since all terms are finite, the total signal energy is bounded by a constant $U_{\text{spec}}$. Combining these results, the expected update magnitude is bounded by

$$\mathbb{E}\left[\left\|\boldsymbol{\Delta}_{t,l}^{\text{spec}}\right\|_F^2\right] \leq 3\gamma^2 n_t^2 \sqrt{C_\phi}(K_{\boldsymbol{\Sigma}})^{1/4}(9^{1/4}d\sigma_+).$$

Defining $U_{\text{spec}} = 3\gamma^2 n_t^2 \sqrt{C_\phi}(9K_{\boldsymbol{\Sigma}})^{1/4}d\sigma_+ < \infty$ completes the proof.

**Proof of Part (c): Interference Mitigation.** Let $\mathcal{I}_{tk} := \mathbb{E}[\langle \boldsymbol{\Delta}_{t,l}, \boldsymbol{\Delta}_{k,l} \rangle_F] = \mathbb{E}[\operatorname{tr}(\boldsymbol{\Delta}_{t,l}^\top \boldsymbol{\Delta}_{k,l})]$ denote the expected inner product of updates from two distinct edit steps $t \neq k$. Decomposing the updates into instance-specific and bias components via $\boldsymbol{\Delta} = \boldsymbol{\Delta}^{\text{spec}} + \boldsymbol{\Delta}^{\text{bias}}$, we expand the inner product by bilinearity:

$$\mathcal{I}_{tk} = \mathbb{E}\left[\langle \boldsymbol{\Delta}_{t,l}^{\text{spec}}, \boldsymbol{\Delta}_{k,l}^{\text{spec}} \rangle_F\right] + \underbrace{\mathbb{E}\left[\langle \boldsymbol{\Delta}_{t,l}^{\text{spec}}, \boldsymbol{\Delta}_{k,l}^{\text{bias}} \rangle_F\right] + \mathbb{E}\left[\langle \boldsymbol{\Delta}_{t,l}^{\text{bias}}, \boldsymbol{\Delta}_{k,l}^{\text{spec}} \rangle_F\right] + \mathbb{E}\left[\langle \boldsymbol{\Delta}_{t,l}^{\text{bias}}, \boldsymbol{\Delta}_{k,l}^{\text{bias}} \rangle_F\right]}_{\mathcal{R}_{tk} \to 0}.$$

The residual term $\mathcal{R}_{tk}$ accounts for bias interactions. We apply the Cauchy-Schwarz inequality ($\mathbb{E}[|\langle \boldsymbol{A}, \boldsymbol{B} \rangle|_F] \leq \sqrt{\mathbb{E}\|\boldsymbol{A}\|_F^2}\sqrt{\mathbb{E}\|\boldsymbol{B}\|_F^2}$) to bound the magnitude of these cross-terms. Since $\mathbb{E}[\|\boldsymbol{\Delta}_{j,l}^{\text{bias}}\|_F^2] \to 0$ asymptotically (as shown in Part (a)) and $\mathbb{E}[\|\boldsymbol{\Delta}_{j,l}^{\text{spec}}\|_F^2]$ is uniformly bounded by $U_{\text{spec}}$ (from Part (b)) for any $j \in \{t, k\}$, we conclude that $\mathcal{R}_{tk} \to 0$ as $t, k \to \infty$.

We now analyze the dominant characteristic interaction term. Decomposing the whitened gradient as $\tilde{\boldsymbol{\zeta}}_{t,l}^i = \tilde{\boldsymbol{s}}_{t,l} + \tilde{\boldsymbol{\epsilon}}_{t,l}^i$, where $\mathbb{E}_{\mathcal{V}}[\tilde{\boldsymbol{\epsilon}}_{t,l}^i | \mathcal{H}] = \boldsymbol{0}$, we have

$$\mathbb{E}\left[\langle \boldsymbol{\Delta}_{t,l}^{\text{spec}}, \boldsymbol{\Delta}_{k,l}^{\text{spec}} \rangle_F\right] = \gamma^2 \mathbb{E}\left[\left\langle \sum_i \tilde{\boldsymbol{\zeta}}_{t,l}^i \boldsymbol{\phi}_{t,l}^i, \sum_j \tilde{\boldsymbol{\zeta}}_{k,l}^j \boldsymbol{\phi}_{k,l}^j \right\rangle_F\right]$$

$$= \gamma^2 \mathbb{E}\left[\left\langle \sum_i (\tilde{\boldsymbol{s}}_{t,l} + \tilde{\boldsymbol{\epsilon}}_{t,l}^i)\boldsymbol{\phi}_{t,l}^i, \sum_j (\tilde{\boldsymbol{s}}_{k,l} + \tilde{\boldsymbol{\epsilon}}_{k,l}^j)\boldsymbol{\phi}_{k,l}^j \right\rangle_F\right]$$

$$= \gamma^2 \mathbb{E}\left[\left\langle \sum_i \tilde{\boldsymbol{s}}_{t,l}\boldsymbol{\phi}_{t,l}^i, \sum_j \tilde{\boldsymbol{s}}_{k,l}\boldsymbol{\phi}_{k,l}^j \right\rangle_F\right] + \underbrace{\mathcal{E}_{\text{cross}}}_{=0}.$$

The cross-term $\mathcal{E}_{\text{cross}}$ involves interactions with the noise components $\tilde{\boldsymbol{\epsilon}}$. For distinct steps $t$ and $k$, the stochastic noise realizations are conditionally independent. Applying the law of iterated expectations conditioning on $\mathcal{H}$:

$$\mathbb{E}_{\mathcal{H},\mathcal{V}}[\tilde{\boldsymbol{\epsilon}}_{t,l}^i \boldsymbol{\phi}_{t,l}^i] = \mathbb{E}_{\mathcal{H}}[\mathbb{E}_{\mathcal{V}}[\tilde{\boldsymbol{\epsilon}}_{t,l}^i | \mathcal{H}]\boldsymbol{\phi}_{t,l}^i] = \boldsymbol{0}.$$

Consequently, all linear cross-terms in $\mathcal{E}_{\text{cross}}$ vanish. The inner product reduces to the correlation of the characteristic update directions. Using the trace property $\langle \boldsymbol{A}, \boldsymbol{B} \rangle_F = \operatorname{tr}(\boldsymbol{A}^\top \boldsymbol{B})$:

$$\lim_{t,k \to \infty} \mathcal{I}_{tk} = \gamma^2 \mathbb{E}\left[(\tilde{\boldsymbol{s}}_{t,l})^\top \tilde{\boldsymbol{s}}_{k,l} \sum_{i=1}^{n_t} \sum_{j=1}^{n_k} \boldsymbol{\phi}_{k,l}^j (\boldsymbol{\phi}_{t,l}^i)^\top\right].$$

This confirms that spurious interference from systematic bias and stochastic noise are eliminated. The asymptotic interference is determined solely by the geometric alignment of the characteristic directions and input features.

$\square$

**Corollary 3.9\*.** (Bounded Update Magnitude, Detailed Version). *As a direct consequence of Theorem 3.8(a) and (b), the total parameter update $\boldsymbol{\Delta}_{t,l}$ satisfies the following second-moment bound:*

$$\mathbb{E}[\|\boldsymbol{\Delta}_{t,l}\|_F^2] \leq 2C_{bias} \cdot \mathrm{MSE}_{t,l}^{(\boldsymbol{\mu})} + 2U_{spec}.$$

*Since $\mathrm{MSE}_{t,l}^{(\boldsymbol{\mu})}$ is non-increasing and vanishes asymptotically, the update magnitude is uniformly bounded by the initial state:*

$$\sup_{t \geq 1} \mathbb{E}[\|\boldsymbol{\Delta}_{t,l}\|_F^2] \leq C_{\boldsymbol{W}},$$

*where $C_{\boldsymbol{W}} := 2C_{bias} \cdot \mathrm{MSE}_{1,l}^{(\boldsymbol{\mu})} + 2U_{spec}$. This explicit, time-independent upper bound confirms that the parameter drift does not explode but is naturally constrained by the initial uncertainty, which means the normalization mechanism internally enforces stability.*

**Corollary D.7** (Invariance under Lipschitz Transformations). *Suppose that a global non-linear transformation operator $\mathcal{F} : \mathbb{R}^{d \times n} \to \mathbb{R}^{d \times n}$ is applied to the entire batch of whitened gradients $\tilde{\boldsymbol{V}}_{t,l} = [\tilde{\boldsymbol{v}}_{t,l}^1, ..., \tilde{\boldsymbol{v}}_{t,l}^{n_t}]$ before the parameter update computation. The resulting update is $\boldsymbol{\Delta}_{t,l}' = -\gamma \mathcal{F}(\tilde{\boldsymbol{V}}_{t,l}) \boldsymbol{\Phi}_{t,l}^\top$, where $\boldsymbol{\Phi}_{t,l} = [\boldsymbol{\phi}_{t,l}^1, \ldots, \boldsymbol{\phi}_{t,l}^{n_t}]$. If $\mathcal{F}$ is $L_{\mathcal{F}}$-Lipschitz continuous with respect to the Frobenius norm, then the asymptotic properties of Bias Decay, Bounded Update Magnitude, and Interference Mitigation established in Theorem 3.8 are preserved.*

*Proof.* We analyze the transformed update $\boldsymbol{\Delta}_{t,l}'$ at the matrix level. Let $\tilde{\boldsymbol{V}}_{t,l} = \tilde{\boldsymbol{D}}_{t,l} + \tilde{\boldsymbol{Z}}_{t,l}$, where $\boldsymbol{D}_{t,l} = [\tilde{\boldsymbol{e}}_{t,l}, \ldots, \tilde{\boldsymbol{e}}_{t,l}]$ is the rank-1 matrix of the estimation error ($\tilde{\boldsymbol{e}}_{t,l} = \hat{\boldsymbol{\Sigma}}_{t,l}^{-1/2}(\boldsymbol{\mu}_{t,l} - \boldsymbol{m}_{t,l})$) and $\tilde{\boldsymbol{Z}}_{t,l} = [\tilde{\boldsymbol{\zeta}}_{t,l}^1, \ldots, \tilde{\boldsymbol{\zeta}}_{t,l}^{n_t}]$ is the matrix of whitened vectors. We decompose the update by adding and subtracting the term $\mathcal{F}(\tilde{\boldsymbol{Z}}_{t,l})$:

$$\boldsymbol{\Delta}_{t,l}' = \underbrace{\left(-\gamma \mathcal{F}(\tilde{\boldsymbol{Z}}_{t,l}) \boldsymbol{\Phi}_{t,l}^\top\right)}_{\boldsymbol{\Delta}_{t,l}'^{\mathrm{signal}}} + \underbrace{\left(-\gamma (\mathcal{F}(\tilde{\boldsymbol{V}}_{t,l}) - \mathcal{F}(\tilde{\boldsymbol{Z}}_{t,l})) \boldsymbol{\Phi}_{t,l}^\top\right)}_{\boldsymbol{\Delta}_{t,l}'^{\mathrm{bias}}}.$$

**(a) Bias Decay Preservation:** By the Lipschitz property of $\mathcal{F}$ on the Frobenius norm, we have

$$\|\mathcal{F}(\tilde{\boldsymbol{V}}_{t,l}) - \mathcal{F}(\tilde{\boldsymbol{Z}}_{t,l})\|_F \leq L_{\mathcal{F}} \|\tilde{\boldsymbol{V}}_{t,l} - \tilde{\boldsymbol{Z}}_{t,l}\|_F = L_{\mathcal{F}} \|\tilde{\boldsymbol{D}}_{t,l}\|_F.$$

The squared Frobenius norm of the transformed bias is bounded by $\|\boldsymbol{\Delta}_{t,l}'^{\mathrm{bias}}\|_F^2 \leq \gamma^2 L_{\mathcal{F}}^2 \|\tilde{\boldsymbol{D}}_{t,l}\|_F^2 \|\boldsymbol{\Phi}_{t,l}\|_F^2$. Taking expectations and using Cauchy-Schwarz:

$$\mathbb{E}[\|\boldsymbol{\Delta}_{t,l}'^{\mathrm{bias}}\|_F^2] \leq \gamma^2 L_{\mathcal{F}}^2 \sqrt{\mathbb{E}[\|\boldsymbol{\Phi}_{t,l}\|_F^4]} \sqrt{\mathbb{E}[\|\tilde{\boldsymbol{D}}_{t,l}\|_F^4]}.$$

The projection term $\mathbb{E}[\|\boldsymbol{\Phi}_{t,l}\|_F^4] = \mathbb{E}[(\sum_i \|\boldsymbol{\phi}_{t,l}^i\|_2^2)^2] \leq n_t \sum_i \mathbb{E}[\|\boldsymbol{\phi}_{t,l}^i\|_2^4] \leq n_t^2 C_{\boldsymbol{\phi}}$ is bounded under Assumption 3.7. For the second term, $\|\tilde{\boldsymbol{D}}_{t,l}\|_F^4 = n_t^2 \|\tilde{\boldsymbol{e}}_{t,l}\|_2^4$. Following the same logic in Theorem 3.8(a), we derived that $\mathbb{E}[\|\tilde{\boldsymbol{e}}_{t,l}\|_2^4]$ is controlled by the eighth moment of the estimation error and fourth moment of the inverse covariance. Consequently, we obtain the bound for the bias term:

$$\mathbb{E}[\|\boldsymbol{\Delta}_{t,l}'^{\mathrm{bias}}\|_F^2] \leq 3\gamma^2 L_{\mathcal{F}}^2 n_t^2 (K_{\boldsymbol{\Sigma}})^{1/4} \sqrt{5C_{\boldsymbol{\phi}}} \, \mathrm{MSE}_{t,l}^{(\boldsymbol{\mu})}.$$

Since $\mathrm{MSE}_{t,l}^{(\boldsymbol{\mu})} \to 0$, the systematic bias component vanishes asymptotically.

**(b) Bounded Instance-Specific Norm Preservation:** We bound the expected norm of the transformed whitened gradient $\mathbb{E}[\|\boldsymbol{\Delta}_{t,l}'^{\mathrm{spec}}\|_F^2]$ by Cauchy-Schwarz:

$$\mathbb{E}[\|\boldsymbol{\Delta}_{t,l}'^{\mathrm{spec}}\|_F^2] \leq \gamma^2 \sqrt{\mathbb{E}[\|\mathcal{F}(\tilde{\boldsymbol{Z}}_{t,l})\|_F^4] \mathbb{E}[\|\boldsymbol{\Phi}_{t,l}\|_F^4]}.$$

The projection term $\mathbb{E}[\|\boldsymbol{\Phi}_{t,l}\|_F^4]$ is bounded by $n_t^2 C_{\boldsymbol{\phi}}$ under Assumption 3.7. To verify $\mathbb{E}[\|\mathcal{F}(\tilde{\boldsymbol{Z}}_{t,l})\|_F^4]$ is finite, we use the linear growth condition that the instance-specific component satisfies $\|\mathcal{F}(\tilde{\boldsymbol{Z}}_{t,l})\|_F \leq L_{\mathcal{F}} \|\tilde{\boldsymbol{Z}}_{t,l}\|_F + \|\mathcal{F}(\boldsymbol{0})\|_F$. Thus, we have

$$\|\mathcal{F}(\tilde{\boldsymbol{Z}}_{t,l})\|_F^4 \leq 8L_{\mathcal{F}}^4 \|\tilde{\boldsymbol{Z}}_{t,l}\|_F^4 + 8\|\mathcal{F}(\boldsymbol{0})\|_F^4.$$

Expanding the whitened gradient norm: $\|\tilde{Z}_{t,l}\|_F^4 = (\sum_{i=1}^{n_t} \|\tilde{\zeta}_{t,l}^i\|_2^2)^2 \leq n_t \sum_{i=1}^{n_t} \|\tilde{\zeta}_{t,l}^i\|_2^4$. In Theorem 3.8(b), we proved that $\mathbb{E}[\|\tilde{\zeta}_{t,l}^i\|_2^4]$ is uniformly bounded by constants depending on $d, \sigma_+, K_{\boldsymbol{\Sigma}}$. Thus, $\mathbb{E}[\|\boldsymbol{\Delta}_{t,l}'^{\text{signal}}\|_F^2]$ is explicitly bounded by a finite constant $U'_{\text{spec}}$:

$$\mathbb{E}[\|\boldsymbol{\Delta}_{t,l}'^{\text{signal}}\|_F^2] \leq \gamma^2 n_t \sqrt{C_\phi} \sqrt{8 L_{\mathcal{F}}^4 n_t^2 \mathbb{E}[\|\tilde{\zeta}_{t,l}^i\|_2^4] + 8\|\mathcal{F}(\mathbf{0})\|_F^4} < \infty.$$

**(c) Interference Mitigation:** Using the same decomposition as in Theorem 3.8(c), the bias interaction terms $\mathcal{R}_{tk}$ are bounded by products of the form $\sqrt{\mathbb{E}[\|\boldsymbol{\Delta}_{t,l}'^{\text{bias}}\|_F^2] \cdot \mathbb{E}[\|\boldsymbol{\Delta}_{k,l}'^{\text{spec}}\|_F^2]}$ or $\sqrt{\mathbb{E}[\|\boldsymbol{\Delta}_{t,l}'^{\text{bias}}\|_F^2] \cdot \mathbb{E}[\|\boldsymbol{\Delta}_{k,l}'^{\text{bias}}\|_F^2]}$. Since the transformed bias norm decays to zero (from point a) and the magnitude of the transformed instance-specific term is bounded (from point b), these cross-terms $\mathcal{R}_{tk}$ vanish asymptotically. The remaining interference is determined solely by the correlation of the transformed instance-specific component, effectively eliminating systematic drift and stochastic noise. $\qquad\square$

*Remark* D.8 (Robustness under Non-linear Architectures). Corollary D.7 broadens the scope of our theoretical guarantees to scenarios where gradients undergo further non-linear transformations. This result applies not only to component-wise activations (e.g., ReLU or GeLU), but also to operations that mix features or samples, such as linear projections and residual connections. As long as the subsequent transformation is Lipschitz continuous—a condition satisfied by standard network layers with bounded operator norms—the normalization mechanism preserves bias decay and asymptotic orthogonality of the parameter updates, independent of the subsequent network architecture.

*Remark* D.9 (Comparative Analysis: The Necessity of Normalization). To rigorously justify the design of normalization mechanism, we formulate the optimization as a Ridge Regression problem: $\min_{\boldsymbol{\Delta}_{t,l}} \|\boldsymbol{\Delta}_{t,l} \boldsymbol{H}_{t,l} - \boldsymbol{V}_{t,l}\|_F^2 + \lambda\|\boldsymbol{\Delta}_{t,l}\|_F^2$. We compare the update dynamics of the **Baseline (Raw Gradient)** versus **Ours (Normalized)**, distinguished by their target construction $\boldsymbol{V}_{t,l}$. With the decomposition $\boldsymbol{\zeta}_{t,l}^i = \boldsymbol{v}_{t,l}^i - \boldsymbol{\mu}_{t,l}$, the updates are

$$\textbf{Baseline:} \quad \boldsymbol{\Delta}_{t,l}^{\text{raw}} = -\gamma \sum_i ( \underbrace{\boldsymbol{\mu}_{t,l}}_{\text{Global Mean}} + \underbrace{\boldsymbol{\zeta}_{t,l}^i}_{\text{Centered Gradient}} )\phi_{t,l}^i,$$

$$\textbf{Ours:} \quad \boldsymbol{\Delta}_{t,l}^{\text{norm}} = -\gamma \sum_i \widehat{\boldsymbol{\Sigma}}_{t,l}^{-1/2}(\underbrace{\boldsymbol{\mu}_{t,l} - \boldsymbol{m}_{t,l}}_{\text{Bias Residual}} + \underbrace{\boldsymbol{\zeta}_{t,l}^i}_{\text{Centered Gradient}} )\phi_{t,l}^i.$$

**Dimension 1: Edit Efficacy vs. Global Compromise.**

- **Baseline:** The term $\boldsymbol{\mu}_{t,l}$ represents the "average gradient direction" satisfying the entire edit dataset $\mathcal{D}$ given the current model state. It acts as a global consensus direction. For any specific instance update, this global component is often a "compromise" that is suboptimal or even contradictory to the instance-specific requirement. Consequently, the baseline update is persistently dragged towards this population-level mean direction, creating a conflict: to learn the new knowledge (contained in $\boldsymbol{\zeta}_{t,l}^i$), the model must redundantly adapt to the global average. This dilutes the parameter update norm allocated to the instance-specific requirement.

- **Ours:** By explicitly subtracting the global consensus via $\boldsymbol{v}_{t,l}^i - \boldsymbol{m}_{t,l}$, the update becomes primarily driven by the centered gradient $\boldsymbol{\zeta}_{t,l}^i$. As shown in Theorem 3.8(a), the systematic bias vanishes asymptotically. This decoupling ensures that the update capacity is allocated almost exclusively to resolving instance-specific discrepancies, maximizing the effectiveness of the current edit.

**Dimension 2: Optimization Efficiency and Adaptive Norm Rescaling.**

- **Baseline:** The magnitude $\|\boldsymbol{\Delta}_{t,l}^{\text{raw}}\|_F$ is heavily influenced by the local curvature of the loss landscape, reflected in the spectrum of the covariance $\boldsymbol{\Sigma}_{t,l}$. In ill-conditioned regimes, the update norm concentrates on a few dominant directions (large eigenvalues of $\boldsymbol{\Sigma}_{t,l}$), while directions with small eigenvalues contribute little to it. This leads to a "vanishing update" problem in minor directions, potentially hindering the learning of fine-grained features. As a result, the baseline update often fails to effectively learn the new knowledge in a single step, leading to under-fitting of current batch.

- **Ours:** The whitening matrix $\widehat{\boldsymbol{\Sigma}}_{t,l}^{-1/2}$ acts as a preconditioner, effectively transforming the regression target $\boldsymbol{V}_{t,l}$ to have approximately homoscedastic errors. This transformation satisfies the conditions of the Gauss-Markov theorem, rendering the Ridge Regression an approximation of the Best Linear Unbiased Estimator (BLUE). Crucially, this mechanism induces "Adaptive Norm Rescaling": it boosts updates in weak eigen-directions (amplifying weak signal) while suppressesing them in directions with large eigenvalues (dampening noise). As proven in Theorem 3.8(b), the expected instance-specific component norm admits uniform upper bounds, ensuring numerical robustness—neither vanishing nor exploding—regardless of the local landscape curvature.

### Dimension 3: Interference and Orthogonality.

- **Baseline:** The interference $\mathcal{I}_{t,k}^{\mathrm{raw}} = \mathbb{E}[\langle \boldsymbol{\Delta}_{t,l}^{\mathrm{raw}}, \boldsymbol{\Delta}_{k,l}^{\mathrm{raw}} \rangle_F]$ expands to

$$\mathcal{I}_{t,k}^{\mathrm{raw}} = \gamma^2 \mathbb{E}\left[ \sum_{i,j} \left( \underbrace{\boldsymbol{\mu}_{t,l}^\top \boldsymbol{\mu}_{k,l}}_{\text{Mean-Mean}} + \underbrace{(\boldsymbol{\mu}_{t,l}^\top \boldsymbol{\zeta}_{k,l}^j + (\boldsymbol{\zeta}_{t,l}^i)^\top \boldsymbol{\mu}_{k,l})}_{\text{Cross-Terms}} \right) \boldsymbol{\phi}_{k,l}^j (\boldsymbol{\phi}_{t,l}^i)^\top + \sum_{i,j} \underbrace{(\boldsymbol{\zeta}_{t,l}^i)^\top \boldsymbol{\zeta}_{k,l}^j}_{\text{Gradient-Gradient}} \boldsymbol{\phi}_{k,l}^j (\boldsymbol{\phi}_{t,l}^i)^\top \right].$$

Since $\boldsymbol{\mu}_{t,l}$ represents the global tendency, the mean correlation $\langle \boldsymbol{\mu}_{t,l}, \boldsymbol{\mu}_{k,l} \rangle$ consistently accumulates across tasks, causing unrelated tasks to interfere systematically along the global mean direction. Additionally, the cross-terms persist because the projection vector $\boldsymbol{\phi}$ and the centered gradient $\boldsymbol{\zeta}$ are coupled. This systematic overlap acts as a primary driver of catastrophic forgetting, forcing parameters to drift in a shared direction regardless of task semantics.

- **Ours:** As proven in Theorem 3.8(c), mean subtraction eliminates both mean-mean correlations and cross-terms. The normalized interference simplifies to

$$\mathcal{I}_{t,k}^{\mathrm{norm}} = \gamma^2 \mathbb{E}\left[ \sum_{i,j} \underbrace{(\tilde{\boldsymbol{\zeta}}_{t,l}^i)^\top \tilde{\boldsymbol{\zeta}}_{k,l}^j}_{\text{Gradient-Gradient}} \boldsymbol{\phi}_{k,l}^j (\boldsymbol{\phi}_{t,l}^i)^\top \right] + \underbrace{\mathcal{O}(\text{bias})}_{\to 0}.$$

The only remaining interaction corresponds to the correlation of characteristic update directions $\langle \tilde{\boldsymbol{s}}_{t,l}, \tilde{\boldsymbol{s}}_{k,l} \rangle$. For semantically unrelated tasks, this term is negligible. Thus, our method actively removes the systematic interference caused by the global mean.

