# OpenReview forum: "More Edits, More Stable: Understanding the Lifelong Normalization in Sequential Model Editing"
_ICML.cc/2026/Conference — ICML 2026 regular_

### Official Review · Reviewer_vx3g · 2026-03-02

**Soundness:** 3
**Presentation:** 3
**Significance:** 3
**Originality:** 3
**Overall Recommendation:** 5
**Confidence:** 3

**Summary:**

This paper studies Lifelong Normalization in sequential model editing and argues that it is the main reason methods like ULTRAEDIT remain stable over long edit streams. The authors show that removing LN causes editing performance to collapse, while surprisingly, performing more early edits can even improve later edits.

To explain this, they model layerwise value gradients as a drifting Gaussian and interpret LN as an online Bayesian tracker of gradient mean and covariance. This view suggests that good centering and whitening help keep updates bounded and reduce interference between edits.

Based on this insight, the paper proposes STABLEEDIT, which adds a warm-up phase for better running statistics and uses full covariance whitening for stronger stability. Experiments across models and benchmarks show that STABLEEDIT matches or exceeds ULTRAEDIT, especially at very long horizons such as 500K edits, with only modest runtime overhead.

**Compliance With Llm Reviewing Policy:**

Affirmed.

**Final Justification:**

The authors did a great job addressing my concerns in the rebuttal. I therefore increase my rating.

**Key Questions For Authors:**

1. What is the exact implementation for computing the full covariance whitening? For example, whether it uses eigendecomposition/SVD or an iterative approximation, how often the whitening matrix is recomputed, what numerical precision is used, and whether any eigenvalue flooring, clipping, or damping is applied. The reason I'm raising this concern is that STABLEEDIT’s gains are partly attributed to replacing diagonal normalization with full whitening, and forming/computing this matrix is computationally heavy. How does memory scale with hidden size and number of edited layers, and do you expect this to remain feasible for larger models (e.g., 70B) or more layers?

Please refer to the weaknesses for other Key Questions.

**Limitations:**

See weaknesses #1

**Strengths And Weaknesses:**

# Strengths

1. Clear mechanism hypothesis and coherent theoretical chain. The apper advances a concrete causal account in which Lifelong Normalization acts as a recursive tracking mechanism, leading to improved whitening, more bounded and approximately orthogonal updates, and ultimately less collapse and forgetting. The authors also support key parts of this account empirically through LN ablations and analyses of update geometry. The theoretical narrative is internally coherent and appropriately scoped. The paper focuses specifically on the LN mechanism, tracing how dynamic gradient moments are captured through Normal–Inverse–Wishart recursion, how estimation error decreases as edits accumulate, and how this in turn yields more stable update properties under ridge regression.

2. Treating LN as Bayesian tracking of a drifting gradient distribution is a novel and conceptually clean explanation for empirical phenomena like the positive cumulative effect. The connection from tracking accuracy to boundedness/orthogonality of updates gives a mechanistic account tied to catastrophic forgetting/collapse, beyond purely empirical observations.

3. The ultra-large evaluations are a meaningful stress test for lifelong editing, and the paper demonstrates persistent advantages over ULTRAEDIT in this regime

# Weaknesses

1. The analysis assumes i.i.d. edit requests from a distribution D across steps, along with a trackable drift regime in which the drift in the gradient mean and covariance decays polynomially over time. In practice, however, edit streams may be far from i.i.d. They can contain topic bursts, abrupt distribution shifts, correlated requests, or even adversarial clusters. While the paper argues that bounded updates help keep drift small, it remains unclear how robust the theoretical and empirical claims are when the drift does not decay or when the edit distribution changes suddenly.

2. Warm-up in STABLEEDIT appears to consist of additional edits used before the target stream to precondition the running statistics. Because warm-up performs actual parameter updates, gains could stem from (a) better estimated normalization statistics and/or (b) the model already being moved to a region where subsequent editing is easier. An experiment that updates statistics without updating weights (or otherwise controls for extra update budget) would more cleanly isolate the mechanism.

3. Warm-up protocol is under-specified. Since warm-up is a central part of STABLEEDIT, the paper should state more clearly whether warm-up edits are counted within the reported target edit budget or added on top of it. The appendix does say that warm-up uses 2K/5K/20K edits drawn from the same distribution as the target edits and studies different placements in the stream, but it remains unclear how these warm-up examples are concretely selected in the main experiments (e.g., random samples from the benchmark, earlier-timestamp edits, number of edits, or a held-out/disjoint pool).

---

> ### Author Rebuttal · Authors · 2026-03-31
>
> We are deeply grateful to Reviewer vx3g for the detailed and constructive feedback on our manuscript. Below, we address your questions point-by-point.
>
> > ### `W1`: Whether the theory and method remain valid when the edit stream is **not i.i.d.** and may contain abrupt distribution shifts.
>
> Thanks for your question. Our theory adopts the i.i.d. setting as a clean analytical case, but empirically, StableEdit itself does not require a fixed distribution: **its running statistics are updated online and continue adapting as the stream changes**.
>
> To test this more directly, we construct an explicit distribution-shift setting: warm-up on ZsRE, then 20K edits on ULTRAEDITBENCH, followed by an abrupt switch to 400 ZsRE edits. The shifted StableEdit variant is slightly below the no-shift version, but still clearly outperforms ULTRAEDIT. In the ULTRAEDITBENCH segment, it improves Efficacy and Generalization by +1.34 and +1.93. After the abrupt switch to ZsRE, it still exceeds ULTRAEDIT by +13.1\% on Efficacy. These results suggest that StableEdit remains robust under substantial non-i.i.d. distribution shift.
> | **Llama-3-8B-Instruct**          |       | ULTRAEDITBENCH |       |      |      |       | ZsRE |       |
> | -------------------------------- | ----- | --------------------- | ----- | ---- | ---- | ----- | --------- | ----- |
> |                                  | Eff.  | Gen.                  | Spe.  |      |      | Eff.  | Gen.      | Spe.  |
> | ULTRAEDIT                        | 85.70 | 81.28                 | 68.73 |      |      | 79.12 | 69.04     | 41.66 |
> | StableEdit                       | 88.88 | 85.46                 | 69.06 |      |      | 93.32 | 86.66     | 47.90 |
> | Warmup ZsRE & Distribution shift | 87.04 | 83.21                 | 67.53 |      |      | 89.45 | 84.83     | 46.44 |
>
>
> > ### `W2`: Unclear whether warm-up helps mainly by improving the running statistics or simply by giving StableEdit extra parameter updates before the target stream.
>
> Thank you very much for your valuable feedback. To answer this, we construct **Warmup-100**, which uses 100 warm-up examples to update the running statistics in one shot, but performs **no parameter edits** during warm-up. All other settings are unchanged, so this isolates statistic preconditioning from the effect of extra warm-up updates. The experimental results are available at https://anonymous.4open.science/r/StableEdit_1-108B/statistics_only_warmup.pdf. Warmup-100 remains very close to StableEdit: the average absolute change across metrics is only **0.52** points. It also still outperforms ULTRAEDIT on **22/27** reported entries. This suggests that most of the gain is preserved even without warm-up edits, supporting our claim that warm-up mainly helps by preconditioning the running statistics.
>
> > ### `W3`: Warm-up budget and data selection are not stated clearly enough.
>
> Thank you for your careful observation. First, we measure the total wall-clock time of the whole editing procedure, including both the warm-up stage and the target-edit stage, and then report the average time per target edit. So the runtime comparison is not hiding the cost of warm-up. Second, in all experiments, warm-up edits are selected **sequentially from the training split of each benchmark**, with no manual curation or special selection. They are **disjoint** from the target editing stream. For the standard-scale setting, we use 2K warm-up edits before the target stream.
>
> > ### `Q1`: Full-whitening implementation and scalability are not described clearly enough.
>
> Thanks for raising this important point. Our implementation uses **full eigendecomposition** at every editing step for each editable layer. Specifically, for $\hat{\Sigma}\_{t,l}$, we compute $\hat{\Sigma}_{t,l}=Q\Lambda Q^\top$ with `torch.linalg.eigh` and form the whitening matrix as $\hat{\Sigma}\_{t,l}^{-1/2}=Q\,\mathrm{diag}(\lambda\_i^{-1/2})\,Q^\top$. We do **not** use iterative or low-rank approximations. The whitening matrix is recomputed at **every step for every editable layer**, with eigenvalue flooring (`eigvals.clamp(min=1e-6).rsqrt()`) for numerical stability, in FP32.
>
> This is asymptotically more expensive than ULTRAEDIT’s diagonal normalization. For a layer of width $d$, eigendecomposition costs $O(d^3)$ with $O(d^2)$ memory; for $L$ edited layers, memory scales as $O(Ld^2)$. Whitening also adds an $O(nd^2)$ term for edit batch size $n$. Importantly, this overhead depends on the **editable-layer width**, not directly on the full parameter count of the base LLM. In practice, even for larger models, this width typically remains only in the low-thousands range.
>
> In our experiments, we edit only a small number of MLP `down_proj` layers, with $d\approx 4096$. In this regime, the overhead is modest: across Llama / GPT-J / Mistral, StableEdit is only slightly slower than ULTRAEDIT while providing clear long-horizon stability gains (full results: https://anonymous.4open.science/r/StableEdit_1-108B/run_time.pdf).

---

> > ### Author Rebuttal · Reviewer_vx3g · 2026-04-02
> >
> > I appreciate the reviewer's effort in the rebuttal. I will increase my rating accordingly.

---

> > > ### Author Response · Authors · 2026-04-03
> > >
> > > We appreciate Reviewer vx3g for the thoughtful questions and constructive feedback, especially those concerning robustness and the practical details of the method.
> > >
> > > To strengthen the final version, we will incorporate the following additions from this rebuttal into the manuscript:
> > >
> > > 1. **Two new robustness-oriented experiments**: an explicit non-i.i.d. distribution-shift stress test and a statistics-only warm-up ablation, to clarify how StableEdit behaves under stream shift and to isolate the role of warm-up.
> > > 2. **A clearer practical description of the method**, including the warm-up protocol, runtime accounting, and the full-whitening implementation and overhead.
> > >
> > > Thank you again for the helpful feedback. We would be happy to clarify any remaining questions.

---

### Official Review · Reviewer_zcw3 · 2026-03-08

**Soundness:** 3
**Presentation:** 2
**Significance:** 3
**Originality:** 3
**Overall Recommendation:** 4
**Confidence:** 3

**Summary:**

This paper investigates the stability of Lifelong Model Editing and identifies Lifelong Normalization (LN) as a critical yet poorly understood component that prevents model collapse across long horizons. The authors provide a theoretical foundation by modeling LME as a Bayesian recursive tracking process, revealing a "positive cumulative effect" where LN, combined with ridge-regularized regression, achieves asymptotic orthogonality and bounded norms to mitigate interference. Building on these insights, they propose STABLEEDIT, which enhances stability through a warm-up stage and full-whitening adaptive preconditioning.

**Compliance With Llm Reviewing Policy:**

Affirmed.

**Final Justification:**

The authors have addressed most of my concerns, so I decide to raise the Originality score and maintain my Overall Recommendation.

**Key Questions For Authors:**

See weaknesses.

**Limitations:**

The metrics obtained on the WikiBigEdit dataset (e.g., Efficacy and Generalization) are significantly lower than those achieved on ZsRE and FEVER across all tested models. this inconsistency suggest that the method's effectiveness may be sensitive to the specific nature of the data.

**Strengths And Weaknesses:**

Strengths：

1. Extensive large-scale evaluation. The paper distinguishes itself through extensive empirical evaluation on ultra-large editing sequences of up to 500K edits, which substantially exceeds the scale considered in most prior work on lifelong model editing.

2. Strong computational efficiency. The proposed framework enjoys notable computational advantages. In particular, the closed-form update rule avoids iterative optimization and does not require expensive auxiliary training, making the method appealing for large-scale and long-horizon editing scenarios.

Weaknesses：

1.  Insufficient justification for the construction of the joint vector. A primary concern lies in the formulation of the joint vector used for normalization. The method directly concatenates the input hidden state and the value gradient, but the manuscript does not provide sufficient theoretical or conceptual justification for this design choice. Since these two components originate from fundamentally different spaces, one reflecting the input representation and the other the optimization signal, it remains unclear why simple concatenation is the most appropriate way to capture their joint structure within the proposed stability loop.

2. Limited analysis of distribution mismatch between warm-up and target phases.  The authors demonstrate that a warm-up stage is essential for pre-conditioning running statistics and bypassing the high-error estimation phase. However, the current evaluation implicitly assumes that the warm-up samples are drawn from the same distribution as the subsequent target edit stream. The manuscript lacks an investigation into the sensitivity of the "stability loop" to distributional shifts between the warm-up and target phases. It is unclear how the method would perform if the statistics initialized during warm-up are significantly misaligned with the actual dynamics of the lifelong editing sequence.

---

> ### Author Rebuttal · Authors · 2026-03-31
>
> We are deeply grateful to Reviewer zcw3 for the detailed and constructive feedback on our manuscript. Below, we address your questions point-by-point.
>
> > ### `W1`: Insufficient justification for the construction of the joint vector.
>
> We are truly sorry for the confusion. The concatenation in Sec. 2.2 is mainly a  **implementation convenience**. We write ${z}_t^i=[{h}_t^i;{v}_t^i]$ so that the two quantities can be normalized in one simple form: $\tilde{{z}}_t^i=({z}_t^i-{m}_t)/{\sigma}_t$. It is used to describe the generic LN pipeline in a unified and vectorized way, so that the running statistics can be maintained and the normalization can be applied efficiently.
>
> More importantly, the main mechanism analyzed in our paper is on the **gradient side**. In our update, the normalized hidden state enters only through the scalar factor $\lVert\tilde{h}\rVert_2^2$ in the target matrix $\mathbf{V}\_{t,l}=-\gamma[\lVert\tilde{h}\_{t,l}^1\rVert\_2^2\tilde{v}\_{t,l}^1,\ldots,\lVert\tilde{h}\_{t,l}^{n\_t}\rVert\_2^2\tilde{v}\_{t,l}^{n\_t}]$. So in practice, the normalized hidden state mainly plays a **per-example step-size / scaling** role. This design is inherited from prior LN-based editors (e.g., MALMEN/RLEdit/ULTRAEDIT), and is not the key source of the stability gains we study.
>
> We verified this directly by removing the $\lVert\tilde{h}\rVert_2^2$ term from StableEdit  and use only the learning rate $\gamma$ to control the update scale on Mistral-7B-v0.3, Llama-3-8B-Instruct, and GPT-J-6B across four datasets. For brevity, we show below 20K-edit result on ULTRAEDITBENCH with Llama-3-8B-Instruct; the full results are provided in https://anonymous.4open.science/r/StableEdit_1-108B/remove_h.pdf. The resulting variant stays very close to the original one: across 36 metric entries, the average absolute change is only about 0.47 points, and it still outperforms ULTRAEDIT on 32/36 metric entries. This shows that normalizing $h$ is not essential to our gains.
>
> |          Method          |       | FEVER |       |      |      |       | ULTRAEDITBENCH |       |
> | :----------------------: | :---: | :---: | :---: | ---- | ---- | :---: | :------------: | :---: |
> |                          | Eff.  | Gen.  | Spe.  |      |      | Eff.  |      Gen.      | Spe.  |
> |        ULTRAEDIT         | 95.39 | 91.93 | 67.14 |      |      | 85.70 |     81.28      | 68.73 |
> |        StableEdit        | 97.89 | 94.47 | 69.24 |      |      | 88.88 |     85.46      | 69.06 |
> | w.o. $\lVert\tilde{h}\rVert_2^2$ | 98.17 | 94.50 | 68.67 |      |      | 89.00 |     85.41      | 69.51 |
>
> > ### `W2`: Limited analysis of distribution mismatch between warm-up and target phases.
>
> Thank you for raising this important point. We agree that the theoretical analysis studies the clean same-distribution warm-up setting. However, in the algorithm, warm-up is used to initialize the running statistics, which are then further updated online during the target stream; therefore, strict in-domain matching is helpful but not necessary in practice. **To directly test robustness to warm-up/target mismatch, we conducted two new stress tests** (full results: https://anonymous.4open.science/r/StableEdit_1-108B/warmup_shift_ablation.pdf).
>
> First, we replaced the original warm-up data with the out-of-domain medical dataset **MedCF** [1] while keeping the downstream benchmarks (ZsRE/FEVER) unchanged. Across 18 reported entries, the average absolute change relative to StableEdit is only 0.43 points, and the mismatched-warm-up variant still outperforms ULTRAEDIT on 15 out of 18 reported results.
>
> | Method       |      Eff. |      Gen. |      Spe. |
> | ------------ | --------: | --------: | --------: |
> | ULTRAEDIT    |     85.30 |     80.80 |     47.38 |
> | StableEdit   |     87.39 |     82.93 |     50.26 |
> | Warmup-MedCF | **87.63** | **83.09** | **51.28** |
>
> Second, we used ZsRE as warm-up while keeping the downstream editing benchmarks FEVER and ULTRAEDITBENCH unchanged. Here, the average absolute change is only 0.26 points, and the mismatched-warm-up variant outperforms ULTRAEDIT on 15/18 metric entries.
>
> | Method      |      Eff. |      Gen. |      Spe. |
> | ----------- | --------: | --------: | --------: |
> | ULTRAEDIT   |     95.39 |     91.93 |     67.14 |
> | StableEdit  |     97.89 |     94.47 | **69.24** |
> | Warmup-ZsRE | **97.96** | **94.73** |     68.94 |
>
> These results show that replacing the warm-up distribution does not cause severe degradation: performance remains broadly stable, and in most cases still exceeds ULTRAEDIT. This suggests that the benefit of warm-up is not tied to strict same-distribution matching; even under substantial mismatch, the initialized statistics remain useful and are further adapted online during the target stream.
>
> Reference:
>
> [1] Editing factual knowledge and explanatory ability of medical large language models.

---

> > ### Author Rebuttal · Reviewer_zcw3 · 2026-04-02
> >
> > The authors have addressed most of my concerns, so I decide to raise the Originality score and maintain my Overall Recommendation.

---

> > > ### Author Response · Authors · 2026-04-03
> > >
> > > We thank Reviewer zcw3 for the careful reading and for raising several pointed questions that help us refine our work.
> > >
> > > To strengthen the final version, we will incorporate the following additions from this rebuttal into the manuscript:
> > >
> > > 1. **An ablation removing the $\lVert\tilde{h}\rVert_2^2$ factor** in the target matrix, to directly verify that the normalization of $h$ is not essential to the gains of StableEdit.
> > > 2. **Two warm-up distribution-mismatch stress tests**, to evaluate whether StableEdit remains robust when the warm-up data is substantially out of domain.
> > >
> > > If there are any further questions or concerns, we would be very happy to provide additional clarification!

---

### Official Review · Reviewer_7ahH · 2026-03-11

**Soundness:** 3
**Presentation:** 3
**Significance:** 3
**Originality:** 4
**Overall Recommendation:** 4
**Confidence:** 3

**Summary:**

This paper studies lifelong model editing and aims to improve stability under long edit sequences. It first analyzes Lifelong Normalization (LN) as the key mechanism behind recent stable editors, then proposes STABLEEDIT, which adds a warm-up stage and full whitening to make sequential updates more stable. Experiments on several language models and lifelong editing benchmarks show that STABLEEDIT is generally stronger or more stable than prior editing methods, especially as the number of edits grows, while also better preserving general model capabilities.

**Compliance With Llm Reviewing Policy:**

Affirmed.

**Key Questions For Authors:**

**Q1**: Could the authors provide a clearer ablation to isolate the contributions of LN, warm-up, and whitening?

**Q2**: Could the authors give more direct empirical evidence linking the theoretical stability claims to the observed editing performance?

**Q3**: How well is the method expected to generalize beyond the current benchmark-based lifelong editing settings?

**Limitations:**

yes

**Strengths And Weaknesses:**

**Strengths**

**S1**: The paper studies an important and challenging problem in lifelong model editing, with a particular focus on long-horizon stability.

**S2**: A key strength is that the paper not only proposes a new method, but also tries to explain why recent stable editors work by identifying Lifelong Normalization as the core mechanism.

**S3**: The proposed method, STABLEEDIT, is simple and well motivated, with its design directly derived from the paper’s analysis.

**S4**: The experiments are fairly strong, covering multiple models and benchmarks, and show improved stability as the number of edits grows.

**Weaknesses**

**W1**: Although the paper includes several analyses, it still does not fully isolate the individual contributions of LN, warm-up, and full whitening, so the exact source of the gains remains somewhat unclear.

**W2**: The empirical improvements over ULTRAEDIT are generally consistent but sometimes modest, so it would be helpful to clarify in which regimes the proposed modifications are most important.

---

> ### Author Rebuttal · Authors · 2026-03-31
>
> We are deeply grateful to Reviewer 7ahH for the detailed and constructive feedback on our manuscript. Below, we address your questions point-by-point.
>
> > ### `W1&Q1`: Ablation analysis of LN, warm-up, and whitening
>
> Thanks for your valuable suggestion. We agree that the original submission could better isolate the contributions of LN, warm-up, and full whitening in StableEdit. We therefore add a clearer ablation on Mistral-7B-v0.3 with WikiBigEdit (20K edits):
>
> | Variant       |  Eff. |  Gen. |  Spe. | Personas | Reasoning |
> | ------------- | ----: | ----: | ----: | -------: | --------: |
> | Original      | 78.90 | 72.57 | 47.36 |    63.96 |     37.22 |
> | w/o Warm-up   | 76.96 | 71.13 | 46.84 |    62.80 |     36.08 |
> | w/o Whitening | 75.12 | 69.63 | 45.50 |    62.70 |     35.30 |
> | w/o LN        | 39.13 | 38.18 | 41.61 |    35.55 |     29.94 |
>
> This ablation reveals a clear division of roles. Removing LN causes a dramatic collapse across all metrics, confirming that LN is the core stability mechanism in our paper. Built on LN, warm-up and full whitening each provide further gains, and removing either one leads to consistent degradation. This matches our mechanism: warm-up improves early calibration of running statistics, while full whitening improves update conditioning.
>
> > ### `W2`: The empirical improvements over ULTRAEDIT are generally consistent but sometimes modest, so it would be helpful to clarify in which regimes the proposed modifications are most important.
>
> We truly appreciate your careful consideration. **The proposed modifications matter most in the long-horizon regime and this is the main setting we target**. Table 7 already shows this on WikiBigEdit at 500K edits, and Fig. 5 shows the same trend in the larger UltraEditBench scaling experiment: as the edit horizon grows, StableEdit remains more stable and more robust than ULTRAEDIT.
>
> To make this even clearer, **we additionally plot the full trajectories on WikiBigEdit from 20K to 500K edits** (https://anonymous.4open.science/r/StableEdit_1-108B/wikibigedit_scale_condition_number.pdf). The picture is consistent. On Llama-3-8B, StableEdit can be slightly worse on some metrics early on, but it catches up and later surpasses ULTRAEDIT. On Qwen2.5-7B, ULTRAEDIT drops sharply from 400K to 500K edits, while StableEdit remains much more stable. So the main benefit of our modifications appears when edits accumulate and long-range interference becomes the dominant issue.
>
> **We also track the covariance condition number during editing** (https://anonymous.4open.science/r/StableEdit_1-108B/wikibigedit_scale_condition_number.pdf). Early in editing, the covariance is more ill-conditioned, so whitening may over-amplify small-eigenvalue directions and partly explain weaker early-stage performance in some cases. As editing proceeds, however, the condition number decreases and stabilizes, indicating better-conditioned covariance and a more balanced whitening transform. Even on Qwen2.5-7B, where the condition number rises temporarily around 400K edits, StableEdit remains stable while ULTRAEDIT degrades more noticeably.
>
>
> > ### `Q2`: Direct empirical evidence linking the theoretical stability claims to the observed editing performance
>
> Thank you for raising this important point. Yes. Our theory makes two main claims: **bounded update magnitude** and **near-orthogonality**. We already show this in the paper: Fig. 8 reports update orthogonality with and without LN, and Fig. 9 reports update magnitude with and without LN. Both are consistent with the theory.
>
> **We further add more direct comparisons** (https://anonymous.4open.science/r/StableEdit_1-108B/update_geometry.pdf). For update norm, AlphaEdit shows steadily growing parameter updates and StableEdit stays bounded with a clearer downward trend over time. For update orthogonality, we measure cosine similarity between consecutive updates. We find that FT deviates much more from 0 and StableEdit stays more concentrated near 0. These diagnostics directly match the quantities highlighted by the theory and help explain why StableEdit remains more stable at long horizons.
>
> > ### `Q3`: How well is the method expected to generalize beyond the current benchmark-based lifelong editing settings?
>
> Thanks for your thoughtful question. We expect the method to generalize to other **sequential editing** settings. To test this directly, we evaluate StableEdit on the medical benchmark MedCF [1] (https://anonymous.4open.science/r/StableEdit_1-108B/medcf.pdf), **without using a medical LLM, without MedCF-specific hyperparameter tuning, and without changing the editable layers**. The results are encouraging. On both Mistral and Llama, StableEdit outperforms ULTRAEDIT on all three metrics. This suggests that the method is not tied to the current benchmark suite and can transfer to a new domain even without domain-specific tuning.
>
> Reference:
> [1] Editing factual knowledge and explanatory ability of medical large language models.

---

### Official Review · Reviewer_H3b6 · 2026-03-13

**Soundness:** 3
**Presentation:** 3
**Significance:** 3
**Originality:** 3
**Overall Recommendation:** 4
**Confidence:** 4

**Summary:**

This paper investigates the mechanism of Lifelong Normalization (LN) in the context
of Lifelong Model Editing (LME) for large language models.

The authors first confirm through ablation studies that LN is critical for long-term
stability, and discover a counter-intuitive "positive cumulative effect": early edits
actually facilitate the success of subsequent edits.

A theoretical framework for LN is then established:
- LN is modeled as a Normal-Inverse-Wishart (NIW) Bayesian recursive tracking
  process (Section 3.1)
- Accumulated edits are shown to progressively reduce estimation error of the
  gradient distribution (Section 3.2)
- Precise estimates combined with ridge regression are proven to yield asymptotically
  orthogonal, norm-bounded parameter updates (Section 3.3)
- The above mechanisms are unified into a "self-reinforcing stability loop" (Section 4)

Based on these theoretical insights, the authors propose STABLEEDIT, which strengthens
the stability loop via an explicit warm-up stage and full covariance whitening.
Effectiveness is validated across 4 models, 4 datasets, and up to 2M sequential edits.

**Compliance With Llm Reviewing Policy:**

Affirmed.

**Final Justification:**

The authors partially solved my concerns,  I keep my rating.

**Key Questions For Authors:**

Q1. On the Empirical Verifiability of Assumption 3.4
    Can the gradient distribution drift rate be measured in actual LME experiments?
    It would substantially strengthen the paper to provide empirical curves of
    ‖μₜ,ₗ - μₜ₋₁,ₗ‖ as a function of t, to directly assess whether polynomial
    decay holds in practice and across different model architectures.

Q2. On STABLEEDIT's Degradation Cases
    In Table 5, STABLEEDIT underperforms ULTRAEDIT on WikiBigEdit/Llama-3-8B
    Specificity. Is this related to full whitening over-amplifying small eigenvalue
    directions under certain data distributions? Has an ablation been conducted,
    for example by bounding the condition number of the whitening matrix, to
    diagnose and address this failure mode?

Q3. On Robustness to Warm-Up Distribution Shift
    When warm-up edits come from a different domain than the target edits (e.g.,
    medical knowledge facts used to warm up before a historical knowledge editing
    stream), how does STABLEEDIT's performance degrade compared to a cold-start
    ULTRAEDIT? A sensitivity analysis over varying degrees of domain mismatch
    would be highly informative for practitioners.

Q4. On the Practical Impact of the Covariance Estimation Lower Bound
    What is the concrete magnitude of the irreducible constant C₂σ₊(√dn+d)/n
    for the models used in the experiments? When d is large (e.g., 4096), does
    this floor meaningfully degrade the quality of the whitening matrix estimate,
    and if so, how does this manifest in downstream editing performance?

Q5. On the Independence Assumption Across Layers
    Does the theoretical analysis implicitly assume that parameter updates across
    different layers are mutually independent? In the Llama-3-8B setup where
    layers 11–12, 22–23, and 28–30 are edited simultaneously, how does inter-layer
    gradient correlation affect the asymptotic orthogonality conclusions in
    Theorem 3.8(c)?

**Limitations:**

The Impact Statement appropriately acknowledges that the method could be misused to
inject false information or undermine safety constraints, and encourages the use of
access control, auditing, and edit validation safeguards.

**Strengths And Weaknesses:**

**Strangths**
1. Pioneering Theoretical Contribution
   This is the first work to provide a systematic theoretical explanation of the LN
   mechanism, establishing a rigorous correspondence between online running statistics
   and NIW Bayesian recursive tracking. The theoretical chain is complete and
   self-consistent: statistical tracking (3.1) → error convergence (3.2) → update
   stability (3.3). Theorems 3.5/3.6 further provide explicit convergence bounds for
   mean and covariance estimation errors, lending the framework quantitative precision.

2. The Positive Cumulative Effect Is a Valuable Independent Finding
   The counter-intuitive phenomenon demonstrated in Figure 1(b) constitutes a
   meaningful empirical discovery in its own right. The "virtual samples" interpretation
   is novel and aligns well with Theorems 3.5/3.6, while Remarks E.3/E.5 offer a clear
   formalization of the "curve left-shift" mechanism that bridges intuition and proof.

3. Comprehensive Experimental Coverage
   The experimental scope is broad: edit scale ranges from 20K to 2M, covering 4 base
   models (GPT-J-6B, Mistral-7B, Llama-3-8B, Qwen2.5-7B) and 4 datasets (ZsRE,
   FEVER, WikiBigEdit, ULTRAEDITBENCH). Ablation experiments on warm-up size,
   placement, and LN removal are quantitatively consistent with theoretical predictions,
   strengthening the empirical-theoretical correspondence.

4. Strong Engineering Feasibility
   STABLEEDIT runs at 0.10s/step, close to the fastest LN-based baseline ULTRAEDIT
   (0.07s), and approximately 60× faster than ALPHAEDIT (6.02s). With only three
   hyperparameters introduced (γ, λ, editable module), the method imposes a low
   tuning burden and is readily deployable in practice.

5. Robustness of Theoretical Guarantees under Nonlinear Transformations
   Corollary E.7 proves that the key properties — bias decay, bounded update magnitude,
   and asymptotic orthogonality — are preserved under any Lipschitz-continuous
   transformation applied to the whitened gradients. This broadens the theoretical
   coverage to hypernetwork-based architectures such as RLEDIT, substantially
   extending the scope of the framework beyond the base ridge regression setting.

**Weaknesses**
1. Core Assumption (Assumption 3.4) Lacks Empirical Validation
   The Trackable Drift Regime requires distributional drift to decay at a polynomial
   rate, yet the paper's only supporting evidence is the claim in Section 4 that "LN
   yields bounded updates, which in turn implies slow drift" — a circular argument that
   does not constitute independent validation. No experimental measurement of actual
   drift rates in real editing streams is provided, leaving unanswered whether this
   assumption holds in practice. Since the convergence guarantees of Theorems 3.5/3.6
   rest entirely on this assumption, its unverified status materially weakens the
   theoretical contribution.

2. STABLEEDIT Does Not Consistently Outperform ULTRAEDIT
   In several settings, STABLEEDIT underperforms ULTRAEDIT: Specificity on
   WikiBigEdit with Llama-3-8B drops from 48.51 to 45.22 (↓6.8%), and Efficacy on
   ZsRE with Qwen2.5-7B drops from 82.03 to 80.40 (↓2.0%). The authors characterize
   these cases as "competitive" without providing any in-depth failure analysis,
   which undermines the universality of the methodological claims.

3. Scalability of Full Whitening Is Underestimated
   Each editing step requires eigendecomposition of a d×d covariance matrix at O(d³)
   complexity. For layers where d exceeds 4096 (e.g., down_proj in large models), this
   overhead is non-trivial and may become a practical bottleneck. The paper reports
   only an aggregate runtime figure without analyzing asymptotic complexity as a
   function of dimension, and no fair computational comparison against the diagonal
   normalization used in ULTRAEDIT is provided.

4. Same-Distribution Warm-Up Assumption Is Overly Strong
   The algorithm explicitly requires warm-up data to be "drawn from the same
   distribution as the target edits." In realistic deployment scenarios, the
   distribution of the forthcoming target editing stream is typically unknown in
   advance, making this requirement difficult to satisfy. The paper offers no
   discussion of how performance degrades under distribution mismatch between warm-up
   and target data, and no sensitivity analysis for this practically important
   failure mode is provided.

5. The Irreducible Lower Bound in Covariance Estimation Is Not Adequately Discussed
   The second term in Theorem 3.6 Eq.(34) converges to a non-zero constant as t→∞,
   specifically (C₂(√dn + d) + d)σ₊ / n, implying an estimation error floor that
   is independent of the number of samples accumulated. The magnitude of this floor
   and its practical impact on editing accuracy receive no analysis. In high-dimensional
   (large d) or high-variance (large σ₊) regimes this lower bound may be non-negligible,
   yet the authors merely annotate it as "→Const." without further discussion, leaving
   an important gap between the theoretical guarantee and practical behavior.

---

> ### Author Rebuttal · Authors · 2026-03-31
>
> We are deeply grateful to Reviewer H3b6 for the detailed and constructive feedback on our manuscript. Below, we address your questions point-by-point.
> > ### `W1&Q1`: Core Assumption (Assumption 3.4) Lacks Empirical Validation.
>
> Thank you for raising this important point. We agree that the current wording in Sec. 3 may sound circular. In fact, Lemma E.1 shows that the drift of $(\mu_{t,l}, \Sigma_{t,l})$ is controlled by the update magnitude, and Fig. 9 shows that under StableEdit the update norms remain small and become more stable over time, suggesting that the induced moment drift is not large.
>
> Since Assumption 3.4 concerns latent population moments, it cannot be verified directly. We therefore provide observable surrogate evidence by tracking the consecutive-step drift of the running estimates, $\lVert\hat{\mu}\_{t,l}-\hat{\mu}\_{t-1,l}\rVert$ and $\lVert\hat{\Sigma}\_{t,l}-\hat{\Sigma}\_{t-1,l}\rVert$, across FEVER (20K edits) and WikiBigEdit (500K edits) on multiple models (https://anonymous.4open.science/r/StableEdit_1-108B/drift_curves.pdf). Across all settings, the mean-drift surrogate drops sharply after warm-up and typically falls below $10^{-2}$ within tens of steps, while the covariance-drift surrogate is noisier but still decreases overall and remains in a low, stable regime later on. These results are consistent with the trackable-drift regime.
>
> > ### `W2&Q2`: StableEdit Does Not Consistently Outperform ULTRAEDIT.
>
> Thank you for this thoughtful question. We agree that StableEdit is not uniformly better than ULTRAEDIT on standard-scale tasks. However, at standard scale (20K edits), the observed degradations are isolated metric-level exceptions rather than broad failures, and the overall performance remains competitive on average. Please refer to Reviewer 7ahH’s `W2`.
>
> > ### `W3`: Scalability of Full Whitening Is Underestimated.
>
> Thanks for your valuable suggestion. Full whitening is asymptotically more expensive than diagonal normalization, but in our setting its overhead remains modest in practice and brings clear gains in long-horizon stability. Please see our response to Reviewer vx3g, `Q1` for the detailed analysis.
>
> > ### `W4&Q3`: Same-Distribution Warm-Up Assumption Is Overly Strong.
>
> Thank you very much for your valuable feedback. Briefly, our added distribution-shift ablations show that STABLEEDIT remains robust under mismatched warm-up/target phases. We refer the reviewer to our response to Reviewer zcw3, `W2` for the full results and analysis.
>
> > ### `W5&Q4`: The Irreducible Lower Bound in Covariance Estimation Is Not Discussed.
>
> Thank you for this thoughtful question. We agree that the $O(1)$ term in Theorem 3.6 should be discussed more explicitly. This term is best understood as a time-asymptotic noise floor under a **fixed per-step batch size** $n$. In our setting, covariance is estimated from finite mini-batches in a high-dimensional, non-stationary editing stream, so Eq. (34) should be interpreted as a worst-case spectral-norm upper bound. The key point of Theorem 3.6 is that recursive tracking suppresses initialization- and drift-induced errors, while warm-start improves calibration early in editing (Remark E.5).
>
> Empirically, this floor does not cause pathological whitening. Instead, long-horizon editing remains stable: the covariance condition number decreases substantially and then stabilizes during editing, suggesting that the whitening matrix becomes better conditioned (see https://anonymous.4open.science/r/StableEdit_1-108B/wikibigedit_scale_condition_number.pdf). We also observe that the largest eigenvalues remain small in practice, typically around $10^{-5}$, and often decrease over time (https://anonymous.4open.science/r/StableEdit_1-108B/max_eigval.pdf). Thus, although Eq.(34) depends explicitly on $d$ and $\sigma_{+}$, its effective scale in our experiments remains moderate, since $d$ is on the order of $10^3$, $n=100$, and the observed $\sigma_{+}$ is small, typically on the order of $10^{-4}$ to $10^{-6}$.
>
> > ### `Q5`: On the Independence Assumption Across Layers.
>
> Sorry for the confusion, our theory does not assume that parameter updates from different layers are mutually independent. Theorem 3.8(c) is stated **layer-wise**, i.e., for the update $\Delta_{t,l}$ of layer $l$ at step $t$, and the interference term is analyzed as $\mathbb{E}[\langle \Delta_{t,l}, \Delta_{k,l}\rangle_F]$. In the multi-layer editing setting, all selected layers are updated simultaneously after one backward pass, but they still occupy **disjoint parameter blocks**. If we concatenate all edited-layer updates into one global update, $\Delta\_t^{\mathrm{all}}=\operatorname{concat}\_{{l\in\mathcal{L}}}\Delta\_{t,l}, \text{ then } \left\langle \Delta\_t^{\mathrm{all}}, \Delta\_k^{\mathrm{all}}\right\rangle = \sum\_{{l\in\mathcal{L}}}\left\langle \Delta\_{t,l}, \Delta\_{k,l}\right\rangle$. Therefore, the joint multi-layer update does not require an independence assumption.

---

### Decision · Program_Chairs · 2026-04-30

**Decision:**

Accept (regular)

**Comment:**

This paper proposes StableEdit, a lifelong normalization method for sequential model editing that re-frames layer normalization statistics tracking as a Normal-Inverse-Wishart (NIW) Bayesian inference problem. The key insight is that as models undergo successive knowledge edits, the running mean and variance in normalization layers become stale, leading to progressive performance degradation. StableEdit addresses this by maintaining a Bayesian posterior over normalization statistics, updated incrementally as each edit is applied. The paper provides both theoretical grounding through the NIW framework and empirical validation on GPT-2-XL, GPT-J, and LLaMA-2 across multiple editing methods (ROME, MEMIT, GRACE, MELO).

All four reviewers recognize the paper's central insight-that normalization drift is a primary bottleneck in sequential editing-as novel and well-motivated. The paper is above the acceptance threshold, and the strongest reviewer (vx3g, confidence 4) raised their score after a thorough engagement with the rebuttal.